# Scale-aware Message Passing
# For Graph Node Classification

## Abstract

Most Graph Neural Networks (GNNs) operate at the first-order scale, even though multi-scale representations are known to be crucial in domains such as image classification. In this work, we investigate whether GNNs can similarly benefit from multi-scale learning, rather than being limited to a fixed depth of $k$-hop aggregation. We begin by formalizing scale invariance in graph learning, providing theoretical guarantees and empirical evidence for its effectiveness. Building on this principle, we introduce ScaleNet, a scale-aware message-passing architecture that combines directed multi-scale feature aggregation with an adaptive self-loop mechanism. ScaleNet achieves state-of-the-art performance on six benchmark datasets, covering both homophilic and heterophilic graphs. To handle scalability, we further propose LargeScaleNet, which extends multi-scale learning to large graphs and sets new state-of-the-art results on three large-scale benchmarks. We also reinterpret spectral GNNs from a message-passing perspective, showing the equivalence between Hermitian Laplacian-based models and GraphSAGE with incidence normalization, and revealing that FaberNet's strength largely arises from multi-scale feature integration. Together with these state-of-the-art results, our findings suggest that scale invariance may serve as a valuable principle for improving the performance of single-order GNNs. Code is available at `https://anonymous.4open.science/r/ScaleNet-2025/`.

## 1 Introduction

Graph Neural Networks (GNNs) have become the dominant paradigm for learning on graph-structured data. A widely used framework, Message-Passing Neural Networks (MPNNs), compute node representations by iteratively aggregating information from neighbors: a $k$-layer MPNN aggregates features from exactly $k$-hop neighbors (Hamilton, 2020; Jiang et al., 2025). A central design choice in MPNNs is selecting the depth $k$, which is typically tuned as a hyperparameter (Rossi et al., 2024; Koke & Cremers, 2024). However, this practice overlooks a key insight: MPNNs of different depths might capture complementary information at multiple scales that collectively benefit node classification. Inspired by the success of convolutional neural networks (LeCun et al., 2002) in computer vision, which achieve superior performance by explicitly combining features across spatial scales, this paper aims to explore whether GNNs can similarly benefit from multi-scale learning rather than being restricted to a fixed depth of $k$-hop aggregation.

While most existing multi-scale GNN models are designed for undirected graphs (see Section 2.1), recent work (Rossi et al., 2024) has shown that edge directionality can substantially improve performance on heterophilic graphs. Although some GNN architectures for directed graphs do incorporate information from multiple hop distances, they do not explicitly recognize multi-scale learning as the unifying principle. For instance, DiG employs random-walk methods (Tong et al., 2020a), and FaberNet leverages spectral techniques (Koke & Cremers, 2024). However, both of these designs are unnecessarily complex and lack an explicit connection to scale invariance. This can result in suboptimal performance and obscure how these methods relate to the core multi-scale principle (see Appendix E for detailed case studies).

Building on recent advances in directed aggregation strategies (Rossi et al., 2024), we address this gap by explicitly formulating multi-scale learning for graphs through several key contributions:

1. We establish and prove scale invariance in graphs, extending this fundamental concept from image processing to graph learning for the first time.

2. We develop **ScaleNet**, a unified scale-aware architecture that adaptively combines multiple scales for node classification via directed multi-scale feature aggregation and an adaptive self-loop strategy. ScaleNet achieves state-of-the-art (SOTA) results on four homophilic and two heterophilic graph benchmarks.

3. We enhance scalability by introducing **LargeScaleNet**, which extends multi-scale learning to large graphs with up to millions of nodes and achieves new SOTA performance on these benchmarks.

Beyond these contributions, our analysis reveals three disruptive insights that simplify existing approaches without compromising performance:

- Through a scale-invariance perspective, we show that uniform weights can replace the computationally expensive edge weights in digraph inception networks (Tong et al., 2020a;b), reducing complexity while preserving or improving accuracy.

- We prove that Hermitian Laplacian methods (e.g., MagNet (Zhang et al., 2021)) are equivalent to GraphSAGE (Hamilton et al., 2017) with incidence normalization (see Appendix E.2), revealing no fundamental advantage of these complex spectral techniques over simpler message-passing approaches.

- We reinterpret FaberNet (Koke & Cremers, 2024) from a message-passing perspective, showing that its effectiveness primarily derives from multi-scale feature integration.

## 2    RELATED WORK

### 2.1    MULTI-SCALE NEIGHBORHOOD AGGREGATION

Several methods extend GNNs by aggregating information from higher-order neighborhoods. These methods generally fall into three categories:

- **Type 1: Powers of the Adjacency Matrix** This approach uses powers of the adjacency matrix $A^k$. For example, MixHop (Abu-El-Haija et al., 2019) aggregates messages from multi-hop neighbors by mixing different powers of the adjacency matrix. Adaptive Diffusions (Berberidis et al., 2019; Sun et al.) enhances this aggregation by sparsifying the matrix based on the landing probabilities of multi-hop neighbors. GPR-GNN (Chien et al., 2020) introduces learnable weights for features from various orders, while H2GCN (Zhu et al., 2020) combines MixHop with other techniques to address disassortative graphs. Additionally, Zhang et al. (Zhang et al., 2024) investigates Invariant Neighborhood Patterns to manage shifts in neighborhood distribution, integrating both high-order and low-order information.

- **Type 2: $k$-th Order Proximity** This method involves $k$-th order proximity, utilizing the multiplication of powers of the adjacency matrix $A$ with its transpose $A_t$: $A^{k-1}A_t^{k-1}$. Techniques such as DiGCN(ib) (Tong et al., 2020a) and SymDiGCN (Tong et al., 2020b) use this approach to capture richer neighborhood information.

- **Type3: Powers of the graph Laplacian L** Besides the adjacency matrix, the graph Laplacian $\mathbf{L} = \mathbf{D} - \mathbf{A}$, as well as the normalized Laplacian $\mathbf{L}_{\text{norm}} = \mathbf{I} - \mathbf{D}^{-1/2}\mathbf{A}\mathbf{D}^{-1/2}$, can be raised to powers to capture higher-scale information. Earlier work exploring multi-scale learning via powers of these matrices includes LanczosNet (Liao et al.) and Truncated Krylov Network (Luan et al.).

Most of the above models are designed for undirected graphs, and recent findings show that ignoring edge directionality can substantially degrade performance on heterophilic graphs (Rossi et al., 2024).

### 2.2    A REVIEW OF GNNS FOR DIRECTED GRAPHS

This section reviews GNN architectures specifically designed for directed graphs. For completeness, architectures targeting undirected graphs are covered in Appendix D.1.1. While traditional GNNs

effectively process undirected graphs, many real-world applications involve directed graphs where edge directionality encodes crucial semantic information. Recent research has produced three main approaches to extending GNNs to directed graphs:

1. **Real Symmetric Laplacians**: These methods convert directed graphs into symmetric representations. MotifNet (Monti et al., 2018) constructs a symmetric adjacency matrix based on motifs, but this approach is constrained by the need for predefined templates and struggles with complex structures. DGCN (Ma et al., 2019), SymDiGCN (Tong et al., 2020b), and DiGCN (Tong et al., 2020a) address this by incorporating the asymmetric adjacency matrix with its transpose through Markov processes. However, these methods are computationally expensive (Kollias et al., 2022) and struggle to scale beyond medium-sized graphs with thousands of nodes.

2. **Hermitian Laplacians**: These methods utilize complex-valued entries in Hermitian matrices to encode directional information while retaining positive semi-definite eigenvalues. MagNet (Zhang et al., 2021) pioneered this idea using complex Hermitian matrices. The more recent model HoloNet (Koke & Cremers, 2024) achieves SOTA performance on large graphs.

3. **Bidirectional Spatial Methods**: FSGNN (Maurya et al., 2021) performs feature selection over different graph signals, including $X$, $AX$, and $(A + I)X$, but it does not explicitly model edge directionality, limiting its effectiveness. More recent methods, such as Dir-GNN (Rossi et al., 2024), explicitly separate in-neighbors and out-neighbors to better capture directional information, achieving improved performance, especially on heterophilic graphs. However, Dir-GNN does not incorporate self-loops, which reduces its effectiveness on homophilic graphs. Similar ideas are also investigated in (Zhuo & Tan, 2024).

Despite these advances, fundamental limitations persist. While prior work (Rossi et al., 2024) notes the weaker performance of bidirectional models on homophilic graphs, we further identify critical issues:

- **Digraph Inception Models:** These approaches require computing edge weights through eigenvalue decomposition of the adjacency matrix to differentiate in- and out-neighbor contributions, as detailed in Appendix E.1. This results in substantial computational overhead with limited empirical gain. We find that simple uniform edge weights achieve comparable or better performance.

- **Hermitian Laplacian Models:** Despite their mathematical elegance, we prove that MagNet (Zhang et al., 2021), with code error, are functionally equivalent to GraphSAGE (Hamilton et al., 2017) with incidence-normalized (Kipf & Welling, 2016) adjacency matrices, offering no fundamental advantages over simpler approaches (see Appendix E.2). Furthermore, HoloNet's ComplexFaberConv module (Koke & Cremers, 2024) is also based on flawed implementation (see the issues reported at `https://github.com/ChristianKoke/HoloNets/issues`), and the strong empirical performance of FaberConv (Koke & Cremers, 2024) primarily arises from capturing multi-scale features (see Appendix E.3).

### 2.3 GRAPH TRANSFORMERS

Graph Transformers (GTs) often assume all nodes are connected, which can lead to a loss of graph-structural information. Learning full attention introduces quadratic computational complexity, yet empirical work (Luo et al., 2024a) shows that classic GNNs match or exceed GT performance on node-classification tasks. In our experiments, randomly assigning edge weights over a wide range performs comparably (Appendix E.1), suggesting that the existence of an edge is often more influential than the exact weight. While GTs have merits in exploring additional connections, their focus on learning precise edge weights introduces unnecessary complexity with limited practical benefit. A more detailed review of Graph Transformers is provided in Appendix D.3.

For a more detailed discussion of related work for directed graphs, please refer to Appendix E.

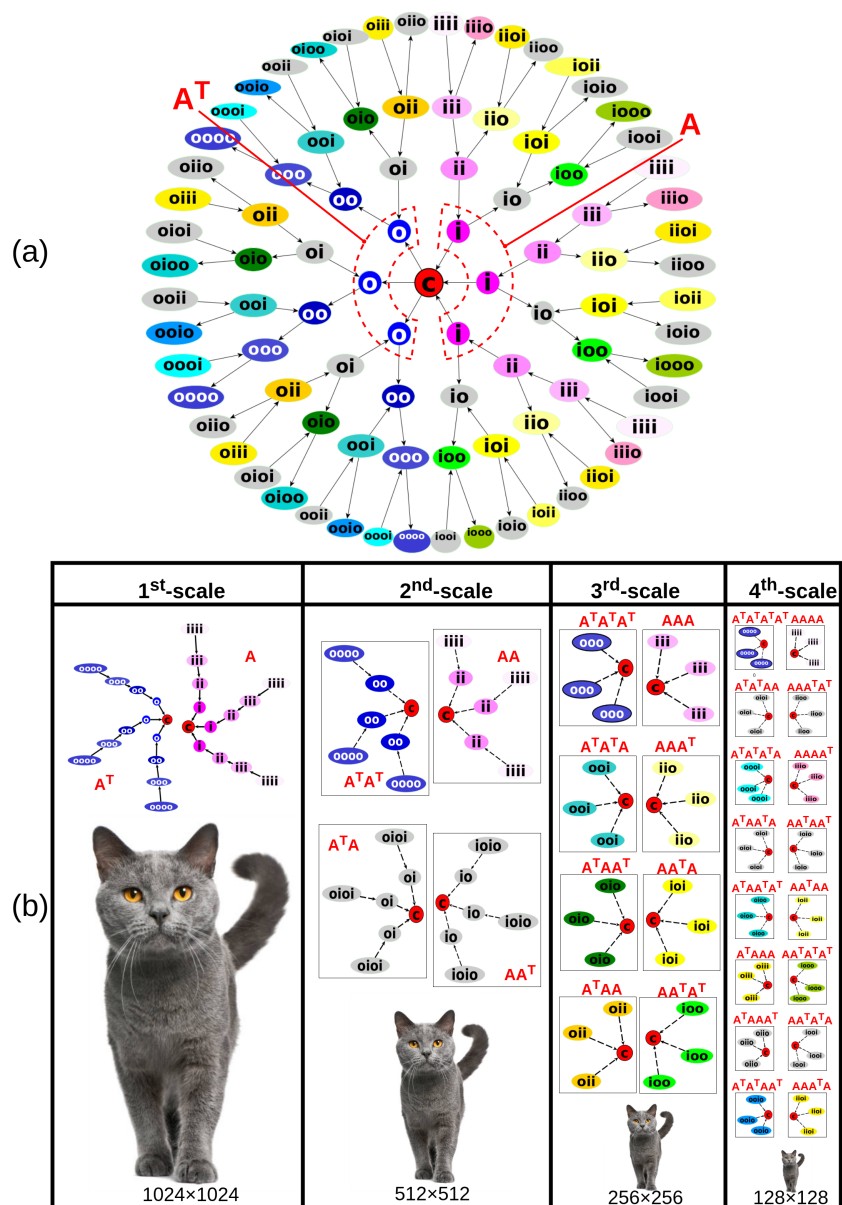

Figure 1: Illustration of ego-graphs. (a) shows the original $4$-depth ego-graph $\mathcal{G}_c^4$ of the central node $c$, illustrating all types of 4-hop neighborhoods. The 1-hop neighborhood can be in-neighbours (labeled as $i$) and out-neighbours (labeled as $o$). A 1-layer GNN aggregates their information via $A$ and $A^\top$, respectively. (b) shows scaled ego-graphs for various scaled-edge types with their corresponding adjacency matrices. For example, $i$ uses $A$ ($ii$ uses $AA$, etc.); $io$ uses $AA^\top$; and $iio$ uses $AAA^\top$. Only $1^{\text{st}}$-scale edges are actual edges of the original graph, while $k^{\text{th}}$-scale edges (for $k > 1$) are virtual lines and represent augmented connections.
*Image of cat © iStock.com/GlobalP*

## 3 SCALE INVARIANCE OF GNNS

### 3.1 PRELIMINARY

Let $\mathcal{G} = (\mathbb{V}, \mathcal{E})$ be a directed graph with $n = |\mathbb{V}|$ nodes and $m = |\mathcal{E}|$ edges, where $\mathbb{V}$ is the set of nodes and $\mathcal{E} \subseteq \mathbb{V} \times \mathbb{V}$ is the set of directed edges. Node features are stored in an $n \times f$ matrix $X$, where $f$ is the dimension of features and the node labels are $y_i \in \{1, \dots, c\}$ for $i \in \{1, \dots, n\}$. The

adjacency matrix $\boldsymbol{A} \in \{0,1\}^{n \times n}$ encodes the edges: $A_{i,j} = 1$ represents the existence of an edge from node $i$ to node $j$, and 0 the non-existence of such an edge. We focus on node classification on directed graphs; an undirected graph can be seen as a special case where each edge is accompanied by its reverse.

**Definition 3.1** (In-neighbour and in-edge $\leftarrow$). An *in-neighbour* of a node $\boldsymbol{v} \in \mathbb{V}$ is a node $\boldsymbol{u} \in \mathbb{V}$ such that there exists a directed edge from $\boldsymbol{u}$ to $\boldsymbol{v}$, i.e., $(\boldsymbol{u}, \boldsymbol{v}) \in \mathcal{E}$. The corresponding *in-edge* is the edge $(\boldsymbol{u}, \boldsymbol{v})$.

**Definition 3.2** (Out-neighbour and out-edge $\rightarrow$). An *out-neighbour* of a node $\boldsymbol{v} \in \mathbb{V}$ is a node $\boldsymbol{u} \in \mathbb{V}$ such that there exists a directed edge from $\boldsymbol{v}$ to $\boldsymbol{u}$, i.e., $(\boldsymbol{v}, \boldsymbol{u}) \in \mathcal{E}$. The corresponding *out-edge* is the edge $(\boldsymbol{v}, \boldsymbol{u})$.

As shown in Figure 1(a), in the case of a central node $\boldsymbol{c}$, nodes labeled $\boldsymbol{i}$ are in-neighbours and those labeled $\boldsymbol{o}$ are out-neighbours.

**Definition 3.3** ($k^{\text{th}}$-scaled edge). A $k^{th}$-*scaled edge*, for $k \geq 1$, denoted as $e_k(\boldsymbol{v}_i, \boldsymbol{v}_j)$, corresponds to an ordered path $(\boldsymbol{v}_i, \boldsymbol{v}_{i+1}), (\boldsymbol{v}_{i+1}, \boldsymbol{v}_{i+2}),..., (\boldsymbol{v}_{j-1}, \boldsymbol{v}_j)$ of $k$ directed edges, where each individual edge can be oriented in either direction. The $k^{\text{th}}$-scaled edge itself is regarded as a pairwise relation from $\boldsymbol{v}_i$ (the start node) to $\boldsymbol{v}_j$ (the end node), summarizing the multi-step path as a direct connection between these two nodes. Examples of scaled-edges are shown in Table 1.

Table 1: Examples of scaled edges. Dashed arrows ($--\rightarrow$) indicate augmented connections that do not exist in the original graph.

| Original Path | Scaled Edge Notation | Visual | Path Sequence | Adjacency Matrix |
|---|---|---|---|---|
| $\boldsymbol{v}_1 \leftarrow \boldsymbol{v}_2$ | $e_1(\boldsymbol{v}_2, \boldsymbol{v}_1)$ | $\boldsymbol{v}_1 \leftarrow \boldsymbol{v}_2$ | in | $\boldsymbol{A}$ |
| $\boldsymbol{v}_1 \rightarrow \boldsymbol{v}_2$ | $e_1(\boldsymbol{v}_1, \boldsymbol{v}_2)$ | $\boldsymbol{v}_1 \rightarrow \boldsymbol{v}_2$ | out | $\boldsymbol{A}^\top$ |
| $\boldsymbol{v}_1 \leftarrow \boldsymbol{v}_2 \leftarrow \boldsymbol{v}_3$ | $e_2(\boldsymbol{v}_3, \boldsymbol{v}_1)$ | $\boldsymbol{v}_1 --\rightarrow \boldsymbol{v}_3$ | in-in | $\boldsymbol{A}\boldsymbol{A}$ |
| $\boldsymbol{v}_1 \leftarrow \boldsymbol{v}_2 \rightarrow \boldsymbol{v}_3$ | $e_2(\boldsymbol{v}_2, \boldsymbol{v}_3)$ | $\boldsymbol{v}_1 --\rightarrow \boldsymbol{v}_3$ | in-out | $\boldsymbol{A}\boldsymbol{A}^\top$ |
| $\boldsymbol{v}_1 \rightarrow \boldsymbol{v}_2 \leftarrow \boldsymbol{v}_3$ | $e_2(\boldsymbol{v}_1, \boldsymbol{v}_3)$ | $\boldsymbol{v}_1 --\rightarrow \boldsymbol{v}_3$ | out-in | $\boldsymbol{A}^\top\boldsymbol{A}$ |
| $\boldsymbol{v}_1 \rightarrow \boldsymbol{v}_2 \rightarrow \boldsymbol{v}_3$ | $e_2(\boldsymbol{v}_1, \boldsymbol{v}_3)$ | $\boldsymbol{v}_1 --\rightarrow \boldsymbol{v}_3$ | out-out | $\boldsymbol{A}^\top\boldsymbol{A}^\top$ |
| $\boldsymbol{v}_1 \rightarrow \boldsymbol{v}_2 \leftarrow \boldsymbol{v}_3 \leftarrow \boldsymbol{v}_4$ | $e_3(\boldsymbol{v}_1, \boldsymbol{v}_4)$ | $\boldsymbol{v}_1 --\rightarrow \boldsymbol{v}_4$ | out-in-in | $\boldsymbol{A}^\top\boldsymbol{A}\boldsymbol{A}$ |

Consider a GNN model that learns from a graph $\mathcal{G}$ using its adjacency matrix $\boldsymbol{A}$ by aggregating information from its in-neighbours (see Appendix D.2 for details about aggregation direction). To instead learn from the out-neighbours, the model should aggregate information from the transpose of the adjacency matrix, i.e., $\boldsymbol{A}^\top$ (Rossi et al., 2024).

To encode scaled-edges, scaled adjacency matrices are formed by sequential multiplication of $\boldsymbol{A}$ and $\boldsymbol{A}^\top$:

$$\mathcal{A}_k = \{\prod_{i=1}^{k} \boldsymbol{M}_i \,|\, \boldsymbol{M}_i \in \{\boldsymbol{A}, \boldsymbol{A}^\top\}\}.$$

For example, to capture $2^{\text{nd}}$-scale neighbours, the model uses matrices from the set $\mathcal{A}_2 = \{\boldsymbol{A}\boldsymbol{A}, \boldsymbol{A}\boldsymbol{A}^\top, \boldsymbol{A}^\top\boldsymbol{A}^\top, \boldsymbol{A}^\top\boldsymbol{A}\}$ as scaled adjacency matrices. These correspond to four types of scaled 2-hop neighbours relative to the central node: $\boldsymbol{i}\boldsymbol{i}$, $\boldsymbol{i}\boldsymbol{o}$, $\boldsymbol{o}\boldsymbol{i}$, and $\boldsymbol{o}\boldsymbol{o}$ — representing in-in, in-out, out-in, and out-out edge sequences respectively (see Figure 1(b)). The correspondence between these neighbour types and their adjacency matrices is detailed in Table 1.

### 3.2 SCALED EGO-GRAPHS

An $\alpha$-depth ego-graph (Alvarez-Gonzalez et al., 2023; Sandfelder et al., 2021) of a central node $\boldsymbol{v}$, denoted by $\mathcal{G}_{\boldsymbol{v}}^{\alpha} = (\mathbb{V}_{\boldsymbol{v}}^{\alpha}, \mathcal{E}_{\boldsymbol{v}}^{\alpha})$, is a subgraph of $\mathcal{G} = (\mathbb{V}, \mathcal{E})$ where $\mathbb{V}_{\boldsymbol{v}}^{\alpha}$ contains $\boldsymbol{v}$ and all nodes reachable from $\boldsymbol{v}$ within $\alpha$ hops, and $\mathcal{E}_{\boldsymbol{v}}^{\alpha}$ contains all edges between nodes in $\mathbb{V}_{\boldsymbol{v}}^{\alpha}$, without considering edge direction. This is visually depicted in Figure 1(a). While the original ego-graph definition applies to undirected, single-edge neighborhoods, we generalize the concept for directed graphs—and further, for multi-step relations—using $k^{\text{th}}$-scaled edges to define **scaled ego-graphs**.

**Definition 3.4.** An $\alpha$-**depth** $k^{\text{th}}$-**scaled ego-graph** of a central node $\boldsymbol{v} \in \mathbb{V}$, denoted by $\mathcal{G}_{\boldsymbol{v}}^{\alpha, e_k} = (\mathbb{V}_{\boldsymbol{v}}^{\alpha, e_k}, \mathcal{E}_{\boldsymbol{v}}^{\alpha, e_k})$, where $e_k$ is a $k^{\text{th}}$-scaled edge, is defined as follows:

- $\mathbb{V}_{\boldsymbol{v}}^{\alpha,e_k}$ contains $\boldsymbol{v}$ and all nodes reachable from $\boldsymbol{v}$ within $\alpha$ hops via $e_k$.

- $\mathcal{E}_{\boldsymbol{v}}^{\alpha,e_k}$ contains all such $k^{\text{th}}$-scaled edges connecting pairs of nodes within $\mathbb{V}_{\boldsymbol{v}}^{\alpha,e_k}$.

For $k = 1$, this recovers directed generalizations of the standard ego-graph: the in-neighbour ego-graph $\mathcal{E}_{\boldsymbol{v}}^{\alpha,\leftarrow}$ and the out-neighbours ego-graph $\mathcal{E}_{\boldsymbol{v}}^{\alpha,\rightarrow}$. The different scaled ego-graphs are shown in Figure 1(b).

### 3.3 Scale Invariance of Graph Neural Networks

In node classification tasks, although the objective is to assign a label to the node $\boldsymbol{v}$, the actual input to be classified is the ego-graph $\mathcal{G}_{\boldsymbol{v}}^{\alpha,e_k}$, centered at that node $\boldsymbol{v}$, corresponding to its target label $y_{\boldsymbol{v}}$. Analogously, in image classification, the objective is to classify the entire image, which can be viewed as an image centered at a particular pixel. Just as an image can be zoomed out to produce lower-resolution views centered around a pixel, an ego-graph can be sampled at various scales centered on its central node. These scaled ego-graphs abstract progressively coarser structural information from the original, full ego-graph.

Figure 1(a) illustrates the original ego-graph, encapsulating the complete local structure around node $\boldsymbol{v}$. The $k$-scaled ego-graphs in Figure 1(b) provide samplings of this structure. Increasing the scale from a $k$-scaled to a $(k+1)$-scaled ego-graph results in progressively coarser approximations—this is analogous to viewing an image from a higher resolution $(2m \times 2m)$ to a lower resolution $(m \times m)$. In image classification, leveraging multiple scales allows convolutional neural networks (CNNs) to combine fine textures from high-resolution images with overall shape and contour from low-resolution images, improving classification accuracy.

Similarly, in graph node classification, different scales of ego-graphs capture complementary structural features: higher-scaled ego-graphs emphasize long-range neighborhood connectivity by including distant nodes, while lower-scaled ego-graphs focus on dense, immediate neighborhoods capturing richer local interactions and community structure. For example, consider a 2-layer GCN (Kipf & Welling, 2016) operating on a graph with original edges. Aggregating using $1^{\text{st}}$-scaled ego-graphs allows the model to incorporate features from nodes reachable by 1 and 2 hops of original edges. In contrast, using $k^{\text{th}}$-scaled ego-graphs, the model aggregates information from nodes located at distances of $k$ and $2k$ hops away in the original graph. Combining these multi-scale representations enables graph neural networks to exploit both detailed local features and expansive long-range structural information, enhancing node classification performance.

**Definition 3.5** (Scale Invariance for GNN). Consider a node classification task on a graph $\mathcal{G}$ with an $l$-layer GNN, where $l$ can be any positive integer. The task exhibits **scale invariance** if the classification of a node $\boldsymbol{v}$ remains invariant in different scales of its ego-graphs. Formally, there exist two distinct scales $k_1 \geq 1$ and $k_2 \geq 1$ ($k_1 \neq k_2$), such that the classification function $f$ (an $l$-layer GNN) produces the same discrete label for the original ego-graph $\mathcal{G}_{\boldsymbol{v}}^{\alpha}$ as it does for ego-graphs explicitly evaluated at those specific scales:

$$f(\mathcal{G}_{\boldsymbol{v}}^{\alpha}) = f(\mathcal{G}_{\boldsymbol{v}}^{\alpha,e_{k_1}}), f(\mathcal{G}_{\boldsymbol{v}}^{\alpha}) = f(\mathcal{G}_{\boldsymbol{v}}^{\alpha,e_{k_2}}),$$

GNNs on graphs may exhibit the property of scale invariance, which implies that multiple scales $k \geq 1$ may exist for which the information necessary for node classification is preserved in the ego-graph. A theoretical proof of scale invariance is provided in Appendix C, while empirical validation is presented in Appendix C.6. Notably, MLPs outperform GNNs on certain datasets (Zheng et al., 2021); thus, we incorporate self-features as the special case where $k = 0$.

## 4 Scale-aware Message Passing Neural Networks

### 4.1 ScaleNet

To leverage the scale invariance in node classification, we introduce **ScaleNet**, a unified and adaptable framework that integrates information from $k^{\text{th}}$-scaled ego-graphs for multiple values of $k$. As depicted in Figure 2, ScaleNet synthesizes rich node representations by flexibly combining features extracted from different directional patterns at scales k=1 and k=2: $\boldsymbol{A}$, $\boldsymbol{A}^{\top}$, $\boldsymbol{AA}$, $\boldsymbol{AA}^{\top}$, $\boldsymbol{A}^{\top}\boldsymbol{A}$,

Table 2: Node classification Accuracy (%). The best results are in **bold** and the second best are underlined. LargeScaleNet and ScaleNet use identical hyperparameters to experimentally demonstrate their equivalence. 10-fold cross validation is performed using the original train/validation/test splits, except WikiCS, which has 20 original splits. OOM indicates out of memory on GPU3090 with 24GB of VRAM. Statistical significance of the comparisons is verified using the Wilcoxon signed-rank test; for details, please refer to Appendix B.5.

| Type | Method | Homophilic Graph | | | | Heterophilic | |
|---|---|---|---|---|---|---|---|
| | | Telegram | Cora-ML | CiteSeer | WikiCS | Chameleon | Squirrel |
| Base models | MLP | 32.8±5.4 | 67.3±2.3 | 54.5±2.3 | 73.4±0.6 | 40.3±5.8 | 28.7±4.0 |
| | GCN | 86.0±4.5 | 81.2±1.4 | 65.8±2.3 | 78.8±0.4 | 64.8±2.2 | 46.3±1.9 |
| | APPNP | 67.3±3.0 | 81.8±1.3 | 65.9±1.6 | 77.6±0.6 | 38.7±2.4 | 27.0±1.5 |
| | ChebNet | 83.0±3.8 | 80.5±1.6 | 66.5±1.8 | 76.9±0.9 | 58.3±2.4 | 38.5±1.4 |
| | SAGE | 74.0±7.0 | 81.7±1.2 | 66.7±1.7 | 79.3±0.4 | 63.4±3.0 | 44.6±1.3 |
| Hermitian | MagNet | 87.6±2.9 | 79.7±2.3 | 66.5±2.0 | 74.7±0.6 | 58.2±2.9 | 39.0±1.9 |
| | SigMaNet | 86.9±6.2 | 71.7±3.3 | 44.9±3.1 | 71.4±0.7 | 64.1±1.6 | OOM |
| | QuaNet | 85.6±6.0 | 26.3±3.5 | 30.2±3.0 | 55.2±1.9 | 38.8±2.9 | OOM |
| Symmetric | Sym | 87.2±3.7 | 81.9±1.6 | 65.8±2.3 | OOM | 57.8±3.0 | 38.1±1.4 |
| | DiG | 82.0±3.1 | 78.4±0.9 | 63.8±2.0 | 77.1±1.0 | 50.4±2.1 | 39.2±1.8 |
| | DiGib | 64.1±7.0 | 77.5±1.9 | 60.3±1.5 | 78.3±0.7 | 52.2±3.7 | 37.7±1.5 |
| Symmetric (**Ours**) | **1**ym | 84.0±3.9 | 80.8±1.6 | 64.9±2.5 | 75.4±0.4 | 54.9±2.7 | 35.5±1.1 |
| | **1**iG | 95.8±3.5 | 82.0±1.3 | 65.5±2.4 | 77.4±0.6 | 70.2±1.6 | 50.7±5.8 |
| | **1**iGi2 | 93.0±5.1 | 81.7±1.3 | 67.9±2.2 | 79.2±0.5 | 58.4±2.5 | 42.7±2.5 |
| | **1**iGu2 | 92.6±4.9 | 82.1±1.2 | 67.6±1.8 | 75.6±0.9 | 60.4±2.4 | 40.4±1.8 |
| Graph Transformer | GraphGPS | 84.4±6.1 | 67.0±4.7 | 63.0±2.0 | 75.5±0.5 | 56.2±3.8 | 56.9±3.0 |
| | SGFormer | 36.6±7.5 | 65.6±1.8 | 54.8±2.2 | 73.7±0.5 | 51.4±1.7 | 35.5±1.7 |
| | Polynormer | 76.6±6.9 | 77.6±1.9 | 63.6±3.3 | 77.8±0.6 | 60.0±1.8 | 38.5±1.4 |
| BiDirection | Dir-GNN | 90.2±4.8 | 79.2±2.1 | 61.6±2.6 | 77.2±0.8 | 79.7±1.3 | 75.6±1.9 |
| **Ours** | ScaleNet | **97.2±2.1** | **82.3±1.1** | **69.1±1.2** | **79.6±0.7** | **80.1±1.5** | **76.0±2.0** |
| | $loop\_\alpha, \beta, \gamma$ | 1_0.5,-1,-1 | 1_2,-1,-1 | 1_0.5,2,-1 | 1_0.5,0.5,-1 | 0_1,1,1 | 0_1,1,1 |
| **Ours** | LargeScaleNet | 96.8±2.7 | 81.0±1.9 | 67.9±1.7 | 79.0±0.5 | 79.9±1.6 | 75.9±2.0 |

$A^\top A^\top$. To ensure optimal performance across diverse graph datasets, ScaleNet further adapts its architecture by selectively incorporating components such as self-loops and batch normalization. Each of these architectural components can be enabled or disabled based on the specific characteristics of the dataset, allowing ScaleNet to maximize its adaptability and achieve state-of-the-art performance across homophilic and heterophilic graphs.

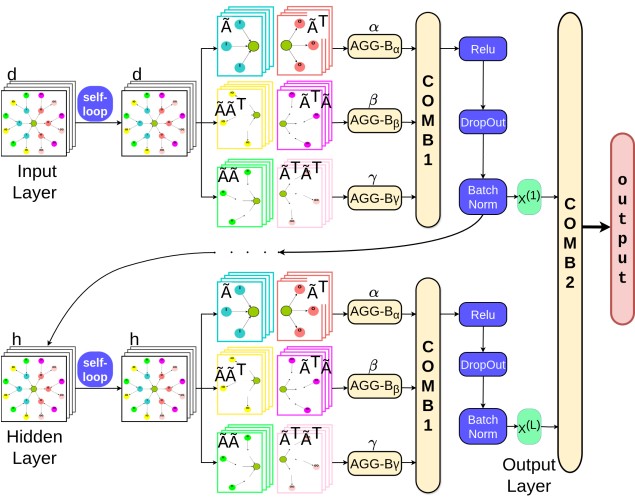

Figure 2: Schematic depiction of multi-layer ($L$-layer) ScaleNet with $d$ input channels and $h$ hidden channels. For layer-wise aggregation, the original graph is derived into two $1^{st}$-scaled and four $2^{nd}$-scaled graphs. Three **AGG-B** blocks determine input selection for **COMB1**, which uses either a jumping knowledge architecture (Xu et al., 2018) or addition. **COMB2** represents the fusion of all layers' outputs. Note that the blue blocks are optional, including self-loop operations, dropout, and layer normalization.

### 4.1.1 Layer-wise Aggregation of ScaleNet

**Bidirectional and Multi-Scale Aggregation**  Following Dir-GNN (Rossi et al., 2024), we adopt a bidirectional aggregation mechanism at each scale by defining the function $\mathbf{AGG\text{-}B}_\alpha(\boldsymbol{M}, \boldsymbol{N}, \boldsymbol{X})$ as follows:

$$(1 + \alpha)\alpha\,\mathbf{AGG}(\boldsymbol{M}, \boldsymbol{X}) + (1 + \alpha)(1 - \alpha)\,\mathbf{AGG}(\boldsymbol{N}, \boldsymbol{X}) \tag{1}$$

where $\mathbf{AGG}$ denotes a generic message-passing neural network (MPNN) operator, such as GCN (Kipf & Welling, 2016), GAT (Veličković et al., 2018), or SAGE (Hamilton et al., 2017). Here, $\boldsymbol{M}$ and $\boldsymbol{N}$ correspond to adjacency matrices encoding opposite edge directions. The hyperparameter adjusts their relative contribution: value 0 uses only the second matrix, value 1 uses only the first matrix, value 0.5 balances both equally, and value -1 excludes both. As shown in Figure 2, we apply this mechanism across multiple scales with different hyperparameters: $\alpha$ for $\boldsymbol{A}$ and $\boldsymbol{A}^\top$, $\beta$ for $\boldsymbol{A}\boldsymbol{A}^\top$ and $\boldsymbol{A}^\top\boldsymbol{A}$, and $\gamma$ for $\boldsymbol{A}\boldsymbol{A}$ and $\boldsymbol{A}^\top\boldsymbol{A}^\top$. In addition, we incorporate the self-features $\boldsymbol{X}\boldsymbol{W}$ along with the aggregated features. We combine outputs as:

$$\boldsymbol{X}^{(l)} = \mathbf{COMB1}(\mathbf{AGG\text{-}B}_\alpha^{(l)}, \mathbf{AGG\text{-}B}_\beta^{(l)}, \mathbf{AGG\text{-}B}_\gamma^{(l)}, \boldsymbol{X}^{(l-1)}\boldsymbol{W}),$$

where $\boldsymbol{X}^{(l)}$ represents updated features after $l$ layers. The function $\mathbf{COMB1}$ can be realized by the Jumping Knowledge (JK) framework (Xu et al., 2018) or element-wise addition. Further details are provided in Appendix B.1.

### 4.1.2 Multi-Layer ScaleNet

After each layer, nodes update their representations by aggregating information from multi-scale (1 to $k$) neighbors. Stacking such layers allows each node to iteratively incorporate features from increasingly distant neighborhoods, reaching nodes at distances from $l$ up to $l \times k$ in the original graph through repeated update steps.

We define the multi-layer ScaleNet as follows:

$$\boldsymbol{Z} = \mathbf{COMB2}(\boldsymbol{X}^{(1)}, \boldsymbol{X}^{(2)}, \ldots, \boldsymbol{X}^{(l)}), \tag{2}$$

where $\boldsymbol{X}^{(i)}$ is the output from the $i$-th layer. The function $\mathbf{COMB2}$ combines outputs across layers, using techniques such as Jumping Knowledge (JK) (Xu et al., 2018) or simply the final layer's output. This design enables ScaleNet to integrate multi-scale, multi-hop information for more expressive node representations. ScaleNet achieves the best performance across all six datasets, as shown in Table 2. We evaluate on four homophilic and two heterophilic directed graphs. More details about datasets and experiments are reported in Appendix B.4, and further discussion of its performance is provided in Appendix B.2.

### 4.2 LargeScaleNet

ScaleNet improves scalability over DiG(ib) (Tong et al., 2020a) by eliminating eigenvalue decomposition. However, it still relies on computing scaled adjacency matrices like $\boldsymbol{A}\boldsymbol{A}$, which can be memory-intensive and computationally expensive for large graphs. As shown in Table E.5, to aggregate features along scaled edges such as $\leftarrow\leftarrow$, ScaleNet first computes the scaled adjacency matrix $\boldsymbol{A}\boldsymbol{A}$, then normalizes it to obtain $\widetilde{\boldsymbol{A}\boldsymbol{A}}$, and finally applies it to the node feature matrix $\boldsymbol{X}$ as $\widetilde{\boldsymbol{A}\boldsymbol{A}}\boldsymbol{X}$. This approach requires storing and multiplying two $n \times n$ adjacency matrices, which becomes prohibitive for large graphs.

Inspired by FaberNet (Koke & Cremers, 2024), LargeScaleNet avoids explicit computation of scaled adjacency matrices by changing the multiplication order. Instead of computing $\widetilde{\boldsymbol{A}\boldsymbol{A}}\boldsymbol{X}$, it computes $\tilde{\boldsymbol{A}}(\tilde{\boldsymbol{A}}\boldsymbol{X})$. While these operations are not numerically identical, they preserve the same sparsity patterns and connectivity structures. Thus, the node embedding changes from $\sigma(\widetilde{\boldsymbol{A}\boldsymbol{A}}\boldsymbol{X}\boldsymbol{W})$ to $\sigma(\tilde{\boldsymbol{A}}(\tilde{\boldsymbol{A}}\boldsymbol{X})\boldsymbol{W})$. By the Universal Approximation Theorem(UAT), a sufficiently expressive linear transformation $\boldsymbol{W}$ could compensate for the numerical differences introduced by multiplication re-ordering, making the two approaches functionally equivalent for feature aggregation. For more about UAT, see Appendix C.5. Similar arguments appear in (Sun et al.) and (Wu et al., 2019).

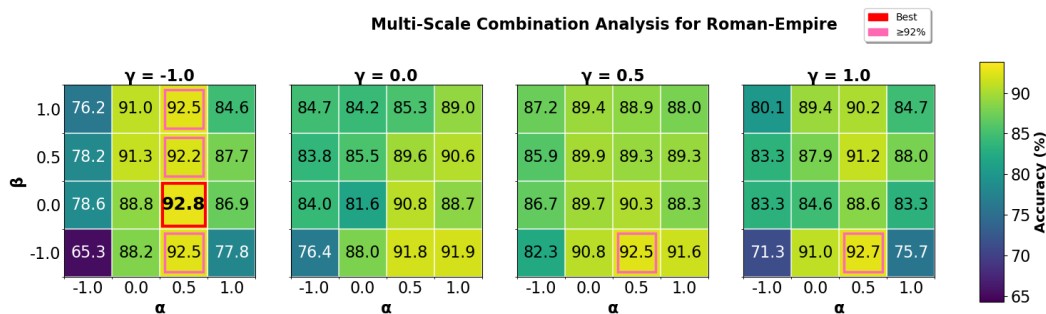

Figure 3: Ablation study showing accuracy (%) with different combinations of multiple scales on the Roman-Empire dataset. Red boxes highlight the best performing configuration, while pink boxes indicate high-performance configurations ($\geq$92.0%).

Consider the Snap-patent dataset with 2.9 million nodes, resulting in an adjacency matrix $A$ of size 2.9M $\times$ 2.9M. Table E.5 (in Appendix) compares the matrix multiplication dimensions between ScaleNet and LargeScaleNet, showing that LargeScaleNet avoids expensive computations and achieves improved scalability. Using an NVIDIA A40 GPU, ScaleNet cannot handle large graphs such as Arxiv-year (169k nodes), whereas LargeScaleNet successfully processes datasets like Snap-patent with 2.9 million nodes. More detail in Appendix E.3.

Table 3: Comparison of ScaleNet and LargeScaleNet in aggregating features along $2^{\text{nd}}$-scale edges. Here, $\tilde{A}$ is the normalized adjacency matrix, and $\widetilde{AA}$ denotes the normalized form of $AA$.

| Edge Type | $\leftarrow\leftarrow$ | $\leftarrow\rightarrow$ | $\rightarrow\leftarrow$ | $\rightarrow\rightarrow$ |
|---|---|---|---|---|
| **ScaleNet** | $\widetilde{(AA)}X$ | $\widetilde{(AA^\top)}X$ | $\widetilde{(A^\top A)}X$ | $\widetilde{(A^\top A^\top)}X$ |
| **LargeScaleNet** | $\tilde{A}(\tilde{A}X)$ | $\tilde{A}^\top(\tilde{A}X)$ | $\tilde{A}(\tilde{A}^\top X)$ | $\tilde{A}^\top(\tilde{A}^\top X)$ |

### 4.2.1 PERFORMANCE OF LARGESCALENET

We first validate the equivalence between LargeScaleNet and ScaleNet on small to medium datasets. Table 2 shows that with identical hyperparameters, both methods achieve very similar accuracy across all 6 datasets, empirically confirming their theoretical equivalence. Notably, ScaleNet consistently achieves slightly better performance, demonstrating that while the methods are theoretically equivalent, ScaleNet's optimized design provides practical benefits on smaller datasets.

For large graph datasets, we evaluate LargeScaleNet on three popular heterophilic benchmarks, with results shown in Table 4. Building on Dir-GNN (Rossi et al., 2024), both LargeScaleNet and FaberNet (Koke & Cremers, 2024) demonstrate improved performance by incorporating higher-scale features. Table E.6 (in Appendix) shows the construction of different scales for both FaberNet and LargeScaleNet. While FaberNet applies weight penalties to higher-scale features (e.g., computing $\alpha AX + (1-\alpha)A^\top X$ with exponential decay), LargeScaleNet achieves competitive results without penalties by including all four directional types at the $2^{\text{nd}}$-scale: $AA$, $AA^\top$, $A^\top A$, $A^\top A^\top$. This captures more comprehensive scaled-edge interactions while maintaining computational efficiency. Wilcoxon testing for the top two models (see Appendix E.3) confirms that LargeScaleNet significantly outperforms FaberNet across the evaluated datasets.

### 4.3 ABLATION STUDY

We validate our key design choices through ablation studies, with additional results in Appendix C.7 and Appendix C.8.

- Scale-aware Selection. Figure 3 and Figure C.2 (in Appendix C.8) show grid search results for different combinations of $1^{\text{st}}$- and $2^{\text{nd}}$-hop directed scales. Multi-scale combinations consistently outperform single-scale baselines, confirming that different directed

Table 4: Node classification Accuracy (%).Results on large real-world directed graph datasets. `OOM` indicates out-of-memory errors. The best performance is shown in **bold**, and the second-best is underlined. $k$ denotes the scale, $\alpha, \beta, \gamma$ are to select directed scale representations.

|  | Arxiv-year | Snap-patents | Roman-Empire |
|---|---|---|---|
| MLP | 36.70±0.21 | 31.34±0.05 | 64.94±0.62 |
| GCN | 46.02±0.26 | 51.02±0.06 | 73.69±0.74 |
| FSGNN | 50.47±0.21 | 65.07±0.03 | 79.92±0.56 |
| DiGCN | OOM | OOM | 52.71±0.32 |
| MagNet | 60.29±0.27 | OOM | 88.07±0.27 |
| DirGNN | 63.97±0.30 | 73.95±0.05 | 91.3±0.46 |
| FaberNet | 64.62±1.01 | 74.55±0.11 | 92.24±0.43 |
| GraphGPS | OOM | OOM | 82.72±0.68 |
| SGphormer | 34.89±0.55 | OOM | 65.07±0.51 |
| Polynormer | 52.50±0.77 | OOM | 90.32±0.36 |
| LargeScaleNet | **65.82±0.36** | **75.05±0.05** | **93.58±0.24** |
| $k\_\alpha, \beta, \gamma$ | 2_0.5,-1,-1 | 2_0.5,0.5,0.5 | 2_0.5,0.5,-1 |

scales capture complementary structural information. Optimal scale combinations vary across datasets, as shown in Table 4, demonstrating the need for adaptive scale selection. Self-Loop Integration.

- We introduce binary parameter *loop*: $loop = 1$ adds self-loops to adjacency matrix $\mathbf{A}$, while $loop = 0$ uses the original matrix. Self-loops improve performance on homophilic graphs but show limited benefit on heterophilic graphs (Kipf & Welling, 2016; Tong et al., 2020a). As shown in Appendix C.7, heterophilic graphs prefer no self-loops while homophilic graphs benefit from self-loop addition. Detailed analysis is in Appendix F.

Both components are essential, with their effectiveness varying across graph structures and requiring dataset-specific optimization. A theoretical analysis and an experimental comparison of computational memory usage and runtime are provided in Appendix G.

## 5 CONCLUSIONS

We introduce scale invariance to graph neural networks, adapting this widely-used concept from computer vision to effectively capture multi-resolution structural information in graphs. By selectively combining scaled representations that provide complementary benefits, we present scale-aware message passing neural networks that achieve state-of-the-art performance on node classification tasks across 6 medium-scale graph benchmarks and 3 large-scale graph datasets.

Our approach consists of two key components. ScaleNet leverages scaled adjacency matrices to enable intuitive understanding and manipulation of graph scale, proving particularly effective for medium-sized graphs. Building upon this foundation, we develop LargeScaleNet, which maintains competitive performance on medium-scale graphs while additionally optimizing the message passing sequence to efficiently scale multi-scale learning to graphs with millions of nodes.

Extensive experimental results demonstrate that selectively combining multiple scaled graph representations effectively captures complementary structural information across different graph resolutions. We hope that this study will establish scale invariance as a fundamental and effective principle for robust graph representation learning and open new avenues for advancing graph neural network architectures through principled multi-scale design.

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

# A  APPENDIX OUTLINE

The Appendix is organized as follows:

- Appendix B provides additional details on the model architecture and experimental settings.
- Appendix C presents the theoretical proof and empirical demonstration of scale invariance.
- Appendix D offers a more detailed and broader literature review.
- Appendix E is a case study for 3 typical models for directed graph.
- Appendix F analyzes the role of adaptive self-loops and their influence on performance.
- Appendix **??** is experimental results of runtime and memory.
- Appendix H is about the use of large language models.

# B  FURTHER DETAILS ABOUT SCALENET AND EXPERIMENTS

## B.1  FURTHER DETAILS ABOUT SCALENET

As discussed in Section 4.1, the hyperparameter $\alpha$ controls the relative contributions of aggregation via the matrices $M$ and $N$, and can take values such as 0, 1, or 0.5. Additionally, setting $\alpha = 2$ combines the adjacency matrices $M$ and $N$ directly before aggregation, while setting $\alpha = 3$ considers their intersection:

$$\textbf{AGG-B}_2(M, N, X) = \textbf{AGG}(M \cup N, X) \tag{A.1}$$

$$\textbf{AGG-B}_3(M, N, X) = \textbf{AGG}(M \cap N, X) \tag{A.2}$$

While $\alpha$ is used to select between $A$ and $A^\top$, $\beta$ determines whether to use $AA^\top$ or $A^\top A$, and $\gamma$ controls the choice between $AA$ and $A^\top A^\top$.

## B.2  PERFORMANCE OF SCALENET ON DIFFERENT GRAPHS

ScaleNet is designed to adapt to the unique characteristics of each dataset, delivering optimal performance on both homophilic and heterophilic graphs. This is achieved through customizable options such as combining directed scaled graphs, incorporating batch normalization, and adding or removing self-loops.

During hyperparameter tuning via grid search, we observed the following key findings:

- **Homophilic Graphs**: Performance improves with the addition of self-loops and the use of scaled graphs derived from opposite directed scaled edges, such as $AA$ and $A^\top A^\top$.

- **Heterophilic Graphs**: Performance benefits from removing self-loops and utilizing scaled graphs with preferred directional scaled edges, while excluding those based on the opposite directional scaled edges.

- Additional findings:

  - For imbalanced datasets such as Telegram, incorporating batch normalization significantly improves performance.

  - The CiteSeer dataset performs better with the removal of nonlinear activation functions.

Our unified model, optimized through grid search, reveals the characteristics of different graph datasets and provides a strong basis for model comparison.

Table 2 summarizes the 10-fold cross-validation results. ScaleNet consistently achieves top performance across all six datasets, significantly outperforming existing models on both homophilic and heterophilic graphs.

Model **1**ym assigns 1 to edge weights of model Sym: similarly, model **1**iG and **1**iGi2 are assigning 1 to edge weights of models DiG and DiGib, respectively. Model **1**iGu2 and **1**iGu3 assign weights of 1 to scaled edges, but use union instead of intersection in DiG**i**b, and the last number k denotes the model includes up to $k^{th}$-scale edges, while DiGib only scales up to $2^{nd}$-order. At the end of model name, "ib" would be used interchangeably with "i2". Parameters $\alpha$, $\beta$, and $\gamma$ controlling ScaleNet components: $\alpha$ controls $A$ and $A^\top$, $\beta$ controls $AA^\top$ and $A^\top A$, and $\gamma$ controls $AA$ and $A^\top A^\top$. Parameter loop is 1 when adding selfloop and 0 when not adding.

## B.3 ROBUSTNESS TO IMBALANCED GRAPHS

Table B.1: Accuracy (%) on imbalanced datasets (imbalance ratio = 100:1). When accuracy is below 45%, only one split is used.

| Type | Method | Cora-ML | CiteSeer | WikiCS |
|------|--------|---------|----------|--------|
| Standard | MagNet | 47.9±5.5 | 29.3 | 62.0±1.5 |
| | Dir-GNN | 41.1 | 25.0 | 62.9±1.4 |
| Augment | DiG | 60.9±1.8 | 36.9 | 72.2±1.4 |
| | DiGib | 55.7±2.9 | 40.4 | 69.8±1.2 |
| | **1**iG | 64.9±4.7 | 42.3 | 71.0±1.5 |
| | **1**iGi2 | 61.9±5.7 | 41.5 | 71.0±1.6 |
| | ScaleNet | 60.3±6.7 | 43.1 | 69.4±1.2 |

ScaleNet improves robustness against imbalanced graphs by leveraging multi-scale graphs, similar to data augmentation techniques.

Table B.1 indicates that ScaleNet consistently outperforms Dir-GNN and MagNet on imbalanced datasets. The imbalance ratio measures the size disparity between the largest and smallest classes. For homophilic graphs, ScaleNet's advantage stems from its use of higher-scale graphs and self-loops, which enhances its ability to capture essential features that Dir-GNN and MagNet might miss. Conversely, single-scale networks like Dir-GNN (Rossi et al., 2024) and MagNet (Zhang et al., 2021) are prone to incorporate irrelevant nodes due to excessive layer stacking when aggregating information from longer-range nodes.

## B.4 MORE DETAILS ON EXPERIMENTS

### B.4.1 DATASETS FOR SCALENET EXPERIMENT IN TABLE 2

We use six widely-adopted real-world datasets in Table 2, comprising four homophilic and two heterophilic graph datasets. To ensure consistency and comparability, we maintain the original train/validation/test splits provided by the source datasets. All datasets have 10 splits, except WikiCS, which originally includes 20 splits. Dataset statistics are reported in Table B.2.

Table B.2: Dataset statistics. Imbalance Ratio is the ratio of the largest class to the smallest class in the training sets. %No-In and %No-Out represent the percentage of nodes with no direct in-neighbors and no direct out-neighbors, respectively. %In-Homo denotes the percentage of nodes whose in-neighbors predominantly share the same label as the node, while %Out-Homo indicates the percentage of nodes whose out-neighbors predominantly share the same label.

| Dataset | #Nodes | #Edges | #Feat. | #C | Imbal-Ratio | %No-In | %In-Homo | %No-Out | %Out-Homo | Label rate |
|---------|--------|--------|--------|----|-----|--------|----------|---------|-----------|-----------|
| CiteSeer | 3312 | 4715 | 3703 | 6 | 1.0 | 30.2 | 52.7 | 41.1 | 44.1 | 3.6% |
| Cora-ML | 2995 | 8416 | 2879 | 7 | 1.0 | 41.7 | 50.2 | 11.7 | 74.7 | 4.8% |
| Telegram | 245 | 8912 | 1 | 4 | 3.0 | 16.7 | 32.2 | 25.3 | 32.7 | 60% |
| PubMed | 19717 | 44327 | 500 | 3 | 1.0 | 33.4 | 54.2 | 34.2 | 54.0 | 0.3% |
| WikiCS | 11701 | 297110 | 300 | 10 | 9.5 | 18.5 | 69.1 | 3.8 | 76.7 | 5% |
| Chameleon | 2277 | 36101 | 2325 | 5 | 1.3 | 62.1 | 10.4 | 0.0 | 25.3 | 48% |
| Squirrel | 5201 | 217073 | 2089 | 5 | 1.1 | 57.6 | 8.5 | 0.0 | 24.2 | 48% |

- **Homophilic Graph Datasets**
    - Telegram is characterized as a network comprising pairwise interactions between various elements, including channels, chats, posts, and URL links within the platform. We follow the splits described in MagNet (Zhang et al., 2021).
    - For citation networks, nodes represent documents and papers, while edges denote citation links. The classes correspond to different research domains.
        * For Citeseer and Cora-ML, we use the splits specified in the DiGCN(ib) paper (Tong et al., 2020a).

       ∗ For the WikiCS dataset, the splits are described in (Mernyei & Cangea, 2020).

- **Heterophilic Graph Datasets**

    - Chameleon and Squirrel are Wikipedia page networks focused on specific topics, with nodes representing web pages and edges representing links between them (Rozember-czki et al., 2021). Nodes have features based on important nouns from the Wikipedia pages and are classified into five categories based on their average monthly traffic. These classifications follow the splits used in GEOM-GCN (Pei et al., 2020).

### B.4.2 Datasets for 1iG vs. DiG in Table E.2

The datasets tested in Table E.2 are comprised of 8 directed graphs and 7 undirected graphs.

**Directed Graph Datasets**

- **WebKB Datasets**: WebKB is a webpage dataset collected from computer science departments of various universities by Carnegie Mellon University. We use the three subdatasets of it, Cornell, Texas, and Wisconsin, where nodes represent web pages, and edges are hyperlinks between them. Node features are the bag-of-words representation of web pages. The web pages are manually classified into the five categories, student, project, course, staff, and faculty.

- **Citation Datasets**: Citeseer, CoraML, WikiCS are the same datasets as reported in Appendix B.4.1. The PubMed dataset are provided through the Deep Graph Library (DGL). We generate 10 splits of PubMed as the same as Citeseer and CoraML.

- **Social Networks**: Telegram is a directed social network, and it is the same datasets as reported in Appendix B.4.1.

**Undirected Graph Datasets**

- **Undirected Citation Datasets**: Cora, Citeseer-U, and Pubmed-U are standard citation network benchmark datasets. In these networks, nodes represent papers, and edges denote citations of one paper by another. Node features are the bag-of-words representation of papers, and node label is the academic topic of a paper.

    10 Splits: 20 nodes per class for training, 30 nodes per class for validation, all rest for testing.

- **Coauthor Datasets**: Coauthor CS and Coauthor Physics are co-authorship graphs based on the Microsoft Academic Graph from the KDD Cup 2016 challenge. Here, nodes are authors, that are connected by an edge if they co-authored a paper; node features represent paper keywords for each author's papers, and class labels indicate most active fields of study for each author.

- **Co-purchasing Datasets**: Amazon Computers and Amazon Photo co- purchase networks (Shchur et al., 2019): The Amazon Computers and Amazon Photo network datasets are both extracted from Amazon co-purchase Net- works. The nodes represents the goods and edges represent that the two nodes(goods) connected are frequently bought together, and the product reviews are bag-of-words node features.

### B.4.3 Large dataset for ScaleBigNet

We adopt the original train/validation/test split provided in the ogbn-arxiv dataset (Hu et al., 2021), which consists of a single predefined split. As a result, no standard deviation across multiple runs is reported.

The ogbn-arxiv dataset is a large-scale directed graph representing the citation network among all Computer Science (CS) papers on arXiv. The task is to predict the subject area (one of 40 classes) for each paper. The training set consists of papers published up to 2017, the validation set includes those from 2018, and the test set comprises papers published since 2019.

The dataset contains 169,343 nodes, 1,166,243 directed edges, and 40 classes in total.

The Snap-patents dataset is sourced from Lim et al. (2021), and Roman-Empire is from Platonov et al. (2024).

**Experiments on Large Graphs**    As shown in Table 4, we run 5 random splits for Arxiv-Year and Snap-patents, following the protocol in Lim et al. (2021), and 10 random splits for Roman-Empire, following Platonov et al. (2024). The results of MLP, GCN, and FSGNN are reported directly from HoloNet (Koke & Cremers, 2024).

### B.4.4    EXPERIMENTAL SETTINGS

**Implementation Details**    All experiments in Table 2 use a GPU3090 with 24GB of memory. For the experiments on imbalanced datasets shown in Table B.1 of the main text, which were conducted on a GPU4070 with 12GB of memory.

For all experiments conducted on homophilic datasets, we employed the Adam optimizer and trained the model for 1,500 epochs, utilizing early stopping based on validation accuracy with a patience of 410 epochs. Additionally, a learning rate scheduler was applied, with a patience of 80 epochs. In contrast, for experiments involving heterophilic datasets, we followed the methodology outlined by Rossi et al. (Rossi et al., 2024), using a learning rate of 0.005, a patience of 400 epochs, and a total training duration of 100,000 epochs, with early stopping applied after 810 epochs of no improvement in validation accuracy. We report the mean and standard deviation of test accuracy over 10 runs, except for WikiCS, where the results are based on 20 splits as originally provided.

**Hyperparameter Tuning**    For each model, we perform a grid search to optimize the following hyperparameters:

- **Number of Layers:** from 1 to 5
- **Learning Rate:** 0.1, 0.01, 0.005
- **L2 Regularization**: 0, 0.0005
- **Dropout Rate:** 0.0, 0.5
- **Batch Normalization:** Enabled(1) or Disabled(0)
- **ReLU Activation:** Applied(1) or Not Applied(0)
- **Jumping Knowledge (JK) Aggregation:** `max`, `cat`, or `0` (no jumping knowledge structure)
- **Self-loop Handling:** Added (add), Removed (remove), or None (0)
- **Directional Parameter** $\alpha, \beta, \gamma$ in Equation 1 : 0, 0.5, 1, 2, 3

We consider both options for JK structures, batch normalization, and ReLU Activation, applied either within layer-wise ScaleNet or across multiple layers of ScaleNet.

For higher scaled adjacency matrices, such as $AA^T$ and $A^T A$, there are options to either remove or retain generated self-loops.

### B.5    STATISTICAL SIGNIFICANCE TEST

We conducted Wilcoxon signed-rank tests to compare the performance of the top models across seven different datasets. The results are presented in Tables B.3 to B.8. Each table displays the mean performance ± standard deviation for each model on the diagonal, with p-values and test statistics for pairwise comparisons on the off-diagonal. P-values less than 0.05 are bolded to indicate statistically significant differences.

The overall results of all datasets are shown in Table B.9.

Table B.3: Wilcoxon signed-rank test results for pairwise comparisons among the top models (ScaleNet, **1iG**, Dir-GNN, and MagNet) on the Telegram dataset. ScaleNet is identified as the superior model for this dataset. The results indicate that ScaleNet consistently outperforms the other models, demonstrating statistically significant improvements in performance.

| Dataset: Telegram | ScaleNet | 1iG | Dir-GNN | MagNet |
|---|---|---|---|---|
| ScaleNet | 95.93 ± 2.99 | **0.0064** (62.0) | **0.0001** (18.0) | **0.0000** (0.0) |
| **1**iG | **0.0064** (62.0) | 93.27 ± 4.30 | 0.3912 (131.0) | **0.0002** (35.5) |
| Dir-GNN | **0.0001** (18.0) | 0.3912 (131.0) | 92.20 ± 3.66 | **0.0000** (20.0) |
| MagNet | **0.0000** (0.0) | **0.0002** (35.5) | **0.0000** (20.0) | 86.60 ± 5.42 |

Table B.4: Wilcoxon signed-rank test results for pairwise comparisons among the models ScaleNet, **1**iGu2, **1**iG, Sym, and **1**ym on the Cora-ML dataset. The best models for the Cora-ML dataset are ScaleNet, 1iGu2, and 1iG, as they show no significant differences.

| Dataset: Cora-ML | ScaleNet | 1iGu2 | 1iG | Sym | 1ym |
|---|---|---|---|---|---|
| ScaleNet | 82.22 ±1.16 | 0.3043 (170.0) | 0.1241 (156.5) | **0.0001** (56.0) | **0.0002** (60.5) |
| 1iGu2 | 0.3043 (170.0) | 82.43 ±1.48 | 0.1642 (163.5) | **0.0001** (53.5) | **0.0001** (41.0) |
| 1iG | 0.1241 (156.5) | 0.1642 (163.5) | 81.85 ±1.27 | **0.0099** (108.5) | **0.0076** (105.0) |
| Sym | **0.0001** (56.0) | **0.0001** (53.5) | **0.0099** (108.5) | 80.75 ±1.84 | 0.6575 (197.0) |
| 1ym | **0.0002** (60.5) | **0.0001** (41.0) | **0.0076** (105.0) | 0.6575 (197.0) | 80.90 ±1.45 |

Table B.5: Wilcoxon signed-rank test results for pairwise comparisons among the top models (ScaleNet, **1**iGi2, Sym, and **1**iGu2) on the CiteSeer dataset. ScaleNet stands out as the best model for the CiteSeer dataset.

| Dataset: CiteSeer | ScaleNet | 1iGi2 | Sym | 1iGu2 |
|---|---|---|---|---|
| ScaleNet | 68.13 ±1.46 | **0.0013** (82.0) | **0.0000** (3.0) | **0.0000** (27.0) |
| **1**iGi2 | **0.0013** (82.0) | 66.98 ±2.09 | **0.0003** (65.0) | 0.1981 (169.0) |
| Sym | **0.0000** (3.0) | **0.0003** (65.0) | 65.44 ±2.43 | 0.0767 (146.0) |
| **1**iGu2 | **0.0000** (27.0) | 0.1981 (169.0) | 0.0767 (146.0) | 66.22 ±1.61 |

Table B.6: Wilcoxon signed-rank test results for pairwise comparisons among the top models (ScaleNet, SAGE, **1iGi2**, DiGib, and Dir-GNN) on the WikiCS dataset. ScaleNet and SAGE are the best models, showing a significant difference from other models while not showing a significant difference from each other.

| Dataset: WikiCS | ScaleNet | SAGE | 1iGi2 | DiGib | Dir-GNN |
|---|---|---|---|---|---|
| ScaleNet | 79.30 ± 0.51 | 0.9515 (229.0) | **0.0000** (38.5) | **0.0000** (1.0) | **0.0000** (0.0) |
| SAGE | 0.9515 (229.0) | 79.22 ± 0.50 | **0.0000** (25.0) | **0.0000** (0.0) | **0.0000** (0.0) |
| **1**iGi2 | **0.0000** (38.5) | **0.0000** (25.0) | 78.52 ± 0.66 | **0.0000** (10.0) | **0.0000** (0.0) |
| DiGib | **0.0000** (1.0) | **0.0000** (0.0) | **0.0000** (10.0) | 77.31 ± 0.73 | **0.0000** (1.0) |
| Dir-GNN | **0.0000** (0.0) | **0.0000** (0.0) | **0.0000** (0.0) | **0.0000** (1.0) | 75.67 ± 0.75 |

Table B.7: Wilcoxon signed-rank test results for pairwise comparisons among the top models ScaleNet$(1, 1, 1)$, ScaleNet$(1, -1, 1)$, ScaleNet$(1, -1, -1)$, Dir-GNN, and 1iG on the Chameleon dataset. The best model is ScaleNet with the $\alpha, \beta, \gamma$ setting of (1,1,1).

| Dataset: Chameleon | ScaleNet $(1, 1, 1)$ | ScaleNet $(1, -1, 1)$ | ScaleNet $(1, -1, -1)$ | Dir-GNN | 1iG |
|---|---|---|---|---|---|
| ScaleNet $(1, 1, 1)$ | 79.27 ±1.60 | **0.0000** (48.0) | **0.0227** (103.0) | **0.0095** (97.5) | **0.0000** (0.0) |
| ScaleNet $(1, -1, 1)$ | **0.0000** (48.0) | 78.03 ±1.66 | 0.0601 (120.5) | 0.2801 (179.5) | **0.0000** (0.0) |
| ScaleNet $(1, -1, -1)$ | **0.0227** (103.0) | 0.0601 (120.5) | 78.59 ±1.34 | 0.5922 (179.5) | **0.0000** (0.0) |
| Dir-GNN | **0.0095** (97.5) | 0.2801 (179.5) | 0.5922 (179.5) | 78.38 ±1.30 | **0.0000** (0.0) |
| 1iG | **0.0000** (0.0) | **0.0000** (0.0) | **0.0000** (0.0) | **0.0000** (0.0) | 71.05 ±1.91 |

Table B.8: Wilcoxon signed-rank test results for pairwise comparisons among the top models ScaleNet$(1, 1, 1)$, ScaleNet$(1, 1, -1)$, ScaleNet$(1, -1, -1)$, and DirGNN on the Squirrel dataset. Dir-GNN shows no significant difference from ScaleNet across the three $\alpha, \beta, \gamma$ settings. Therefore, the best models for the Squirrel dataset are ScaleNet and Dir-GNN.

| Dataset: Squirrel | ScaleNet $(1, 1, 1)$ | ScaleNet $(1, 1, -1)$ | ScaleNet $(1, -1, -1)$ | Dir-GNN |
|---|---|---|---|---|
| ScaleNet $(1, 1, 1)$ | 75.31 ±1.85 | **0.0159** (106.0) | **0.0106** (110.0) | 0.3085 (181.5) |
| ScaleNet $(1, 1, -1)$ | **0.0159** (106.0) | 74.75 ±1.84 | 0.9515 (229.0) | 0.3274 (160.0) |
| ScaleNet $(1, -1, -1)$ | **0.0106** (110.0) | 0.9515 (229.0) | 74.68 ±2.09 | 0.3387 (185.0) |
| Dir-GNN | 0.3085 (181.5) | 0.3274 (160.0) | 0.3387 (185.0) | 75.00 ±1.91 |

Table B.9: Results of the Wilcoxon signed-rank test for the top two models, based on 30 splits. Statistical significance is indicated by p-values less than 0.05, with significant p-values shown in bold. If p-value is not below 0.05, both models are considered equally effective.

| | Telegram | Cora-ML | CiteSeer | WikiCS | Chameleon | Squirrel |
|---|---|---|---|---|---|---|
| Methods | ScaleNet 1iG | ScaleNet 1iGu2 | ScaleNet 1iGi2 | ScaleNet SAGE | ScaleNet Dir-GNN | ScaleNet Dir-GNN |
| p-value (Statistic) | **0.0064** (62.0) | 0.1241 (156.5) | **0.0013** (82.0) | 0.9515 (229.0) | **0.0095** (97.5) | 0.3085 (181.5) |
| Best Model | ScaleNet | Both | ScaleNet | Both | ScaleNet | Both |

# C    PROOF OF SCALE INVARIANCE

In this section, we present a proof of scale invariance for Graph Neural Networks (GNNs), exploring the relationship between standard and scaled adjacency matrices in node classification tasks. First, we derive the output of a $k$-layer GCN using the adjacency matrix $A$. We then extend this to scaled adjacency matrices with bidirectional aggregation, demonstrating that the resulting models are equivalent to dropout versions of lower-scale, bidirectional GCNs that aggregate using both $A$ and $A^\top$. These discussions are separated into two cases: one where self-loops are added, and one where they are not. We focus on the Graph Convolutional Network (GCN) model (Kipf & Welling, 2016) as it represents the basic form of neighborhood aggregation.

Non-linear functions are integral components of all models discussed here. To streamline the exposition and focus on core mechanisms, they are omitted in following presentation.

## C.1    PRELIMINARIES

Let $X$ denote node features, $A$ denote the adjacency matrix (where an element is 1 if an edge exists and 0 otherwise). Let $D$ denote the degree matrix of $A$, and $I$ be the identity matrix. For weight matrices, $W$ is used as a generic term, with $W_0, W_1, \ldots$ representing distinct weight matrices. For a scaled edge $e_k$ (as defined in Definition 3.3), let $X_{e_k}$ represent the 1-hop neighbors of $X$ through $e_k$, for examaple, $X_i$ denote 1-hop in-neighbors of $X$. $X^k$ denotes representation of nodes after $k$-layer GNN.

**Theorem C.1.** *The layer-wise propagation of a GCN is:*

- *Without self-loops:* $\sigma(\sum X_i W)$

- *With self-loops:* $\sigma(\sum X_i W_1 + X W_0)$,

*where $\sigma$ is non-linear function.*

Non-linear functions are integral components of all models discussed here. To streamline the exposition and focus on core mechanisms, they are omitted in following presentation.

*Proof.* As outlined in Table D.1 (provided in Appendix D.1.1), $\tilde{A}$ denotes incidence-normalized A, the layer-wise propagation of a GCN (Kipf & Welling, 2016)is represented as follows:

$$\sigma(\tilde{A} X W)(\text{no self-loops}), \quad \sigma(\widetilde{(A + I)} X W)(\text{with self-loops})$$

Since incidence normalization corresponds to a component-wise multiplication with the normalization matrix $N$, we have $\tilde{A} X W = (N \odot A) X W$. What is zero in $A$ remains zero in $\tilde{A}$, and vice versa; that is, $\tilde{A}$ does not change the structure of the graph. The element-wise multiplication with $N$ only affects the scaling of features and thus influences the learning of $W$, without altering the connectivity pattern encoded by $A$. Therefore, by the Universal Approximation Theorem(UAT) (Hornik et al., 1989; Hornik, 1991; Lu et al., 2017), this operation is equivalent in approximation power to $\sigma(A X W)$. For detailed proof of UAT application, please refer to Appendix C.5. Here, $A X$ represents the aggregation of neighbor features, and thus $A X = \sum X_i$, where $i$ represents the 1-hop in-edges. Similarly, $\widetilde{(A + I)} X W$ is $\sum X_i W_1 + X W_0$.                            □

In particular, $X$ is an $n \times f$ matrix, where each of the $n$ rows is the $f$-dimensional feature vector of a node. In contrast, $X_i$ is not a matrix. It has $n$ rows, but each row is a list of feature vectors, one for each 1-hop in-neighbor of that node. For example, row $k$ of $X_i$ contains the feature vectors of all 1-hop in-neighbors of node $k$.

**Theorem C.2.** *For all natural numbers $n > 0$, the output of an $n$-layer GCN without self-loops can be expressed as follows:*

$$X^n \approx \sigma(\sum X_{\underbrace{i\ldots i}_{n}} W),$$

*where $X_{\underbrace{i\ldots i}_{n}}$ denotes neighbours reached by $n$-hop in-edges, $X^n$ denotes representation of nodes after $n$-layer GNN.*

**Theorem C.3.** *For an $n$-layer GCN with self-loops, the output can be expressed as follows:*

$$\boldsymbol{X}^n \approx \sigma(\sum \boldsymbol{X}_{\underbrace{\boldsymbol{i}...\boldsymbol{i}}_{n}}\boldsymbol{W}_1 + \sum \boldsymbol{X}_{\underbrace{\boldsymbol{i}...\boldsymbol{i}}_{n-1}}\boldsymbol{W}_2 + ... + \boldsymbol{X}\boldsymbol{W}_{n+1}).$$

The proofs of theorems C.2 and C.3 are presented in Appendix C.4.

Next, we will prove a fundamental property of GNNs for directed graphs: scale invariance. We will demonstrate that when the input graph undergoes scaling transformations, the GCN's output remains unchanged, considering both scenarios—whether or not self-loops are added. This proof highlights that the GNN's architecture inherently preserves its effectiveness and consistency across scaled graph representations, ensuring robust performance in diverse scenarios.

## C.2 PROOF OF SCALE INVARIANCE IN GCN WITHOUT SELF-LOOPS

For different adjacency matrices, the layer-wise propagation rules and $k$-layer outputs are as follows:

### C.2.1 SINGLE-DIRECTIONAL AGGREGATION

- $\boldsymbol{A}$ as the adjacency matrix:
  1-layer: $\sigma(\sum \boldsymbol{X}_{\boldsymbol{i}}\boldsymbol{W})$; $k$-layer: $\sigma(\sum \boldsymbol{X}_{\underbrace{\boldsymbol{i}...\boldsymbol{i}}_{k}}\boldsymbol{W})$ for $k \geq 1$

- $\boldsymbol{A}^\top$ as the adjacency matrix:
  1-layer: $\sigma(\sum \boldsymbol{X}_{\boldsymbol{o}}\boldsymbol{W})$; $k$-layer: $\sigma(\sum \boldsymbol{X}_{\underbrace{\boldsymbol{o}...\boldsymbol{o}}_{k}}\boldsymbol{W})$ for $k \geq 1$

- $\boldsymbol{A}\boldsymbol{A}$ as the adjacency matrix:
  1-layer: $\sigma(\sum \boldsymbol{X}_{\boldsymbol{ii}}W)$; $k$-layer: $\sigma(\sum \boldsymbol{X}_{\underbrace{\boldsymbol{i}...\boldsymbol{i}}_{2k}}W)$ for $k \geq 1$

- $\boldsymbol{A}\boldsymbol{A}^\top$ as the adjacency matrix:
  1-layer: $\sigma(\sum \boldsymbol{X}_{\boldsymbol{io}}W)$; $k$-layer: $\sigma(\sum \boldsymbol{X}_{\underbrace{\boldsymbol{io}...\boldsymbol{io}}_{k \text{ pairs } \boldsymbol{io}}}W)$ for $k \geq 1$

Similar patterns for $\boldsymbol{A}^\top\boldsymbol{A}^\top$ and $\boldsymbol{A}^\top\boldsymbol{A}$.

From above, we can deduce:

1. $k$-layer GCN with $\boldsymbol{A}\boldsymbol{A}$ is equivalent to $2k$-layer GCN with $\boldsymbol{A}$;
2. $k$-layer GCN with $\boldsymbol{A}^\top\boldsymbol{A}^\top$ is equivalent to $2k$-layer GCN with $\boldsymbol{A}^\top$.

### C.2.2 BIDIRECTIONAL AGGREGATION

If the model uses bidirectional aggregation (Rossi et al., 2024), the $k$-layer outputs ($k \geq 1$) are as follows:

- $\boldsymbol{A}$ and $\boldsymbol{A}^\top$ as the adjacency matrices:
  $\sigma(\sum \boldsymbol{X}_{\underbrace{\boldsymbol{ii}...\boldsymbol{i}}_{k\boldsymbol{i}}}\boldsymbol{W}_1 + \sum \boldsymbol{X}_{\underbrace{\boldsymbol{ii}...\boldsymbol{io}}_{(k-1)\boldsymbol{i},1\boldsymbol{o}}}\boldsymbol{W}_2 + \cdots + \sum \boldsymbol{X}_{\underbrace{\boldsymbol{oo}...\boldsymbol{oi}}_{(k-1)\boldsymbol{o},1\boldsymbol{i}}}\boldsymbol{W}_{(2^k-1)} + \sum \boldsymbol{X}_{\underbrace{\boldsymbol{oo}...\boldsymbol{o}}_{k \text{ times } \boldsymbol{o}}}\boldsymbol{W}_{2^k})$

  There are $2^k$ possible ordered patterns formed by combinations of $\boldsymbol{i}$ and $\boldsymbol{o}$.

- $\boldsymbol{A}\boldsymbol{A}^\top$ and $\boldsymbol{A}^\top\boldsymbol{A}$ as the adjacency matrices:
  $\sigma(\sum \boldsymbol{X}_{\underbrace{\boldsymbol{io}...\boldsymbol{io}}_{k \text{ pairs } \boldsymbol{io}}}\boldsymbol{W}_1 + \cdots + \sum \boldsymbol{X}_{\underbrace{\boldsymbol{oi}...\boldsymbol{oi}}_{k \text{ pairs } \boldsymbol{oi}}}\boldsymbol{W}_{2^k})$

  Here too, there are $2^k$ possible ordered patterns formed by combinations of $\boldsymbol{io}$ and $\boldsymbol{oi}$.

- Similarly, using $\boldsymbol{A}\boldsymbol{A}$ and $\boldsymbol{A}^\top\boldsymbol{A}^\top$:
  There are $2^k$ patterns formed by ordered combinations of $\boldsymbol{ii}$ and $\boldsymbol{oo}$.

Overall, a $2k$-layer GCN with $\boldsymbol{A}$ and $\boldsymbol{A}^\top$ has $2^{2k} = 4^k$ possible ordered patterns of $\boldsymbol{i}$ and $\boldsymbol{o}$. This can be seen as combining:

- $2^k$ patterns formed by ordered combinations of $\boldsymbol{ii}$ and $\boldsymbol{oo}$; and
- $2^k$ patterns formed by ordered combinations of $\boldsymbol{io}$ and $\boldsymbol{oi}$.

From the above, we can deduce:

1. A $k$-layer GCN with $\boldsymbol{AA}^\top$ and $\boldsymbol{A}^\top\boldsymbol{A}$ is a dropout version of a $2k$-layer GCN with $\boldsymbol{A}$ and $\boldsymbol{A}^\top$. In this context, "dropout" refers to the selective aggregation of information, where specific subsets of neighbors are preserved rather than aggregating information from all neighbors at each step.

2. A $k$-layer GCN with $\boldsymbol{AA}$ and $\boldsymbol{A}^\top\boldsymbol{A}^\top$ is also a dropout version of a $2k$-layer GCN with $\boldsymbol{A}$ and $\boldsymbol{A}^\top$.

Synthesizing Section C.2.1 and Section C.2.2, we conclude that all single-directional aggregation models are dropout versions of their bidirectional counterparts. For example, a model using only $\boldsymbol{A}$ corresponds to a bidirectional model with both $\boldsymbol{A}$ and $\boldsymbol{A}^\top$, and a model using $\boldsymbol{AA}$ corresponds to a bidirectional model with both $\boldsymbol{AA}$ and $\boldsymbol{A}^\top\boldsymbol{A}^\top$. Finally, we can conclude that all models—whether single-directional or bidirectional—are dropout versions of $\boldsymbol{A}$ and $\boldsymbol{A}^\top$. Similar analysis for GCN with self-loops is presented as follows in Appendix C.3.

## C.3 Proof of Scale Invariance of GCN with Self-loops

Similarly, cases of GCN which adds selfloops are as follows:

### C.3.1 Single-Directional Aggregation

- $\boldsymbol{A}$ as adjacency matrix
  1 layer: $\sigma(\boldsymbol{XW}_0 + \sum \boldsymbol{X_i W}_1)$
  k layer: $\sigma(\boldsymbol{XW}_0 + \sum_{j=1}^k \sum \boldsymbol{X}_{\underbrace{\boldsymbol{i \ldots i}}_{j}} \boldsymbol{W}_{k-j})$

- $\boldsymbol{A}^\top$ as adjacency matrix
  1 layer: $\sigma(\boldsymbol{XW}_0 + \sum \boldsymbol{X_o W}_1)$
  k layer: $\sigma(\boldsymbol{XW}_0 + \sum_{j=1}^k \sum \boldsymbol{X}_{\underbrace{\boldsymbol{o \ldots o}}_{j}} \boldsymbol{W}_{k-j})$

- $\boldsymbol{AA}$ as adjacency matrix
  1 layer: $\sigma(\boldsymbol{XW}_0 + \sum \boldsymbol{X_{ii} W}_1)$
  k layer: $\sigma(\boldsymbol{XW}_0 + \sum_{j=1}^k \sum \boldsymbol{X}_{\underbrace{\boldsymbol{ii \ldots ii}}_{j \text{ times } \boldsymbol{ii}}} \boldsymbol{W}_{k-j})$

- $\boldsymbol{AA}^\top$ as adjacency matrix
  1 layer: $\sigma(\boldsymbol{XW}_0 + \sum \boldsymbol{X_{io} W}_1)$
  k layer: $\sigma(\boldsymbol{XW}_0 + \sum_{j=1}^k \sum \boldsymbol{X}_{\underbrace{\boldsymbol{io \ldots io}}_{j \text{ times } \boldsymbol{io}}} \boldsymbol{W}_{k-j})$

Similar patterns for $\boldsymbol{A}^\top\boldsymbol{A}^\top$ and $\boldsymbol{A}^\top\boldsymbol{A}$.

From above, we can deduce that:

1. $k$ layer $\boldsymbol{AA}$ is equivalent of dropout of $2k$ layer $\boldsymbol{A}$.

2. $k$ layer $\boldsymbol{A}^\top\boldsymbol{A}^\top$ is equivalent of dropout of $2k$ layer $\boldsymbol{A}^\top$.

### C.3.2 Bi-Directional Aggregation

- $\boldsymbol{A}$ and $\boldsymbol{A}^\top$ as adjacency matrix

1 layer: $\sigma(\boldsymbol{X}\boldsymbol{W}_0 + \sum \boldsymbol{X_i}\boldsymbol{W}_1 + \sum \boldsymbol{X_o}\boldsymbol{W}_2)$

k layer: $\sigma(\boldsymbol{X}\boldsymbol{W}_0 + \sum \boldsymbol{X_i}\boldsymbol{W}_{1,1} + \sum \boldsymbol{X_o}\boldsymbol{W}_{1,2} + \ldots + \sum \boldsymbol{X}_{\underbrace{\boldsymbol{ii\ldots i}}_{k}}\boldsymbol{W}_{k,1} + \sum \boldsymbol{X}_{\underbrace{\boldsymbol{oi\ldots i}}_{k}}\boldsymbol{W}_{k,2} +$

$\ldots + \sum \boldsymbol{X}_{\underbrace{\boldsymbol{o\ldots o}}_{k}}\boldsymbol{W}_{k,2^k})$

or more simply: $\sigma(\boldsymbol{X}\boldsymbol{W}_0 + \sum_{j=1}^{k}\sum_{e_j \in \{\boldsymbol{i},\boldsymbol{o}\}^j} \boldsymbol{X}_{e_j}\boldsymbol{W}_{j,\mathrm{pos}(e_j)})$, where $\{\boldsymbol{i},\boldsymbol{o}\}^j$ is the set of all possible sequence combinations of length $j$, and $\mathrm{pos}(e_j)$ is the positional index of sequence $e_j$ (ranging from 1 to $2^j$).

- $\boldsymbol{AA}$ and $\boldsymbol{A}^\top\boldsymbol{A}^\top$ as adjacency matrix
  1 layer: $\sigma(\boldsymbol{X}\boldsymbol{W}_0 + \sum \boldsymbol{X_{ii}}\boldsymbol{W}_1 + \sum \boldsymbol{X_{oo}}\boldsymbol{W}_2)$
  k layer: $\sigma(\boldsymbol{X}\boldsymbol{W}_0 + \sum_{j=1}^{k}\sum_{e_{2j} \in \{\boldsymbol{ii},\boldsymbol{oo}\}^j} \boldsymbol{X}_{e_j}\boldsymbol{W}_{j,\mathrm{pos}(e_{2j})})$,
  where $\{\boldsymbol{ii},\boldsymbol{oo}\}^j$ is the set of all possible sequence combinations of length $j$, and $\mathrm{pos}(e_j)$ is the positional index of sequence $e_{2j}$ (ranging from 1 to $2^j$).

- $\boldsymbol{AA}^\top$ and $\boldsymbol{A}^\top\boldsymbol{A}$ as adjacency matrix
  1 layer: $\sigma(\boldsymbol{X}\boldsymbol{W}_0 + \sum \boldsymbol{X_{io}}\boldsymbol{W}_1 + \sum \boldsymbol{X_{oi}}\boldsymbol{W}_2)$

  k layer: $\sigma(\boldsymbol{X}\boldsymbol{W}_0 + \sum_{j=1}^{k}\sum_{e_{2j} \in \{\boldsymbol{io},\boldsymbol{oi}\}^j} \boldsymbol{X}_{e_j}\boldsymbol{W}_{j,\mathrm{pos}(e_{2j})})$,
  where $\{\boldsymbol{io},\boldsymbol{oi}\}^j$ is the set of all possible sequence combinations of length $j$, and $\mathrm{pos}(e_j)$ is the positional index of sequence $e_{2j}$ (ranging from 1 to $2^j$).

From above, we can deduce that:

1. $k$ layer $\boldsymbol{AA} + \boldsymbol{A}^\top\boldsymbol{A}^\top$ is equivalent of dropout of $2k$ layer $\boldsymbol{A}$ and $\boldsymbol{A}^\top$.
2. $k$ layer $\boldsymbol{AA}^\top + \boldsymbol{A}^\top\boldsymbol{A}$ is equivalent of dropout of $2k$ layer $\boldsymbol{A}$ and $\boldsymbol{A}^\top$.

In conclusion, our theoretical analysis confirms that propagating information through higher-scale adjacency matrices is fundamentally equivalent to applying lower-scale graph operations or their dropout variants. This equivalence not only supports the theoretical validity of scale invariance in graph neural networks but also ensures that the use of multi-scale graphs retains the benefits of invariance across different graph structures.

Furthermore, as undirected graphs can be treated as a special case of directed graphs, where in-neighbors and out-neighbors are identical, the proof of scale invariance extends seamlessly to undirected graph structures. These findings provide a solid foundation for developing more efficient and scalable graph neural network models that leverage multi-scale graph representations.

While we demonstrate the proof of scale invariance specifically for GCN, similar mathematical arguments can be constructed for GraphSAGE and other GNN variants. These findings provide a solid foundation for developing more efficient and scalable graph neural network models that leverage multi-scale graph representations.

## C.4 OUTPUT OF $n$-LAYER GCN

Non-linear functions are integral components of all models discussed here. To streamline the exposition and focus on core mechanisms, they are omitted in following presentation.

### C.4.1 $n$-LAYER GCN (WITHOUT SELF-LOOPS)

The proof of Theorem C.2 is presented in this section.

*Proof.* **Base Case:** In a 1-layer GCN, the output is expressed as:
$$\boldsymbol{X}^1 = \sigma(\sum \boldsymbol{X_i}\boldsymbol{W}),$$
which is consistent with the desired form.

For a 2-layer GCN, the output becomes:
$$\boldsymbol{X}^2 = \sigma\big(\sum (\boldsymbol{X}^1)_i \boldsymbol{W}_1\big) = \sigma\big(\sum (\sum \boldsymbol{X_i}\boldsymbol{W}_2)_i \boldsymbol{W}_1\big).$$

According to the Universal Approximation Theorem, neural networks can approximate any continuous function, given sufficient capacity and appropriately chosen weights. This theorem implies that rearranging the order of linear operations, such as:

$$\sigma\big(\sum(\sum \boldsymbol{X_i}\boldsymbol{W_2})_i\boldsymbol{W_1}\big) = \sigma\big(\sum(\sum(\boldsymbol{X_i})_i\boldsymbol{W_2})\boldsymbol{W_1}\big) \approx \sigma\Big(\big(\sum\sum(\boldsymbol{X_i})_i\big)\boldsymbol{W}\Big)$$

does not affect the network's ability to approximate the target function. Since a node's 1-hop neighbor's 1-hop neighbors are the node's 2-hop neighbors, we have:

$$\boldsymbol{X}^2 \approx \sigma\big(\sum \boldsymbol{X_{ii}}\boldsymbol{W}\big).$$

This satisfies the form for $n = 2$.

**Inductive Step:** Assume that the statement holds for $n = k$, i.e.,

$$\boldsymbol{X}^k \approx \sigma\big(\sum \boldsymbol{X}_{\underbrace{\boldsymbol{i\ldots i}}_{k}}\boldsymbol{W}\big).$$

For $n = k + 1$:

$$\boldsymbol{X}^{k+1} \approx (\boldsymbol{X}^k)^1 \approx \sigma\big(\sum \boldsymbol{X}_i^k\boldsymbol{W_1}\big) \approx \sigma\left(\sum\sigma\big(\sum(\boldsymbol{X}_{\underbrace{\boldsymbol{i\ldots i}}_{k}})_i\boldsymbol{W_2}\big)\boldsymbol{W_1}\right)$$

$$\approx \sigma\big(\sum\sum(\boldsymbol{X}_{\underbrace{\boldsymbol{i\ldots i}}_{k}})_i\big)\boldsymbol{W_2}\boldsymbol{W_1}\big) \approx \sigma\big(\sum \boldsymbol{X}_{\underbrace{\boldsymbol{i\ldots i}}_{k+1}}\boldsymbol{W}\big)$$

Thus, the statement holds for $n = k + 1$.

**Conclusion:** By the principle of mathematical induction, we conclude that for all $n \geq 1$,

$$\boldsymbol{X}^n \approx \sigma\big(\sum \boldsymbol{X}_{\underbrace{\boldsymbol{i\ldots i}}_{n}}\boldsymbol{W}\big).$$

$\square$

### C.4.2 $n$-LAYER GCN (WITH SELF-LOOPS)

The proof of Theorem C.3 is presented in this section.

*Proof.* **Base Case:** For $n = 1$, the output of a 1-layer GCN with self-loops is:

$$\boldsymbol{X}^1 \approx \sigma\big(\sum \boldsymbol{X_i}\boldsymbol{W_1} + \boldsymbol{X}\boldsymbol{W_2}\big),$$

which matches the desired form.

**Case $n = 2$:** According to layer-wise propagation rule, the output is:

$$\boldsymbol{X}^2 = \sigma\left(\sum\sigma\left(\sum(\boldsymbol{X}^1)_i\boldsymbol{W_2}\right)\boldsymbol{W_1} + \boldsymbol{X}^1\boldsymbol{W_3}\right).$$

Substituting $\boldsymbol{X}^1 \approx \sum \boldsymbol{X_i}\boldsymbol{W_1} + \boldsymbol{X}\boldsymbol{W_2}$:

$$\boldsymbol{X}^2 \approx \sigma\big(\sum \boldsymbol{X_{ii}}\widehat{\boldsymbol{W}}_1 + \sum \boldsymbol{X_i}\widehat{\boldsymbol{W}}_2 + \boldsymbol{X}\widehat{\boldsymbol{W}}_3\big),$$

where $\widehat{\boldsymbol{W}}_1$ and $\widehat{\boldsymbol{W}}_2$ are suitably combined weight matrices. This matches the desired form for $n = 2$.

**Inductive Step:** Assume that the statement holds for $n = k$, i.e.,

$$\boldsymbol{X}^k \approx \sigma\big(\sum \boldsymbol{X}_{\underbrace{\boldsymbol{i\ldots i}}_{k}}\boldsymbol{W_1} + \sum \boldsymbol{X}_{\underbrace{\boldsymbol{i\ldots i}}_{k-1}}\boldsymbol{W_2} + \cdots + \boldsymbol{X}\boldsymbol{W}_{k+1}\big). \qquad (A.3)$$

From the GCN formulation, we have:

$$\boldsymbol{X}^{k+1} = \sigma\left(\sum\sigma\big(\sum(\boldsymbol{X}^k)_i\boldsymbol{W_2}\big)\boldsymbol{W_1} + \boldsymbol{X}^k\boldsymbol{W_3}\right).$$

Substituting the inductive hypothesis for $\boldsymbol{X}^k$ from Equation equation A.3, we get:

$$\boldsymbol{X}^{k+1} \approx \sigma\Big(\sum \boldsymbol{X}_{\underbrace{i\ldots i}_{k+1}}\boldsymbol{W}_1 + \sum \boldsymbol{X}_{\underbrace{i\ldots i}_{k}}\boldsymbol{W}_2 + \cdots + \boldsymbol{X}\boldsymbol{W}_{k+2}\Big).$$

**Conclusion:** By the principle of mathematical induction, the statement holds for all $n \geq 1$:

$$\boldsymbol{X}^{n} \approx \sigma\Big(\sum \boldsymbol{X}_{\underbrace{i\ldots i}_{n}}\boldsymbol{W}_1 + \sum \boldsymbol{X}_{\underbrace{i\ldots i}_{n-1}}\boldsymbol{W}_2 + \cdots + \boldsymbol{X}\boldsymbol{W}_{n+1}\Big).$$

$\square$

$\square$

### C.5 PROOF OF UAT APPLICATION IN THIS PAPER

The Universal Approximation Theorem (UAT) states that a single hidden layer with enough neurons and a non-linear activation (e.g., sigmoid or ReLU) can approximate any continuous function on compact subsets of $\mathbb{R}^n$ to arbitrary precision (Hornik, 1991; Hornik et al., 1989).

#### C.5.1 EMPIRICAL VERIFICATION OF SUFFICIENT NEURONS

To empirically verify the Universal Approximation Theorem, we use the best hyperparameter settings for Chameleon and test the ScaleNet model by varying the number of hidden neurons from 1 to 22.

As shown in Figure C.1, the model achieves a high accuracy of 76.7% with 20 hidden neurons. Increasing the number of neurons to 21 and 22 results in a slight decrease in accuracy. The highest observed accuracy is 80.1%, achieved with 128 hidden neurons. These results demonstrate that even with as few as 20 neurons, the model is capable of effective function approximation.

This supports the theoretical claim that a single hidden layer with a sufficient number of neurons can approximate complex functions, and it further suggests that commonly used sizes such as 64 or 128 neurons are more than adequate in practice.

#### C.5.2 APPLICATION OF UAT IN THIS PAPER

**Simplifying a nested non-linearity model by removing the inner non-linearity** The UAT justifies our key architectural simplification: removing inner non-linearities while preserving approximation power. For a two-layer network with nested activations:

$$\sigma\Big(\boldsymbol{A}\sigma(\boldsymbol{A}\boldsymbol{X}\boldsymbol{W}_1)\boldsymbol{W}_2\Big) \approx \sigma\Big(\boldsymbol{A}\boldsymbol{A}\boldsymbol{X}\boldsymbol{W}_1\boldsymbol{W}_2\Big) \tag{A.4}$$

where the approximation holds because single-layer networks with sufficient width is enough to approximate target function, which is the classification of nodes in graph.

For notational simplicity, we omit the non-linearity, leading to the simplified expression:

$$\boldsymbol{A}(\boldsymbol{A}\boldsymbol{X}\boldsymbol{W}_1)\boldsymbol{W}_2 \approx \boldsymbol{A}\boldsymbol{A}\boldsymbol{X}\boldsymbol{W}_1\boldsymbol{W}_2. \tag{A.5}$$

This step follows directly from standard function approximation principles under UAT, which states that a sufficiently expressive neural network can approximate any continuous function. The simplification of nested activations into a single activation is well-established in the literature (Lu et al., 2017; Hornik, 1991; Hanin & Sellke, 2017).

**Parameter Compression via Weight Combination** Further, leveraging UAT, Equation A.5 can be simplified by merging the two weight matrices:

$$\boldsymbol{A}\boldsymbol{A}\boldsymbol{X}\boldsymbol{W}_1\boldsymbol{W}_2 \approx \boldsymbol{A}\boldsymbol{A}\boldsymbol{X}\boldsymbol{W}, \tag{A.6}$$

where $\boldsymbol{W}$ is a newly parameterized matrix independent of $\boldsymbol{W}_1$ and $\boldsymbol{W}_2$. This simplification is valid because the number and specific arrangement of weight matrices do not affect the model's universal approximation capability.

Our proofs in Appendix C.4 and C.3 crucially rely on these simplifications.

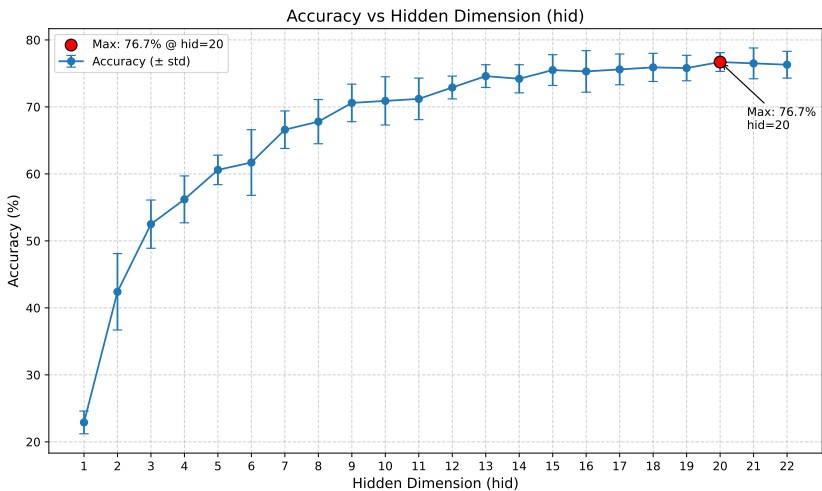

Figure C.1: Accuracy trend with increasing hidden neurons from 1 to 22.

**Previous Application of UAT in GNN** Similar applications of UAT in Graph Neural Networks (GNNs) have been explored in prior work (Xu et al., 2019), reinforcing our approach. Empirical studies have also validated the effectiveness of removing inside non-linearity in GNN architectures (Wu et al., 2019).

## C.6 EMPIRICAL DEMONSTRATION OF SCALE INVARIANCE

Table C.1: Accuracy obtained by each scaled ego-graph using a single split per ego-graph. Higher scaled graphs maintain the discerning ability of their lower scale counterparts, even after removing the shared edges from lower scale graphs (in parentheses). For homophilic graphs, both $A$ and $A^\top$ perform well, and all $2^{nd}$-scale graphs preserve this performance. In heterophilic graphs, $A$ performs well while $A^\top$ does not. $AA$ and $AA^\top$ preserve the performance of $A$, whereas $A^\top A^\top$ and $A^\top A$ inherit the poor performance of $A^\top$. ( The final column presents the performance with all zero input for comparison. The value in parentheses is the scaled graph after removing the shared edges with $A$ or $A^\top$. "+" denotes the addition of aggregation outputs.)

| Type | Dataset | $A$ | $A^\top$ | $A+A^\top$ | $AA^\top$ | $A^\top A$ | $AA+A^\top A$ | $AA$ | $A^\top A^\top$ | $AA+A^\top A^\top$ | None |
|---|---|---|---|---|---|---|---|---|---|---|---|
| | Telegram | 68 | 74 | 100 | 60 (58) | 68 (78) | 92 (94) | 72 (50) | 80 (78) | 92 (92) | 38 |
| | Cora-ML | 75 | 70 | 78 | 78 (73) | 77 (69) | 79 (78) | 75 (80) | 72 (78) | 77 (79) | 29 |
| Homophilic | CiteSeer | 56 | 59 | 62 | 62 (57) | 64 (59) | 63 (61) | 57 (53) | 60 (60) | 63 (63) | 20 |
| | WikiCS | 73 | 66 | 75 | 74 (76) | 65 (69) | 76 (78) | 73 (75) | 66 (67) | 73 (77) | 23 |
| Heterophilic | Chameleon | 78 | 30 | 68 | 66 (70) | 29 (29) | 68 (71) | 70 (76) | 30 (31) | 66 (69) | 22 |
| | Squirrel | 75 | 33 | 68 | 73 (73) | 31 (31) | 70 (67) | 75 (73) | 32 (33) | 66 (66) | 19 |

Table C.2: This figure illustrates the distribution of nodes in the Chameleon and Squirrel datasets, categorized by the predominant label of their aggregated neighbors in relation to the node's own label. The analysis reveals that when using $A^\top$ as the adjacency matrix, a majority of nodes have zero aggregated neighbors. This lack of connectivity results in poor model performance

| Dataset | Direction | Homo. | Hetero. | No Neigh. |
|---|---|---|---|---|
| Chameleon | $A$ | 576 | 1701 | 0 |
| | $A^\top$ | 237 | 627 | 1413 |
| Squirrel | $A$ | 1258 | 3943 | 0 |
| | $A^\top$ | 441 | 1764 | 2996 |

In Table C.1, we demonstrate the presence of scale invariance in graphs through various experiments. We represent the graph structure using a scaled adjacency matrix, which is then fed into a GNN for node classification. The results in Table C.1 show that higher-scale graphs consistently achieve performance comparable to their lower-scale counterparts, confirming scale invariance. In contrast, if scale invariance were absent, the performance of higher-scale graphs would be similar to the results shown in the last column, where no input is provided.

Additionally, combinations of ego-graphs with adverse directional edges tend to yield better results than the individual ego-graphs. For example, $AA^\top + A^\top A$ generally achieves better accuracy compared to $AA^\top$ and $A^\top A$ individually, and $AA + A^\top A^\top$ performs better than $AA$ and $A^\top A^\top$.

For heterophilic graphs like Chameleon and Squirrel:

- A outperforms $A^\top$ in classification tasks. This means aggregating information from out-neighbors works better than from in-neighbors for these datasets.

- This trend extends to higher-order relationships: $AA^\top$ and $AA$ perform better than $A^\top A$ and $A^\top A^\top$. This suggests mutual out-neighbors or 2-hop out-neighbors capture similarity more effectively than their in-neighbor counterparts.

- The reason: Most nodes have 0 neighbors when using $A^\top$. After aggregation, these nodes' features are updated to all zeros.

Overall, Table C.1 demonstrates that higher-scale graphs consistently perform no worse than their lower-scale counterparts, confirming the scale invariance of graph structures. The table also provides insights into how performance varies with different graph scales and characteristics across datasets.

## C.7 ABLATION STUDY: HYPERPARAMETER SENSITIVITY

To understand the impact of key hyperparameters on model performance, we conduct comprehensive ablation studies on aggregation direction and self-loops across homophilic and heterophilic graphs. The results are presented in Table C.3.

**Aggregation Direction Sensitivity.** We examine the sensitivity of our model to the direction parameters $\alpha$, $\beta$, and $\gamma$ by testing three configurations: unidirectional forward (1), unidirectional backward (0), and bi-directional (0.5). The results reveal a clear pattern based on graph homophily. For homophilic graphs (Telegram, Cora-ML, CiteSeer, and WikiCS), bi-directional aggregation ($\alpha = \beta = \gamma = 0.5$) consistently achieves the best performance across all scales. This suggests that homophilic graphs benefit from information flow in both directions, allowing nodes to effectively leverage similar neighboring nodes regardless of edge direction.

In contrast, heterophilic graphs (Chameleon and Squirrel) demonstrate the opposite behavior, achieving optimal performance with unidirectional aggregation ($\alpha = \beta = \gamma = 1$). This finding indicates that for heterophilic graphs, preserving the original edge direction is crucial for maintaining the structural information that distinguishes dissimilar connected nodes.

**Self-Loop Analysis.** The self-loop ablation study compares performance with and without self-loops across multiple dataset splits. For homophilic graphs, adding self-loops consistently improves performance, with improvements ranging from 0.6% on Telegram to 4.2% on WikiCS. This enhancement can be attributed to nodes being able to retain their own features while aggregating information from similar neighbors, creating a beneficial reinforcement effect in homophilic settings.

Conversely, heterophilic graphs show degraded performance when self-loops are added, with decreases of 6.7% on Chameleon and 8.9% on Squirrel. This degradation occurs because self-loops dilute the heterophilic signal by mixing a node's features with dissimilar neighboring information, reducing the model's ability to distinguish between different node types.

These findings provide clear guidelines for hyperparameter selection: use bi-directional aggregation with self-loops for homophilic graphs, and unidirectional aggregation without self-loops for heterophilic graphs.

Table C.3: Ablation study on aggregation direction and self-loops. Direction ablation uses single-split accuracy (%) for efficiency, while self-loop ablation results are averaged over multiple splits. The best direction for each scale is shown in **bold**. Homophilic graphs achieve better performance with bi-directional aggregation ($\alpha = \beta = \gamma = 0.5$) and self-loops, while heterophilic graphs perform better with uni-directional aggregation ($\alpha = \beta = \gamma = 1$) without self-loops.

| Scale | Direction | Homophilic Graphs | | | | Heterophilic Graphs | |
|---|---|---|---|---|---|---|---|
| | | Telegram | Cora-ML | CiteSeer | WikiCS | Chameleon | Squirrel |
| $\alpha$ | 0 | 74.0 | 70.3 | 58.9 | 68.0 | 29.8 | 31.0 |
| | 1 | 68.0 | 75.4 | 56.4 | 74.9 | **77.9** | **74.7** |
| | 0.5 | **100.0** | **77.6** | **62.4** | **78.6** | 68.0 | 67.0 |
| $\beta$ | 0 | 68.0 | 77.2 | 64.1 | 66.6 | 28.5 | 31.6 |
| | 1 | 60.0 | 77.8 | 61.6 | 66.5 | 65.8 | **70.5** |
| | 0.5 | **92.0** | **78.6** | **63.0** | **75.4** | **67.8** | 62.6 |
| $\gamma$ | 0 | 80.0 | 72.0 | 59.6 | 65.1 | 30.0 | 31.9 |
| | 1 | 72.0 | 74.9 | 56.9 | 72.2 | **70.0** | **74.0** |
| | 0.5 | **92.0** | **77.1** | **62.5** | **75.3** | 66.2 | 66.6 |
| Self-loop | 0 | 96.6±2.7 | 80.4±1.6 | 66.9±1.6 | 75.5±0.8 | **80.1±1.5** | **76.0±2.0** |
| | 1 | **97.2±2.1** | **82.3±1.1** | **69.1±1.2** | **79.6±0.7** | 73.4±1.6 | 67.1±1.5 |

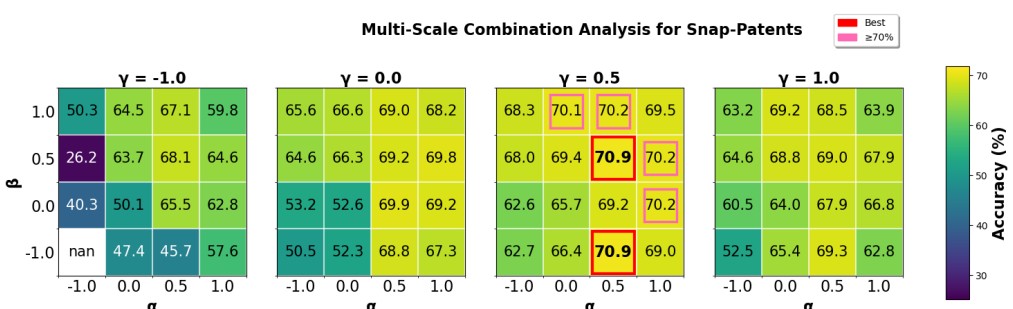

Figure C.2: Ablation study showing accuracy (%) with different combinations of multiple scales on the Snap-Patents dataset. Red boxes highlight the best performing configuration, while pink boxes indicate high-performance configurations ($\geq 70.0\%$).

## C.8 ABLATION STUDY ON LARGE GRAPHS

We conduct comprehensive ablation studies on additional large graph datasets to validate the generalizability of our multi-scale learning approach. For the Roman-Empire dataset (Table 3), we systematically vary the scale parameters $\alpha$, $\beta$ and $\gamma$ while keeping all other hyperparameters fixed: learning rate = 0.01, 5 layers, early stopping after 80 consecutive epochs without improvement. We use LargeScaleNet with a single data split for computational efficiency.

The ablation study on Snap-Patents is presented in Figure C.2. Consistent with the Roman-Empire results, Snap-Patents demonstrates that various multi-scale combinations achieve strong performance, confirming the robustness of our approach across different graph structures. For this dataset, we maintain consistent experimental settings with only scale components varying: early stopping after 10 epochs without improvement, 2 layers, learning rate = 0.0001, hidden dimension = 64, and single data split.

These results across multiple large-scale datasets provide strong empirical evidence that multi-scale learning consistently captures complementary structural information, leading to improved performance regardless of the specific graph domain or characteristics.

# D  A MORE DETAILED REVIEW

## D.1  REVIEW IN BROADER CONTEXT

### D.1.1  REVIEW OF GNNs

GNNs extend MLPs by adding a message passing sub-layer before the linear transformation and activation. Let $X$ denote node features, $A$ denote the adjacency matrix, $\hat{A}$ denote normalized $A$, $W$ denote the weight matrix, and $I$ denote the identity matrix. Different GNN architectures primarily vary in their message passing mechanisms. In this paper, non-linear activation is omitted for simplicity of expression, while all models have non-linear activations. Table D.1 lists the message passing filters for these architectures.

Table D.1: Different message passing filters of GNNs. $\odot$ denote element-wise multiplication.

| Model | Configuration | Layer-wise Propagation Output |
|---|---|---|
| **MLP** | | $XW$ |
| **GCN** | No SelfLoop | $\tilde{A}XW$ |
| | Add SelfLoop | $\widetilde{(I+A)}XW$ |
| **SAGE** | No SelfLoop | $D^{-1}AXW_1 + XW_0$ |
| | Add SelfLoop | $(D+I)^{-1}(I+A)XW_1 + XW_0$ |
| **GAT** | Learned edge weights $W_{att}$ for each neighbour | |
| | No SelfLoop | $(W_{att} \odot A)XW$ |
| | Add SelfLoop | $\big(W_{att} \odot (A+I)\big)XW$ |
| **ChebNet** | Recurrence Formula: $T_1(A) = I, T_2(A) = I - \tilde{A}, T_{k+2}(A) = 2\tilde{A}T_{k+1}(A) - T_k(A)$ | |
| | K=1 | $T_1(A)XW = XW$ |
| | K=2 | $XW_1 + (I - \tilde{A})XW_2$ |
| | K=k | $XW_1 + (I - \tilde{A})XW_2 + ... + T_k(A)XW_k$ |
| **APPNP** | Recurrence Formula: $P_{k+1} = (1-\alpha)\tilde{A}P_k + \alpha XW$ | |
| | K=1(No Selfloop) | $P_1 = (1-\alpha)\tilde{A}XW_1 + \alpha XW_2$ |
| | K=2(No Selfloop) | $P_2 = (1-\alpha)\tilde{A}P_1 + \alpha XW_3$ |
| | | $= (1-\alpha)^2(\tilde{A})^2XW_1 + (1-\alpha)\alpha\tilde{A}XW_2 + \alpha XW_3$ |
| | K=k(No Selfloop) | $P_k = \big((1-\alpha)\tilde{A}\big)^k XW_1 + \big((1-\alpha)\tilde{A}\big)^{(k-1)}\alpha XW_2 + ...+$ |
| | | $\big((1-\alpha)\tilde{A}\big)^2\alpha XW_{k-1} + \big((1-\alpha)\tilde{A}\big)\alpha XW_k + \alpha XW_{k+1}$ |
| | Add Selfloop | replace $\tilde{A}$ with $\widetilde{(I+A)}$ in $P_k$ |

1. GCN Graph Convolutional Network (GCN) (Kipf & Welling, 2016) aggregates and normalizes neighbor information. $\tilde{A}$ denotes incidence-normalized $A$.

2. SAGE GraphSAGE (Hamilton et al., 2017) concatenates self-node with its neighbors. For neighbor feature aggregation, GraphSAGE computes the average embedding of the neighboring nodes, which is equivalent to row normalization.

   • Without self-loops: $D^{-1}AXW_1 + XW_0$
   • With self-loops: $(D+I)^{-1}(A+I)XW_1 + XW_0$

   Note that adding self-loops does not affect the nodes that GraphSAGE gathers from, but only influence their weights. While SAGE can assign different weights to self-nodes and neighbor nodes, making it potentially more expressive than GCN, it lacks incidence normalization (Kipf & Welling, 2016). This can lead to performance degradation when high-degree nodes receive disproportionate weights.

3. ChebNet ChebNet (Defferrard et al., 2016) is typically recognized as a spectral method, which uses Chebyshev polynomials of the normalized Graph Laplacian $\tilde{L} = I - \tilde{A}$, following:

$$T_{k+2} = 2\tilde{L}T_{k+1} - T_k = 2(I - \tilde{A})T_{k+1} - T_k, \tag{A.7}$$

where $T_1 = I$ and $T_2 = I - \tilde{A}$. By combining these polynomials with learnable weights ($\sum_{i=1}^{k} T_i X W_i$), ChebNet effectively aggregates information from k-hop neighborhoods, making it functionally equivalent to a spatial method despite its spectral formulation.

## D.2 AGGREGATION DIRECTION OF DIFFERENT MODELS

To aggregate features from neighbors, different GNN architectures handle aggregation directions differently. Dir-GCN (Rossi et al., 2024) uses $(AX)W$, where $W$ is a learnable weight matrix. In contrast, GCN (Kipf & Welling, 2016) uses a propagation function propagate($A$, $XW$), where the propagation step performs message passing using $A$.

However, empirical results show that these two approaches aggregate from opposite directions: using $A$ in matrix multiplication (as in Dir-GCN) is equivalent to using $A^\top$ in the propagation step (as in GCN). In other words, $AX = $ propagate($A^\top$, $X$).

## D.3 REVIEW OF GRAPH TRANSFORMERS

With the rise of Transformer architectures (Vaswani et al.) and their success in large language models and computer vision, Graph Transformers (GTs) attempt to bring these advantages to graph-structured data. However, GTs face two major problems: (1) by using full global attention, they lose graph-structural information, and (2) full attention introduces quadratic computational complexity.

To make up for the loss of structural information, GTs rely on positional and structural encodings (PE/SE). These encodings augment the original node features with graph-aware information—such as Laplacian eigenvectors (Dwivedi & Bresson)—to give the otherwise graph-unaware Transformer a sense of each node's location in the graph (Rampášek et al.). PE/SE can be added to node features, used to guide message passing, or create attention biases between distant nodes. Several powerful positional and structural encodings have been proposed (Srinivasan & Ribeiro; Mialon et al.).

Another problem of GT is quadratic complexity of full attention. To solve this, GraphMamba (Behrouz & Hashemi, 2024), inspired by Mamba (Gu & Dao), tried to reduce the quadratic computational cost to linear time. When GT models try to improve scalability, however, they often suffer in performance. Recent models such as Polynormer (Deng et al.) attempt to balance this trade-off between expressivity and scalability. GraphGPS (Rampášek et al.) tried to tackle both structural encoding and scalablity, and provide a modular framework.

In addition, while multi-head is helpful in Large Language models, SGFormer (Wu et al.) shows that multi-head attention is not necessarily better than single-head attention and achieves competitive performance without positional encodings, while remaining scalable.

GTs have achieved strong results in graph-level tasks involving small graphs (e.g., molecular graphs) (Ying et al., a; Luo et al.). However, recent empirical work (Luo et al., 2024a) shows that classic GNNs still match or even exceed GTs on node-classification tasks.

## D.4 REVIEW OF INVARIANT LEARNING

### D.4.1 INVARIANT LEARNING TECHNIQUES

Invariant classifiers generally exhibit smaller generalization errors compared to non-invariant techniques (Wu et al., 2022). Therefore, explicitly enforcing invariance in GNNs could potentially improve their robustness and accuracy (Sokolic et al., 2017). To improve the generalization ability of GNNs, in this research we focus on invariant learning techniques for node classification.

An invariant classifier (Sokolic et al., 2017) is less affected by specific transformations of the input than non-invariant classifiers. Invariant learning techniques have been well studied in Convolutional Neural Networks (CNNs), where they address translation, scale, and rotation invariance for image classification (Cohen & Welling, 2016; Lenc & Vedaldi, 2015).

However, the application of invariant classifiers in GNNs is less explored. The non-Euclidean nature of graphs introduces extra complexities that make it challenging to directly achieve invariance, and thus the invariance methods derived from CNNs cannot be straightforwardly applied.

### D.4.2 INVARIANCE OF GRAPHS

In Graph Neural Networks (GNNs), two types of permutation invariance are commonly utilized:

1. **Global Permutation Invariance**: Across the entire graph, the output of a GNN should remain consistent regardless of the ordering of nodes in the input graph. This property is particularly useful for graph augmentation techniques (Xie et al., 2023).

2. **Local Permutation Invariance**: At each node, permutation-invariant aggregation functions ensure that the results of operations remain unaffected by the order of input elements within the node's neighborhood.

Despite these established forms of invariance, the exploration of invariance in graphs is still limited. Current research is primarily preliminary, focusing on aspects such as generalization bounds (Garg et al., 2020; Verma & Zhang, 2019) and permutation-invariant linear layers (Maron et al., 2018), with few advances beyond these initial investigations.

Current research on graph invariance learning techniques can be categorized into two main areas (Sui et al., 2023).

**1. Invariant Graph Learning** focuses on capturing stable features by minimizing empirical risks, primarily to tackle out-of-distribution generalization. For instance, (Chen et al., 2023) learn the invariance among data from different environments. (Xia et al., 2024) learns invariant representation across different clusters.

**2. Graph Data Augmentation** encompasses both random and non-random methods, as detailed below:

- **Random augmentation** introduces variability into graph features to improve generalization (You et al., 2020) and may include adversarial strategies (Suresh et al., 2021). However, excessive random augmentation can disrupt stable features and lead to uncontrolled distributions.

- **Non-Random Augmentation** involves specifically designed techniques such as graph rewiring (Sun et al., 2024) and graph reduction (Hashemi et al., 2024). Graph reduction creates various perspectives of a graph through reductions at different ratios, thus augmenting the data for subsequent models. Examples include graph pooling (Gao et al., 2020), multi-scale graph coarsening (Liang et al., 2021; Ying et al., b), and using synthetic nodes to represent communities (Gao & Wu, 2023). Among these, augmenting connections with high-order neighborhoods is a particularly popular technique.

To the best of our knowledge, there is currently no graph data augmentation method based on invariance.

## E CASE STUDIES OF DIRECTED GRAPH MODELS

### E.1 DIGRAPH INCEPTION NETWORKS

Table E.1: Performance of Inception models on the Telegram dataset. "BN" indicates the addition of batch normalization to the original model. The **R**iG(ib) model assigns random weights in uniform distribution to edges within the range [0.0001, 10000], and The **1**iG(ib) model assigns weight 1 to all scaled edges.

| Model | No BN | BN | Model | No BN | BN |
|-------|-------|----|-------|-------|----|
| DiG | 67.4±8.1 | 63.0±7.6 | DiGib | 68.4±6.2 | 77.4±5.1 |
| **1**iG | 86.0±3.4 | 95.8±3.5 | **1**iGib | 86.2±3.2 | 94.2±2.7 |
| **R**iG | 85.2±2.5 | 91.0±6.3 | **R**iGib | 86.4±6.2 | 86.4±6.6 |

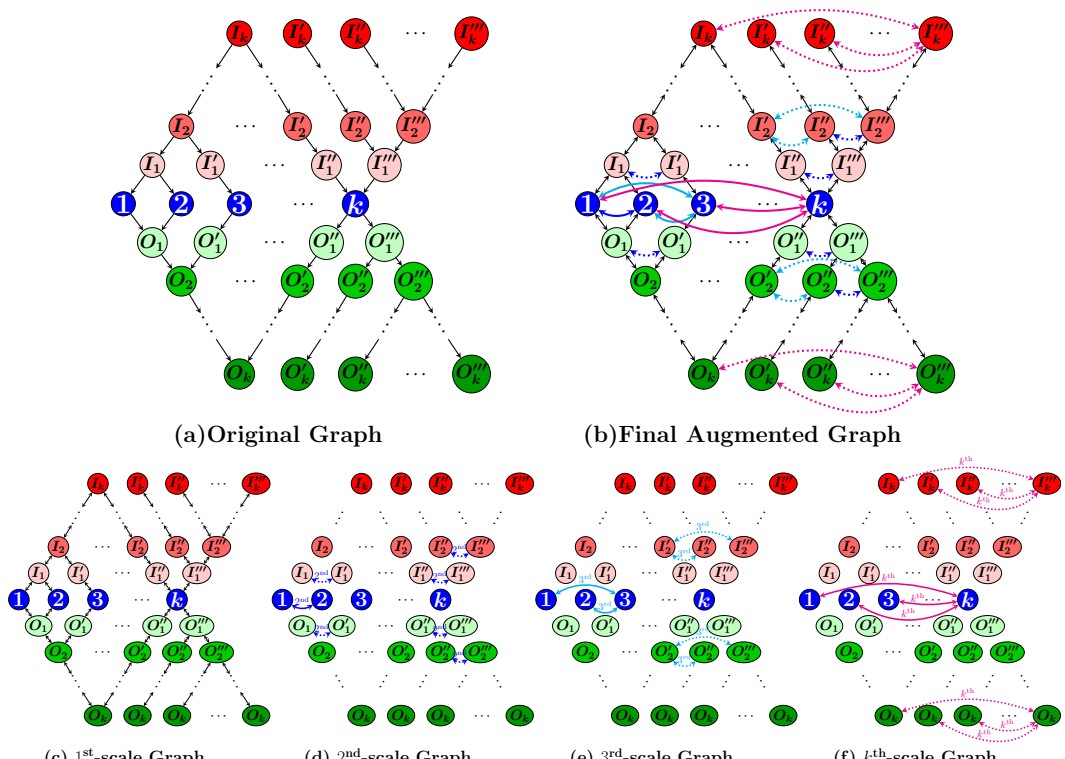

Figure E.1: Edge augmentation by stacking multi-scale graphs.

State-of-the-art Graph Neural Networks (GNNs) for homophilic graphs include Digraph Inception Networks such as DiGCN(ib) (Tong et al., 2020a) and SymDiGCN (Tong et al., 2020b), which incorporate higher-order proximity to obtain multi-scale features. However, these methods are based on random walks, making edge weights across various scales crucial. DiGCN(ib) uses resource-intensive eigenvalue decomposition to determine weights, while SymDiGCN relies on costly incidence normalization based on node degrees in the original graph, which limits their scalability to larger graphs.

In contrast, our method is simpler and leverages scale invariance. By transforming the graph at different scales and fusing predictions from these scaled graphs, our inception model effectively incorporates multi-scale features without the heavy computational costs incurred by existing methods.

Figure E.1 illustrates the corresponding graph transformation incorporating the same scaled edges as DiGCN(ib).

DiGCN(ib) incorporates various proximities:

- $1^{st}$-**order proximity** involves augmenting adverse edges, as shown in Figure E.1(c).

- $2^{nd}$-**order proximity** can be represented by the intersection or union of $\boldsymbol{A}^{\top}\boldsymbol{A}$ and $\boldsymbol{A}\boldsymbol{A}^{\top}$. The intersection includes real line edges, while the union adds dotted edges in Figure E.1(d). For intersection, node 1 and node 2 share 1-hop meeting node $\boldsymbol{M}_1$ and a 1-hop diffusion node $\boldsymbol{D}_1$, thus they are pair nodes with $2^{nd}$-order proximity.

- $3^{rd}$-**order proximity** can be represented by the intersection or union of $\boldsymbol{A}^{\top}\boldsymbol{A}^{\top}\boldsymbol{A}\boldsymbol{A}$ and $\boldsymbol{A}\boldsymbol{A}\boldsymbol{A}^{\top}\boldsymbol{A}^{\top}$ as well. In Figure E.1(e), the real line edges and dotted line edges denote intersection and union, respectively. For instance, node 1 and node 3 share a 2-hop meeting node $\boldsymbol{M}_2$ and a 2-hop diffusion node $\boldsymbol{D}_2$. Notably, node 2 and node 3 also share these nodes.

In the $k$-th scale graph, node 1 and node $k$ have $k$-th order proximity as they share a $(k-1)$-hop meeting node $\boldsymbol{M}_k$ and a $(k-1)$-hop diffusion node $\boldsymbol{D}_k$. All nodes with lower-order proximity than

$k$ with node 1 would also have $k$-th order proximity with $k$, as shown in Figure E.1(f), explains why higher scale graphs might get denser.

For fusion, we explore various approaches. Adding all scaled edges to the original graph results in the final augmented graph shown at Figure E.1(b). DiGCN(ib) uses individual GNNs for each scaled graph and simply aggregates their outputs. To fairly compare graph transformation-based inception with DiGCN(ib), we adopt their settings, with the primary difference being the assignment of scaled edge weights.

From Table 4, **1**iG (our model) significantly outperforms DiG on the Telegram, Chameleon, and Squirrel datasets and is on par with DiG in other datasets. **1**iGi2 (our model) shows similar improvements over DiGib. DiG and **1**iG represent the $1^{st}$-scale proximity models, while DiGib and **1**iGi2 correspond to the $2^{nd}$-scale inception models.

We also tested **R**iG(i2), which assigns random weights to scaled edges within the range [0.0001, 10000].

Since the original DiG(ib) model lacks batch normalization, we experimented with both adding it and leaving it out. The results, shown in Table E.1, demonstrate that even **R**iGCN(ib) outperforms DiG(ib). This suggests that the computationally intensive procedure for assigning weights in DiGCN is less effective than simply assigning random weights, as done in RiGCN(ib), which surpasses DiGCN(ib).

We explain in Appendix E.1.1 why the edge weights generated by DiGCN(ib) are inferior to random weights.

Additionally, we replaced the edge weights in SymDiGCN with those in **1**ym, whose performance is comparable to SymDiGCN across all datasets.

In summary, our inception models derived from scale-invariance-based graph transformations outperform or match state-of-the-art models derived from random walks. Our methods simplify and accelerate the process by omitting edge weight calculations, yet yield better results.

### E.1.1 UNDESIRABLE EDGE WEIGHTS BY DiGCN(IB)

To assess the necessity of the computationally expensive edge weights used in DiGCN(ib) (Tong et al., 2020a), we assigned random weights to edges, drawn from a uniform distribution in the range of $[0.0001, 10000]$. This model, named **R**iG, outperformed DiGCN(ib) on the Telegram dataset by a significant margin. The explanation for this performance is as follows:

As shown in Figure E.2, the distribution of edge weights generated by DiGCN(ib) exhibits two peaks—one around 1 and the other around 0—whereas the random weights in **R**iG follow a uniform distribution generated by a random function.

To further investigate this, we generated sets of random weights with intentionally structured peaks, as illustrated in Figure E.3. These sets included two-peak and three-peak distributions. The two-peak distribution resulted in poor performance, with an accuracy of $36.5 \pm 4.0$, while the three-peak distribution showed significantly better performance, achieving $72.6 \pm 4.9$, which is slightly better than DiGCN. This indicates that the specific structure of the edge weight distribution plays a crucial role in model effectiveness.

Overall, the computationally expensive edge weights generated by DiGCN(ib) are not necessarily desirable.

### E.1.2 EXPLAINING THE REASON FOR PERFORMANCE DIFFERENCE OF DiG AND RiG

The performance variation stems from edge weight distributions. DiGCN(ib) produces a bimodal distribution peaked at 0 and 1 (Figure E.2), while **R**iG shows uniform distribution. Experiments with structured random weights (Figure E.3) show two-peak distributions perform poorly (accuracy: $36.5 \pm 4.0$), while three-peak distributions achieve better results ($72.6 \pm 4.9$), exceeding DiGCN. This indicates weight distribution structure significantly impacts model effectiveness.

Table E.2: Ablation study comparing DiG(ib) with two variants: **1**iG(ib) where edge weights are set to 1, and **R**iG(ib) where edge weights are randomly sampled from [0.0001, 10000]. Results on 15 graphs (8 directed, 7 undirected) show that **1**iG(ib) achieves comparable performance to DiG(ib), while **R**iG(ib) occasionally outperforms it (e.g., on Telegram), suggesting that the cost of edge weight computation in DiG(ib) may be unnecessary. Each cell shows accuracy (top). Entries marked OOM indicate Out of Memory on NVIDIA A40 GPUs with 48GB VRAM. The reported execution times are based on various GPUs, including the GeForce GTX 1060, 4070, 2080 Ti, and 3090. Although the hardware is not unified across all experiments, for each dataset, all methods were evaluated on the same GPU to ensure fair comparison. '-' denotes missing data. The last two columns provide dataset statistics, with the '# Node' cell showing total nodes (top) and training nodes (bottom).

| Type | Datasets | DiG | DiGib | 1iG | 1iGib | RiG | RiGib | # Node | # Edge |
|---|---|---|---|---|---|---|---|---|---|
| | **Cornell** | 55.4±7.3 | 69.2±5.4 | 57.0±6.7 | 66.5±7.1 | 44.6±6.9 | 67.8±4.7 | 183 | 298 |
| | Time(s) | 93 | 148 | 87 | 144 | 94 | 174 | | |
| | **Wisconsin** | 64.7±6.8 | 78.0±6.1 | 64.5±5.3 | 74.7±6.6 | 47.3±6.6 | 72.4±4.8 | 251 | 515 |
| | Time(s) | 82 | 118 | 84 | 121 | 77 | 126 | | |
| | **Texas** | 62.2±5.1 | 73.0±8.6 | 67.8±5.8 | 70.5±6.2 | 58.6±6.1 | 67.8±9.2 | 183 | 325 |
| | Time(s) | 88 | 135 | 93 | 149 | 89 | 163 | | |
| **Directed** | **CiteSeer** | 60.4±2.0 | 66.6±1.5 | 66.6±2.2 | 62.8±2.0 | 52.7±2.4 | 65.5±2.0 | 3,312 | 4,715 |
| **Graphs** | Time(s) | 36094 | 127182 | 107 | 161 | 84 | 172 | | |
| | **CoraML** | 77.0±1.9 | 76.6±2.1 | 81.0±1.8 | 81.7±1.3 | 79.7±2.5 | 79.5±2.6 | 2,995 | 8,416 |
| | Time(s) | 33333 | 75735 | 135 | 158 | - | - | | |
| | **PubMed** | 74.3±0.6 | 76.9±0.6 | 76.3±0.9 | 76.7±0.2 | 59.0±1.5 | 59.1±1.4 | 19,717 | 44,327 |
| | Time(s) | 3242 | 3195 | - | - | 1328 | 2001 | | |
| | **WikiCS** | 77.1±1.0 | 78.4±0.6 | 79.1±1.0 | 78.9±0.6 | 73.0±0.5 | 78.6±0.5 | 11,701 | 297,110 |
| | Time(s) | 2074 | OOM | 1348 | 3403 | - | - | | |
| | **Telegram** | 76.8±4.5 | 66.0±5.5 | 95.8±3.5 | 93.0±5.1 | 87.2±3.7 | 89.0±4.1 | 245 | 8,912 |
| | Time(s) | 1294 | 2672 | 142 | 201 | - | - | | |
| | **CiteSeer-U** | 69.2±0.6 | 68.9±0.7 | 69.3±0.6 | 68.8±1.0 | 42.0±1.2 | 40.5±5.5 | 3,327 | 4,732 |
| | Time(s) | 366 | 1017 | 364 | 588 | 237 | 938 | | |
| | **Cora** | 79.1±0.7 | 80.8±0.9 | 80.3±1.0 | 80.0±0.7 | 51.5±1.3 | 50.3±2.6 | 2,708 | 5,429 |
| | Time(s) | 23240 | 34708 | 58 | 98 | - | - | | |
| | **PubMed-U** | OOM | OOM | 78.3±0.2 | 77.5±0.4 | 36.8±5.7 | 77.3±0.6 | 19,717 | 108,365 |
| **Undirected** | Time(s) | OOM | OOM | 1024 | 4096 | 1044 | 2593 | | |
| **Graphs** | **CoA-CS** | 91.1±0.4 | 95.1±0.1 | 89.6±0.5 | 95.0±0.1 | 30.4±0.1 | 88.6±0.4 | 18,333 | 163,788 |
| | Time(s) | OOM | OOM | 843 | 2051 | 866 | 2378 | | |
| | **CoA-Physics** | 95.8±0.2 | 96.8±0.0 | 95.8±0.1 | 96.8±0.0 | 90.9±0.1 | 88.4±0.3 | 34,493 | 495,924 |
| | Time(s) | 14853 | 19108 | 1669 | 7556 | - | - | | |
| | **Photo** | 93.2±0.2 | 91.8±0.3 | 91.8±0.4 | 91.7±0.1 | 28.1±3.3 | 88.4±0.1 | 7,650 | 238,162 |
| | Time(s) | 1178 | 4796 | 946 | 4298 | 646 | 4411 | | |
| | **Computers** | 87.5±0.3 | OOM | 89.5±0.3 | OOM | 83.5±0.6 | OOM | 13,752 | 491,722 |
| | Time(s) | 15042 | OOM | 4612 | OOM | 1603 | OOM | | |

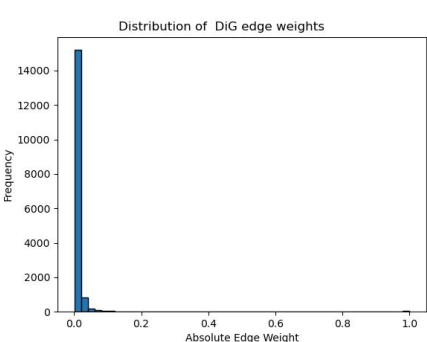
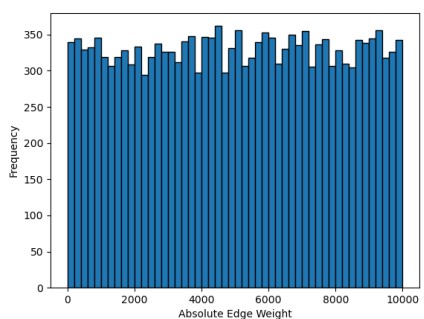

(a) Accuracy with edge weights generated by DiGCN(ib): $67.4 \pm 8.1$

(b) Accuracy with randomly generated edge weights: $85.2 \pm 2.5$

Figure E.2: DiG VS. RiG

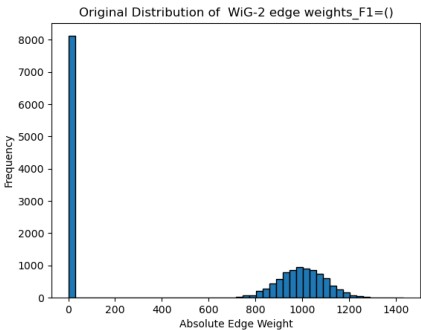
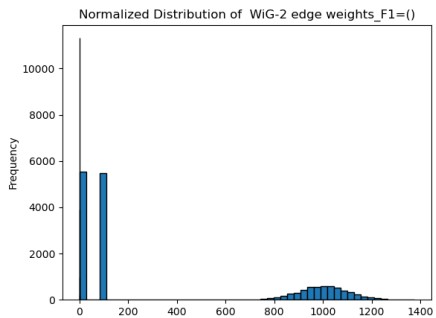

(a) Accuracy with randomly generated edge weights (2 peaks): $36.5 \pm 4.0$

(b) Accuracy with randomly generated edge weights (3 peaks): 72.6±4.9

Figure E.3: Random edge weights

## E.2 HERMITIAN GNNS

### E.2.1 THEORETICAL PROOF

Not only do digraph inception models, but another popular branch of GNNs for directed graphs, Hermitian models, also suffer from unnecessary complexity. We will now prove this.

As MagNet (Zhang et al., 2021) pioneered the introduction of Hermitian matrices in graph learning, we focus our analysis on this foundational work. In this section, we prove that MagNet is mathematically equivalent to GraphSAGE (Hamilton et al., 2017) with GCN's normalization. This equivalence reveals that the apparent complexity of Hermitian-based approaches may not offer fundamental advantages over simpler methods.

For directed graphs, the adjacency matrix $\boldsymbol{A}$ differs from its transpose $\boldsymbol{A}^\top$. MagNet (Zhang et al., 2021) uses $\boldsymbol{A}_s = (\boldsymbol{A} + \boldsymbol{A}^\top)/2$ to obtain a symmetric adjacency matrix, which is then incidence-normalized to $\tilde{\boldsymbol{A}}_s$. Using a phase angle to distribute weight between real and imaginary parts, $\boldsymbol{\Theta} = e^{2\pi q j(\boldsymbol{A} - \boldsymbol{A}^\top)}$, the adjacency matrix becomes:

$$\hat{\boldsymbol{A}}_s = \tilde{\boldsymbol{A}}_s \odot \boldsymbol{\Theta} = \tilde{\boldsymbol{A}}_s \odot e^{2\pi q j(\boldsymbol{A} - \boldsymbol{A}^\top)}$$
$$= \tilde{\boldsymbol{A}}_s \odot \cos(2\pi q(\boldsymbol{A} - \boldsymbol{A}^\top)) + j\tilde{\boldsymbol{A}}_s \odot \sin(2\pi q(\boldsymbol{A} - \boldsymbol{A}^\top))$$
$$= \hat{\boldsymbol{A}}_s(real) + j\hat{\boldsymbol{A}}_s(imag),$$

where $\odot$ denote component-wise multiplication, and the values of $\hat{\boldsymbol{A}}_s(real)$ and $\hat{\boldsymbol{A}}_s(imag)$ in various cases of edge directions are enumerated in Table E.3. As shown in Table E.3, $\hat{\boldsymbol{A}}_s(real)$ and $\hat{\boldsymbol{A}}_s(imag)$ are either $\cos\alpha$, $\sin\alpha$, 0 or 1.

As a result, $\hat{\boldsymbol{A}}_s(real)$ is a real symmetric matrix $\overline{\boldsymbol{A}_s}$ similar to $\tilde{\boldsymbol{A}}_s$, except that for unidirectional edges, the value is scaled by $\cos\alpha$ relative to the corresponding value in $\tilde{\boldsymbol{A}}_s$. In contrast, $\hat{\boldsymbol{A}}_s(imag)$ is a skew-symmetric matrix:

$$\hat{\boldsymbol{A}}_s(imag) = 0.5\sin\alpha \boldsymbol{D}^{-\frac{1}{2}}(\boldsymbol{A} - \boldsymbol{A}^\top)\boldsymbol{D}^{-\frac{1}{2}}, \tag{A.8}$$

where $\boldsymbol{D}$ is the degree matrix of the graph.

Table E.3: Case enumeration of the elements in adjacency matrices for different edge types. Here, $\boldsymbol{A}_s = 0.5(\boldsymbol{A} + \boldsymbol{A}^\top)$ is the symmetrized adjacency matrix, $\tilde{\boldsymbol{A}}_s$ is the incidence-normalized adjacency matrix, and $\hat{\boldsymbol{A}}_s$ is the complex-valued adjacency matrix for MagNet with parameter $\alpha = 2\pi q$. The variable $d$ denotes the node degree.

| | **MagNet**($\alpha = 2\pi q$) | | | | |
|---|---|---|---|---|---|
| **Edges** | $\boldsymbol{A}_s$ | $\tilde{\boldsymbol{A}}_s$ | $\hat{\boldsymbol{A}}_s$ | $\hat{\boldsymbol{A}}_s(real)$ | $\hat{\boldsymbol{A}}_s(imag)$ |
| $m \to n$ | 0.5 | $\frac{0.5}{d}$ | $\frac{0.5}{d}e^{2\pi q\cdot j}$ | $\frac{0.5}{d}\cos\alpha$ | $\frac{0.5}{d}\sin\alpha$ |
| $n \to m$ | 0.5 | $\frac{0.5}{d}$ | $\frac{0.5}{d}e^{-2\pi q\cdot j}$ | $\frac{0.5}{d}\cos\alpha$ | $-\frac{0.5}{d}\sin\alpha$ |
| $m \leftrightarrow n$ | 1 | $d^{-1}$ | $d^{-1}$ | $d^{-1}$ | 0 |
| $m \not\leftrightarrow n$ | 0 | 0 | 0 | 0 | 0 |

In the end, they concatenate real and imaginary parts. Thus, the total output would be:

$$Out(total) = \boldsymbol{Z}(real)\boldsymbol{W}_1 + \boldsymbol{Z}(imag)\boldsymbol{W}_2$$

Specifically, they define the node feature as $\widehat{\boldsymbol{X}} = \boldsymbol{X} + j\boldsymbol{X}$. When trying to follow ChebNet (Defferrard et al., 2016), MagNet made a coding mistake in obtaining the Laplacian. They redundantly subtracted $\boldsymbol{I}$ for $\boldsymbol{L} = \boldsymbol{I} - \hat{\boldsymbol{A}}_s$. Thus their $\hat{\boldsymbol{L}} = \boldsymbol{I} - \hat{\boldsymbol{A}}_s - \boldsymbol{I} = -\hat{\boldsymbol{A}}_s$, and they mistakenly get:

$$\hat{\boldsymbol{T}}_{k+2} = 2\hat{\boldsymbol{L}}T_{k+1} - \boldsymbol{T}_k$$

where $\boldsymbol{T}_1$ is $\boldsymbol{I}$ while their $\boldsymbol{T}_2$ is mistakenly $-\tilde{\boldsymbol{A}}_s$:

$$\hat{\boldsymbol{T}}_2 = -\hat{\boldsymbol{A}}_s$$

- When $K = 1$, the Hermitian output is:

$$\boldsymbol{Z}_1 = \widehat{\boldsymbol{T}}_1 \boldsymbol{X} \boldsymbol{W}_1 = (\boldsymbol{I} + j0)(\boldsymbol{X} + j\boldsymbol{X})\boldsymbol{W}_1 = (\boldsymbol{X} + j\boldsymbol{X})\boldsymbol{W}_1 \tag{A.9}$$

- When $K = 2$, with two weight matrices $\boldsymbol{W}_1$ and $\boldsymbol{W}_2$, the Hermitian output is:

$$\begin{aligned}
\boldsymbol{Z}_2 &= \boldsymbol{Z}_1 + (\widehat{\boldsymbol{T}}_2(real) + j\widehat{\boldsymbol{T}}_2(imag))(\boldsymbol{X} + j\boldsymbol{X})\boldsymbol{W}_2 \\
&= \boldsymbol{Z}_1 + (-\hat{\boldsymbol{A}}_s(real) - j\hat{\boldsymbol{A}}_s(imag))(\boldsymbol{X} + j\boldsymbol{X})\boldsymbol{W}_2 \\
&= \boldsymbol{X}\boldsymbol{W}_1 + \hat{\boldsymbol{A}}_s(imag)\boldsymbol{X}\boldsymbol{W}_2 - \hat{\boldsymbol{A}}_s(real)\boldsymbol{X}\boldsymbol{W}_2 \\
&\quad + j\big(\boldsymbol{X}\boldsymbol{W}_1 - \hat{\boldsymbol{A}}_s(real)\boldsymbol{X}\boldsymbol{W}_2 - \hat{\boldsymbol{A}}_s(imag)\boldsymbol{X}\boldsymbol{W}_2\big)
\end{aligned} \tag{A.10}$$

Given that $\hat{\boldsymbol{T}}_k$ is linear combinations of $\hat{\boldsymbol{A}}_s$ up to the $k$-th power, and concatenating real and imaginary parts of the Hermitian output, the total output for different values of $k$ becomes:

- For $k = 1$: Based on Equation A.9, we get the total output:

$$Out_1(total) = \boldsymbol{Z}_1(real)\boldsymbol{W}_1 + \boldsymbol{Z}_1(imag)\boldsymbol{W}_2 = \boldsymbol{X}\boldsymbol{W}_1 + \boldsymbol{X}\boldsymbol{W}_2 \tag{A.11}$$

- For $k = 2$: Based on Equation A.10, we get:

$$\begin{aligned}
Out_2(total) &= \boldsymbol{Z}_2(real)\boldsymbol{W}_3 + \boldsymbol{Z}_2(imag)\boldsymbol{W}_4 \\
&= \big(\boldsymbol{X}\boldsymbol{W}_1 + \hat{\boldsymbol{A}}_s(imag)\boldsymbol{X}\boldsymbol{W}_2 - \hat{\boldsymbol{A}}_s(real)\boldsymbol{X}\boldsymbol{W}_2\big)\boldsymbol{W}_3 + \\
&\quad \big(\boldsymbol{X}\boldsymbol{W}_1 - \hat{\boldsymbol{A}}_s(real)\boldsymbol{X}\boldsymbol{W}_2 - \hat{\boldsymbol{A}}_s(imag)\boldsymbol{X}\boldsymbol{W}_2\big)\boldsymbol{W}_4 \\
&= \boldsymbol{X}\widehat{\boldsymbol{W}}_1 + \hat{\boldsymbol{A}}_s(real)\boldsymbol{X}\widehat{\boldsymbol{W}}_2 + \hat{\boldsymbol{A}}_s(imag)\boldsymbol{X}\widehat{\boldsymbol{W}}_3
\end{aligned} \tag{A.12}$$

According to Equation A.8, Equation A.12 can further be simplified to :

$$Out_2(total) = \boldsymbol{X}\boldsymbol{W}_1 + \boldsymbol{D}^{-\frac{1}{2}}\overline{\boldsymbol{A}_s}\boldsymbol{D}^{-\frac{1}{2}}\boldsymbol{X}\boldsymbol{W}_2 + 0.5\sin\alpha\,\boldsymbol{D}^{-\frac{1}{2}}(\boldsymbol{A} - \boldsymbol{A}^\top)\boldsymbol{D}^{-\frac{1}{2}}\boldsymbol{X}\boldsymbol{W}_3$$

$$\tag{A.13}$$

As layer-wise propagation of GraphSAGE is $\boldsymbol{X}\boldsymbol{W}_1 + \boldsymbol{D}^{-1}\boldsymbol{A}\boldsymbol{X}\boldsymbol{W}_2$, the first two terms of Equation A.13 are equivalent to GraphSAGE but with GCN's normalization in place of the original row normalization scheme. The third term of Equation A.13 computes the difference between the sum of features from in-neighbors and out-neighbors. This information provides little help in node classification as MagNet performs poorly on heterophilic graphs (Rossi et al., 2024), thus $\sin\alpha$ should be 0 or close to 0 for better performance.

This analysis explains why MagNet and GraphSAGE perform similarly across most datasets (Zhang et al., 2021), differing primarily in their normalization schemes: MagNet uses GCN's normalization while GraphSAGE employs row normalization through mean aggregation (Luo et al., 2024b). Among all datasets tested in the original MagNet paper, this distinction only leads to MagNet outperforming GraphSAGE in the Telegram dataset, where degree information plays a crucial role in prediction. In Table E.4, it achieves 89.4 accuracy using only node degrees with MLP, while Dir-GNN (Rossi et al., 2024) , with row normalization and no features, achieves only 38 accuracy, indicating a complete failure to learn. Row normalization diminishes this vital degree information (Jiang et al., 2025), explaining MagNet's superior performance on Telegram over GraphSAGE.

Without coding mistakes or using these tricks, the performance of MagNet would be the same as ChebNet, which is worse than GraphSAGE in most cases.

For a more comprehensive review, please refer to Sections D.1.1 and D.4.

Table E.4: Classification accuracy (%) of Dir-GNN Rossi et al. (2024) with different feature configurations and normalization schemes on Telegram datasets. Feature configurations include: original node features from datasets (Origin Feature), constant features (No Feature, all set to 1), and node degree variants (in-degree, out-degree, or both). **Bold** value indicates learning failure with row normalization and no features. Underline value indicates that predictions can be made accurately using only node degrees.

| Telegram | Feature | MLP | None | Row | Sym | Dir |
|---|---|---|---|---|---|---|
| Origin | 1 | 38.0±7.2 | 95.6±2.8 | 74.2±5.5 | 93.0±4.1 | 92.8±4.7 |
| No Feature | 1 | 38.0±0.0 | 95.4±4.0 | **38.0±0.0** | 93.0±4.7 | 93.0±3.0 |
| In-degree | 1 | 64.0±5.7 | 95.8±3.8 | 80.8±4.1 | 92.6±3.3 | 94.4±2.1 |
| Out-degree | 1 | 63.0±5.7 | 96.4±2.5 | 78.4±5.9 | 90.6±7.4 | 93.6±4.5 |
| Both degrees | 2 | 89.4±5.8 | 95.0±3.4 | 80.0±4.3 | 93.4±2.1 | 94.4±4.3 |

Table E.5: Comparison of matrix multiplication dimensions between **ScaleNet** and **LargeScaleNet**. Both models follow a three-step process to compute aggregated features. Each cell shows the matrix sizes; darker colors mean more computationally intensive steps. **ScaleNet** performs an expensive matrix–matrix multiplication of two $[2.9M \times 2.9M]$ matrices during preprocessing, while **LargeScaleNet** avoids this step using cheaper multiplications.

| Model | Step 1 | Step 2 | Step 3 |
|---|---|---|---|
| **ScaleNet** | $\boldsymbol{AA}$ computation $[2.9M \times 2.9M] \times [2.9M \times 2.9M]$ | $\widetilde{\boldsymbol{AA}}$ normalization $[2.9M \times 2.9M] \cdot [2.9M \times 1]$ | $\widetilde{\boldsymbol{AA}}X$ aggregation $[2.9M \times 2.9M] \times [2.9M \times 269]$ |
| **Large-ScaleNet** | $\tilde{\boldsymbol{A}}$ normalization $[2.9M \times 2.9M] \cdot [2.9M \times 1]$ | $\tilde{\boldsymbol{A}}X$ aggregation $[2.9M \times 2.9M] \times [2.9M \times 269]$ | $\tilde{\boldsymbol{A}}(\tilde{\boldsymbol{A}}X)$ aggregation $[2.9M \times 2.9M] \times [2.9M \times 269]$ |

### E.3 FABERNET

Both ScaleNet and FaberNet rely on multi-scale learning, but FaberNet can work on large graphs. Thus, we develop LargeScaleNet inspired by FaberNet, as shown in Table E.5. A detailed comparison of scaled graph components between FaberNet (Koke & Cremers, 2024) and LargeScaleNet is presented in Table E.6. We further conduct a Wilcoxon signed-rank test to compare the top two models on large graphs. Each model is run 30 times; if the original number of splits is fewer than 30, we repeat experiments with different random seeds to reach 30 runs. As shown in Table E.7, LargeScaleNet is significantly better than FaberNet in performance.

Beyond these, there are other GNNs for directed graphs, including newer spectral methods (Yang et al., 2022; Huang et al., 2024) and graph transformer models (Bo et al., 2023; Veličković et al., 2018; Shirzad et al., 2023; Rosenbluth et al.). However, as shown in Appendix E.2, spectral methods like MagNet effectively reduce to GraphSAGE when incidence normalization is applied. Moreover, the HoloNet implementation contains critical flaws in its Complex-Faber variant, and its FaberConv can also be explained from the message passing perspective.

Table E.6: Computation formulas at different scales. FaberNet and LargeScaleNet are identical when $k = 1$; when $k = 2$, LargeScaleNet is comprehensive for all four types of $2^{nd}$-order connections, while FaberNet only has two. For $k = 3$, FaberNet still has only two types of connections, whereas LargeScaleNet currently has four and can be expanded to cover all $3^{rd}$-order edge types.

| $k$ | FaberNet | LargeScaleNet |
|---|---|---|
| 1 | $\alpha\boldsymbol{AX} + (1-\alpha)\boldsymbol{A}^\top\boldsymbol{X}$ | $\alpha\boldsymbol{AX} + (1-\alpha)\boldsymbol{A}^\top\boldsymbol{X}$ |
| 2 | $\frac{\alpha\boldsymbol{AAX}}{2} + \frac{(1-\alpha)\boldsymbol{A}^\top\boldsymbol{AX}}{2}$ | $\beta\boldsymbol{AA}^\top\boldsymbol{X} + (1-\beta)\boldsymbol{A}^\top\boldsymbol{AX} + \gamma\boldsymbol{AAX} + (1-\gamma)\boldsymbol{A}^\top\boldsymbol{A}^\top\boldsymbol{X}$ |
| 3 | $\frac{\alpha\boldsymbol{AAAX}}{4} + \frac{(1-\alpha)\boldsymbol{A}^\top\boldsymbol{AAX}}{4}$ | $\beta\boldsymbol{AAA}^\top\boldsymbol{X} + (1-\beta)\boldsymbol{A}^\top\boldsymbol{A}^\top\boldsymbol{AX} + \gamma\boldsymbol{AAAX} + (1-\gamma)\boldsymbol{A}^\top\boldsymbol{A}^\top\boldsymbol{A}^\top\boldsymbol{X}$ |

Table E.7: Results of the Wilcoxon signed-rank test for the top two models on 30 splits. p-values less than 0.05 (shown in bold) indicate significant difference; otherwise, models are considered equally effective.

|  | Arxiv-year | Snap-patents | Roman-Empire |
|---|---|---|---|
| Best Model | LargeScaleNet **65.49±0.57** | LargeScaleNet **74.98±0.12** | LargeScaleNet **93.53±0.30** |
| Runner-up | FaberNet 64.39±0.67 | FaberNet 74.55±0.11 | FaberNet 92.32±0.30 |
| p-value (Statistic) | **2.61e-08** (6.0) | **1.86e-09** (0.0) | **1.86e-09** (0.0) |

Moreover, as we demonstrate in Appendix E.1, randomly assigning edge weights in the range [0.0001,10000] performs comparably to computationally expensive learned weighting schemes. Our findings suggest that in GNNs, the mere existence of an edge is often far more influential than the precise value of its non-zero weight. While graph transformers have some merits in exploring additional connections, their focus on learning exact edge weights introduces unnecessary complexity that offers limited practical benefit in most real-world tasks (Luo et al., 2024a). In fact, GNNs primarily learn node feature transformations through linear layers (e.g., multiplying features by a weight matrix $W$) combined with aggregating neighborhood information. Consequently, explicitly learning precise edge weights is often redundant and offers limited additional benefit in many applications.

Still Graph Transformer is very valuable in exploring all possible connections. If we are going to do Graph Transformer, we would first try not to learn the specific weights of each connection, but a binary value edge weight: its exsitence, 0 or 1. In addition, structural features such as Positional encoding concatenated to original node feature might be helpful too.

Based on these insights, this paper deliberately focuses on simpler and more essential architectures—specifically, message passing neural networks (MPNNs)—and does not further explore spectral methods or graph transformer models.

# F  SELF-LOOPS

In this section, we first discuss the influence of manually adding self-loops, then we present that self-loops can be generated in higher order proximity matrix.

## F.1  INFLUENCE OF ADDING SELF-LOOPS

Adding self-loops is traditionally understood as assigning the highest weight to a node itself. However, in this paper, we highlight that adding self-loops also facilitates the incorporation of multi-scale information into the graph representation.

Given an adjacency matrix $A$, the modified adjacency matrix $\hat{A}$ with self-loops is defined as:

$$\hat{A} = A + I,$$

where $I$ is the identity matrix. The resulting products involving $\hat{A}$ are as follows:

- $\hat{A}\hat{A}^\top$:
$$\hat{A}\hat{A}^\top = (A + I)(A^\top + I) = AA^\top + A + A^\top + I$$

- $\hat{A}^\top\hat{A}$:
$$\hat{A}^\top\hat{A} = (A^\top + I)(A + I) = A^\top A + A + A^\top + I$$

- $\hat{A}\hat{A}$:
$$\hat{A}\hat{A} = (A + I)(A + I) = AA + 2A + I$$

- $\hat{A}^\top\hat{A}^\top$:
$$\hat{A}^\top\hat{A}^\top = (A^\top + I)(A^\top + I) = A^\top A^\top + 2A^\top + I$$

The influence of adding self-loops can be analyzed at two levels:

1. **Layer-wise Influence** Adding self-loops to $\boldsymbol{A}$ results in an augmented adjacency matrix, $\boldsymbol{A} + \boldsymbol{I}$, that incorporates all the original directed edges and self-loops. As a result, the propagation of a 1-layer GCN using $\boldsymbol{AA}$ as the adjacency matrix will, after adding self-loops, becomes $(\boldsymbol{A}+\boldsymbol{I})(\boldsymbol{A}+\boldsymbol{I})$, include not only the information from $\boldsymbol{AA}$ but also direct contributions from $\boldsymbol{A}$ and the self-loops in its output.

   Since heterophilic graphs connect different types of nodes, adding self-loops dilutes this important difference-based structure. This can reduce model performance in tasks requiring heterophilic relationship learning.

2. **Multi-layer Influence**

   For a 2-layer GCN, omitting the non-linear activation for notational simplicity (while the actual model includes non-linearity), the propagation outputs are[1]:

   - **Without self-loops:**

   $$\tilde{\boldsymbol{A}}(\tilde{\boldsymbol{A}}\boldsymbol{X}\boldsymbol{W}_1)\boldsymbol{W}_2 \approx \tilde{\boldsymbol{A}}^2 \boldsymbol{X}\boldsymbol{W} = \boldsymbol{D}^{-\frac{1}{2}}\boldsymbol{A}^2\boldsymbol{D}^{-\frac{1}{2}}\boldsymbol{X}\boldsymbol{W}$$

   - **With self-loops:**

   $$\widetilde{(\boldsymbol{A}+\boldsymbol{I})}\big((\widetilde{\boldsymbol{A}+\boldsymbol{I}})\boldsymbol{X}\boldsymbol{W}_1\boldsymbol{W}_2\big) \approx \widetilde{(\boldsymbol{A}+\boldsymbol{I})}^2 \boldsymbol{X}\boldsymbol{W} = \boldsymbol{D}^{-\frac{1}{2}}(\boldsymbol{A}\boldsymbol{A} + 2\boldsymbol{A} + \boldsymbol{I})\boldsymbol{D}^{-\frac{1}{2}}\boldsymbol{X}\boldsymbol{W}$$

   Consequently, for a $k$-layer GCN, adding self-loops enables the network to aggregate information from $k$-hop neighbors, $(k-1)$-hop neighbors, ..., 1-hop neighbors, and the node itself. In contrast, a $k$-layer GCN without self-loops aggregates information strictly from $k$-hop neighbors.

Adding self-loops enhances the propagation mechanism, facilitating the incorporation of multi-scale information. This can boost performance on homophilic datasets, where nodes with similar features are more likely to be connected. However, on heterophilic datasets, where nodes with dissimilar features tend to be connected, adding self-loops may degrade performance. Therefore, it is essential to treat the inclusion of self-loops as a dataset-specific strategy, optimizing it based on the characteristics of the data to achieve the best performance when building our models.

### F.2 GENERATED SELF-LOOPS

#### F.2.1 PROOF OF GENERATED SELF-LOOPS IN HIGHER ORDER PROXIMITY MATRIX

According to DiGCN(ib) (Tong et al., 2020a), the $k^{th}$-order proximity between node $i$ and node $j$ is both or either $\boldsymbol{M}_{i,j}^{(k)}$ and $\boldsymbol{D}_{i,j}^{(k)}$ are non-zero for:

$$\boldsymbol{M}^{(k)} = (\underbrace{\boldsymbol{A}\cdots\boldsymbol{A}}_{k-1\text{ times}})(\underbrace{\boldsymbol{A}^{\top}\cdots\boldsymbol{A}^{\top}}_{k-1\text{ times}})$$

$$\boldsymbol{D}^{(k)} = (\underbrace{\boldsymbol{A}^{\top}\cdots\boldsymbol{A}^{\top}}_{k-1\text{ times}})(\underbrace{\boldsymbol{A}\cdots\boldsymbol{A}}_{k-1\text{ times}})$$

**Proposition F.1.** *The diagonal entries would be non-zero in $\boldsymbol{M}^{(k)}$ and $\boldsymbol{D}^{(k)}$ ($k \geq 2$) if this node has in-edge or out-edge in $\boldsymbol{A}$. We call this generated self-loops.*

*Proof.* Suppose node $a$ has an out-edge $a \to b$. This implies that there exists a path $a \to b \leftarrow a$ due to the edges $a \to b$ and $b \leftarrow a$. As a result, the entry $\boldsymbol{M}_{a,a}^{(2)}$ in the matrix $\boldsymbol{M}^{(k)}$ will be 1, indicating a non-zero diagonal entry.

Similarly, suppose node $a$ has an in-edge $a \leftarrow c$. This implies that there is a path $c \to a$ due to the edge $a \leftarrow c$, resulting in the entry $\boldsymbol{D}_{a,a}^{(2)}$ in the matrix $\boldsymbol{D}^{(k)}$ being 1, again indicating a non-zero diagonal entry.

For all $k \geq 2$, $\boldsymbol{M}_{a,a}^{(k)}$ will be non-zero, as a self-loop can be traversed any number of steps and still return to the same node. The same applies to $\boldsymbol{D}_{a,a}^{(k)}$. $\qquad\square$

---

[1] Here, $\approx$ reflects the approximation aligned with the Universal Approximation Theory (UAT).

Suppose $M_{a,a}^{(k)}$ is non-zero, indicating that node $a$ has a self-loop in the $k$-th order proximity matrix. If nodes $d$ and $e$ have a $k$-th order path meeting at node $a$, then $M_{d,e}^{(k+n)}$ (for $n \geq 1$) will remain non-zero. This implies that higher-order matrices $M^{(k)}$ become denser as the order increases.

If we remove the self-loops from $M^{(k)}$, this densification effect can be eliminated. The same applies to $D^{(k)}$.

### F.2.2 EXAMPLE

We will use an example to explain the above observation. A simple graph is shown below:

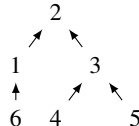

Let's define the adjacency matrix $A$ where non-zero entries $A_{i,j}$ represent directed edges from node $i$ to node $j$. Given the directed edges (1,2), (3,2), (4,3), (5,3), (6,1), the adjacency matrix $A$ is as follows:

$$A = \begin{bmatrix} 0 & 1 & 0 & 0 & 0 & 0 \\ 0 & 0 & 1 & 0 & 0 & 0 \\ 0 & 0 & 0 & 0 & 0 & 0 \\ 0 & 0 & 1 & 0 & 0 & 0 \\ 0 & 0 & 1 & 0 & 0 & 0 \\ 0 & 1 & 0 & 0 & 0 & 0 \end{bmatrix}$$

The transpose of $A$, denoted $A^\top$, represents the reversed edges of $A$. Therefore, the adjacency matrix $A^\top$ is as follows:

$$A^\top = \begin{bmatrix} 0 & 0 & 0 & 0 & 0 & 0 \\ 1 & 0 & 0 & 0 & 0 & 1 \\ 0 & 1 & 0 & 1 & 1 & 0 \\ 0 & 0 & 0 & 0 & 0 & 0 \\ 0 & 0 & 0 & 0 & 0 & 0 \\ 0 & 0 & 0 & 0 & 0 & 0 \end{bmatrix}$$

Matrix $AA^\top$, which is also $M^2$, represents nodes connected with scaled edge $\rightarrow\leftarrow$ [2], which are (1,3), (4,5). beacause of paths

- $1 \rightarrow 2 \leftarrow 3$
- $4 \rightarrow 3 \leftarrow 5$

Selfloops are generated because of paths

- $1 \rightarrow 2 \leftarrow 1$
- $3 \rightarrow 2 \leftarrow 3$
- $4 \rightarrow 3 \leftarrow 4$
- $5 \rightarrow 3 \leftarrow 5$

Below are the matrices $M^2$ and $\hat{M}^2$ ($M^2$ with self-loops removed):

$$
\overset{\textstyle M^2}{\begin{bmatrix} 1 & 0 & 1 & 0 & 0 & 0 \\ 0 & 0 & 0 & 0 & 0 & 0 \\ 1 & 0 & 1 & 0 & 0 & 0 \\ 0 & 0 & 0 & 1 & 1 & 0 \\ 0 & 0 & 0 & 1 & 1 & 0 \\ 0 & 0 & 0 & 0 & 0 & 1 \end{bmatrix}}
\overset{\textstyle \hat{M}^2}{\begin{bmatrix} 0 & 0 & 1 & 0 & 0 & 0 \\ 0 & 0 & 0 & 0 & 0 & 0 \\ 1 & 0 & 0 & 0 & 0 & 0 \\ 0 & 0 & 0 & 0 & 1 & 0 \\ 0 & 0 & 0 & 1 & 0 & 0 \\ 0 & 0 & 0 & 0 & 0 & 0 \end{bmatrix}}
$$

---

[2] as the backward of $\rightarrow\leftarrow$ is still $\rightarrow\leftarrow$, this scaled edge is bidirected, getting symmetric matrix.

Matrix $\boldsymbol{A}\boldsymbol{A}(\boldsymbol{A}^\top\boldsymbol{A}^\top)$ represents undirected scaled edge $\rightarrow\rightarrow\leftarrow\leftarrow$[3], which are (4,6), (5,6), because of paths:

- $4 \rightarrow 3 \rightarrow 2 \leftarrow 2 \leftarrow 6$
- $5 \rightarrow 3 \rightarrow 3 \leftarrow 1 \leftarrow 6$

One couple of $2^{nd} - order$ proximity nodes was introduced here because of self-loop in $\boldsymbol{A}\boldsymbol{A}^\top$, making paths:

- $4 \rightarrow 3 \rightarrow 2 \leftarrow 3 \leftarrow 5$

One Self-loop is generated because of path:

- $6 \rightarrow 1 \rightarrow 2 \leftarrow 1 \leftarrow 6$

Below are the matrices $\boldsymbol{M}^3$ and $\hat{\boldsymbol{M}}^3$(generated from $\hat{\boldsymbol{M}}^2$):

$$
\boldsymbol{M}^3 \qquad\qquad \hat{\boldsymbol{M}}^3
$$

$$
\begin{bmatrix}
0 & 0 & 0 & 0 & 0 & 0 \\
0 & 0 & 0 & 0 & 0 & 0 \\
0 & 0 & 0 & 0 & 0 & 0 \\
0 & 0 & 0 & 1 & 1 & 1 \\
0 & 0 & 0 & 1 & 1 & 1 \\
0 & 0 & 0 & 1 & 1 & 1
\end{bmatrix}
\begin{bmatrix}
0 & 0 & 0 & 0 & 0 & 0 \\
0 & 0 & 0 & 0 & 0 & 0 \\
0 & 0 & 0 & 0 & 0 & 0 \\
0 & 0 & 0 & 0 & 0 & 1 \\
0 & 0 & 0 & 0 & 0 & 1 \\
0 & 0 & 0 & 1 & 1 & 0
\end{bmatrix}
$$

As demonstrated, both manually added self-loops to the original adjacency matrix and self-loops generated during multiplications of previous matrix contribute to the densification of $\boldsymbol{M}^k$ (or $\boldsymbol{D}^k$). This densification occurs because these self-loops introduce lower-order proximity edges into the higher-order proximity matrix. However, to obtain a pure $k^{\text{th}}$-order proximity matrix, where non-zero entries exclusively represent $k^{\text{th}}$-order proximities, it is crucial to remove the self-loops from each $\boldsymbol{M}^k$ and $\boldsymbol{D}^k$ before using it to compute the $(k+1)^{\text{th}}$-order proximity.

## G  RUNTIME AND MEMORY COMPARISON

### G.1  THEORETICAL COMPARISON IN MODEL COMPLEXITY

We provide here a comparison of the dominant computational components of each related model.

For a standard GCN layer, the main operations are:

- $\boldsymbol{A}\boldsymbol{X}$: neighborhood aggregation
- $\boldsymbol{X}\boldsymbol{W}$: linear transformation
- $\boldsymbol{N} \cdot \boldsymbol{A}$: normalization of the adjacency

In addition to these, ScaleNet includes an extra adjacency transformation $\boldsymbol{A}\boldsymbol{A}$, which produces a $2^{nd}$-scale adjacency before aggregation.

GCN aggregates only through $\boldsymbol{A}$, while Dir-GNN aggregates through both $\boldsymbol{A}$ and $\boldsymbol{A}^\top$; as shown in Table G.1, this doubles the aggregation cost relative to GCN. Built on Dir-GNN, ScaleNet introduces four $2^{nd}$-scale adjacencies: $\boldsymbol{A}\boldsymbol{A}$, $\boldsymbol{A}^\top\boldsymbol{A}$, $\boldsymbol{A}\boldsymbol{A}^\top$, and $\boldsymbol{A}^\top\boldsymbol{A}^\top$. Thus, ScaleNet performs 4 computations whose complexity matches that of $\boldsymbol{A}\boldsymbol{A}$-type products. LargeScaleNet aggregates via $\boldsymbol{A}$ and $\boldsymbol{A}^\top$ for the $1^{st}$-scale, and for each of the four $2^{nd}$-scale components performs two aggregations, resulting in a total of 10 aggregations with complexity matching that of $\boldsymbol{A}\boldsymbol{X}$.

Table G.1 summarizes the relative number of core operations across models. From this breakdown, we observe that, in terms of theoretical model complexity, LargeScaleNet requires fewer than ten times the computations of GCN.

---

[3]as its backward version is itself, it is bidirected

Table G.1: Theoretical complexity comparison of models. Computations are ordered from left to right by approximate computational cost (heavy to light).

| Model | $AA$ | $AX$ | $XW$ | $N \cdot A$ |
|---|---|---|---|---|
| GCN | 0 | 1 | 1 | 1 |
| Dir-GNN | 0 | 2 | 2 | 2 |
| FaberNet | 0 | 6 | 6 | 4 |
| ScaleNet | 4 | 6 | 6 | 6 |
| LargeScaleNet | 0 | 10 | 10 | 6 |

## G.2 EXPERIMENTAL COMPARISON IN RUNTIME AND MEMORY USAGE

We evaluate all datasets in Table 2 and Table 4 using an NVIDIA A40 GPU. For each model, we report (i) peak GPU memory during the first epoch and (ii) runtime per epoch averaged over epochs 2–11, which excludes one-time setup overhead. All models use 2 layers. Because CPU usage is small and similar across all methods, we compare only GPU usage.

Since the number of hidden features affects different models in different ways—and thus influences both memory and runtime—we test two hidden sizes per model. For small and medium graphs, we use hidden dimensions of 512 and 16 (Figure G.6 to Figure G.7). For large graphs, we use hidden dimensions of 1 and 16 (Figure G.8 to Figure G.9). For simplicity, we abbreviate ChebNet as "Cheb" in the figures, and LargeScaleNet is abbreviated as LScaleNet in some of the figures.

The original implementations of GCN, GraphSAGE, and APPNP use dense tensors for edge inputs, which is memory intensive. For fairness, we additionally evaluate versions that use SparseTensor edge representations, denoted GCN-sp, SAGE-sp, and APPNP-sp.

To summarize the complexity of LargeScaleNet and its scalability relative to ScaleNet, we list their resource usage across all datasets in Table G.2. The results show that as the graph size increases, the gap between LargeScaleNet and GCN widens, reaching approximately five times the GPU memory of GCN on the largest tested graph (Snap-Patents), which is consistent with our theoretical upper bound of ten times.

| | **GPU Peak Memory (MB)** | | | | | | | | |
|---|---|---|---|---|---|---|---|---|---|
| Model | Tel | Cham | Cora | CiteSeer | Sqrl | WCS | Roman | ArX | Patents |
| GCN-sp | 65.36 | 90.44 | 99.43 | 113.08 | 138.72 | 123.86 | 103.34 | 348.71 | 6021.45 |
| LargeScaleNet | 66.45 | 215.87 | 303.80 | 401.78 | 394.79 | 223.37 | 265.60 | 887.60 | 29538.33 |
| ScaleNet | 72.41 | 351.50 | 316.88 | 405.71 | 3737.55 | 2832.50 | 334.15 | OOM | OOM |

Table G.2: GPU peak memory (MB) across datasets. Abbreviations: Sqrl = Squirrel, Cham = Chameleon, WCS = WikiCS, ArX = ArXiv-Year. OOM indicates out-of-memory on the NVIDIA A40 GPU. As graph size increases, ScaleNet runs out of memory, while LargeScaleNet continues to run but consumes substantially more memory than GCN-approximately five times the memory of GCN on the Snap-Patents dataset. All models are 2-layer with a hidden dimension of 16. GCN-sp indicates that the GCN uses a SparseTensor for its edge representation.

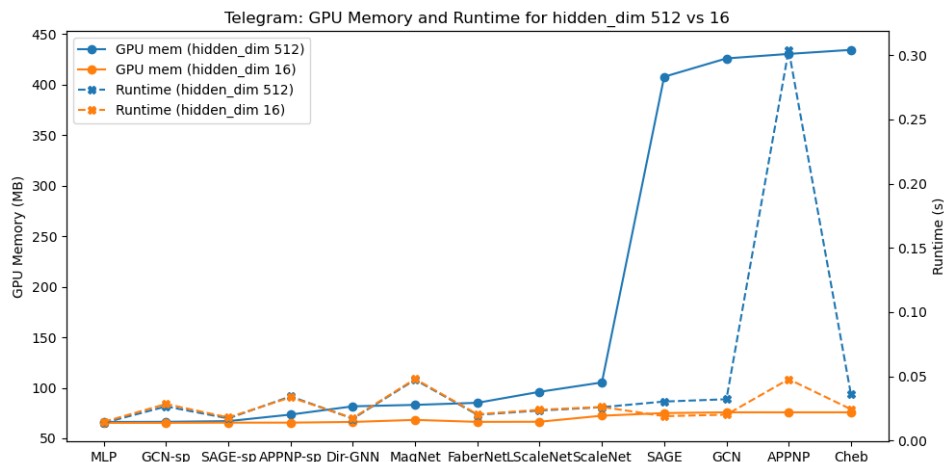

Figure G.1: Empirical evaluation of model complexity on the Telegram dataset. LargeScaleNet is abbreviated as LScaleNet.

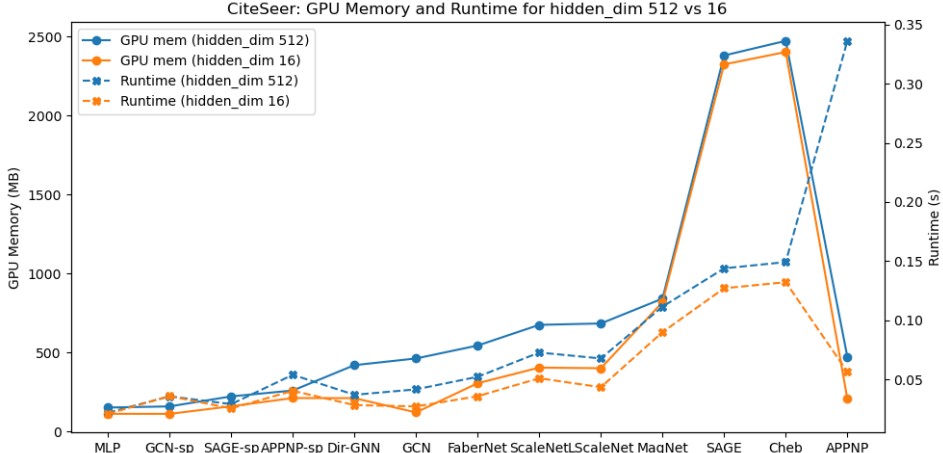

Figure G.2: Empirical evaluation of model complexity on the CiteSeer dataset. LargeScaleNet is abbreviated as LScaleNet.

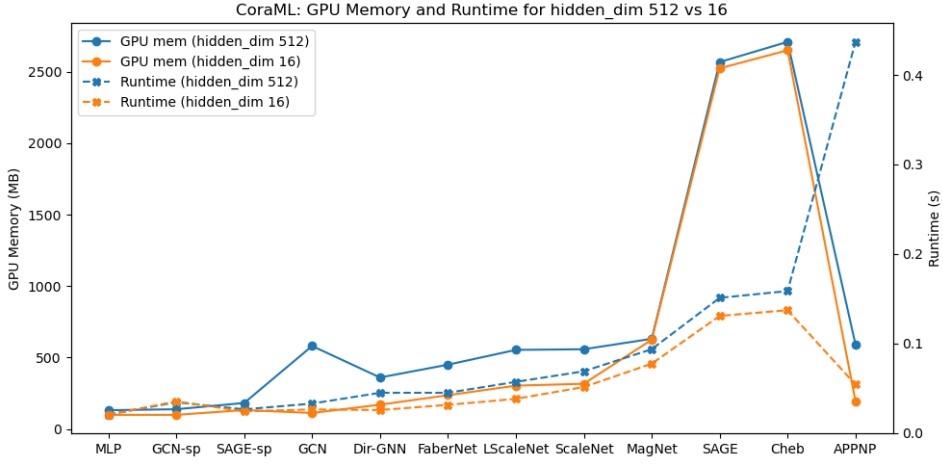

Figure G.3: Cora-ML

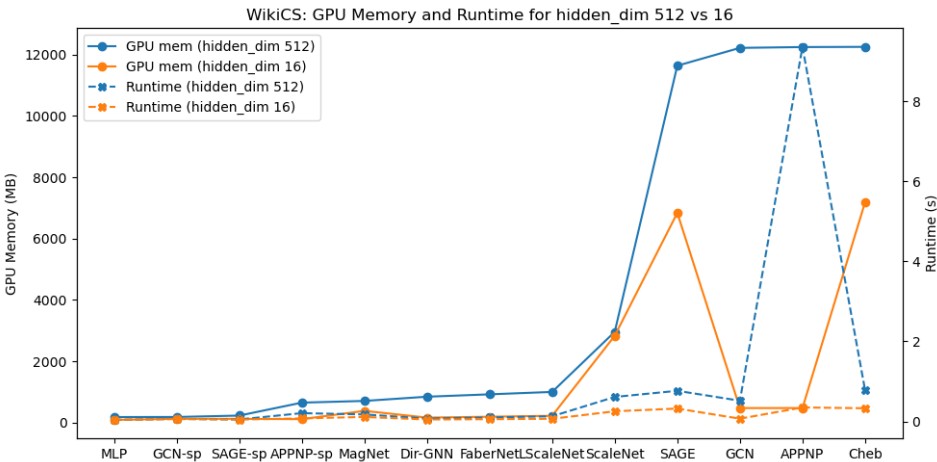

Figure G.4: Empirical evaluation of model complexity on the WikiCS dataset. LargeScaleNet is abbreviated as LScaleNet.

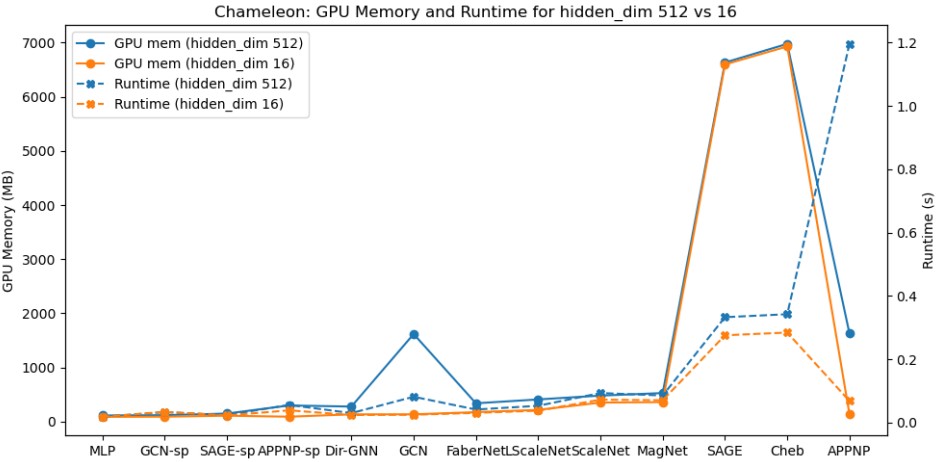

Figure G.5: Empirical evaluation of model complexity on the Chameleon dataset. LargeScaleNet is abbreviated as LScaleNet.

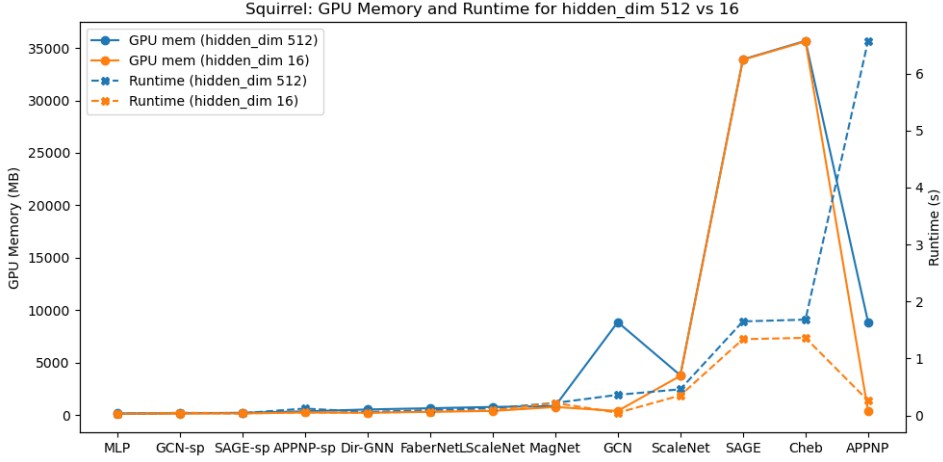

Figure G.6: Empirical evaluation of model complexity on the Squirrel dataset. LargeScaleNet is abbreviated as LScaleNet.

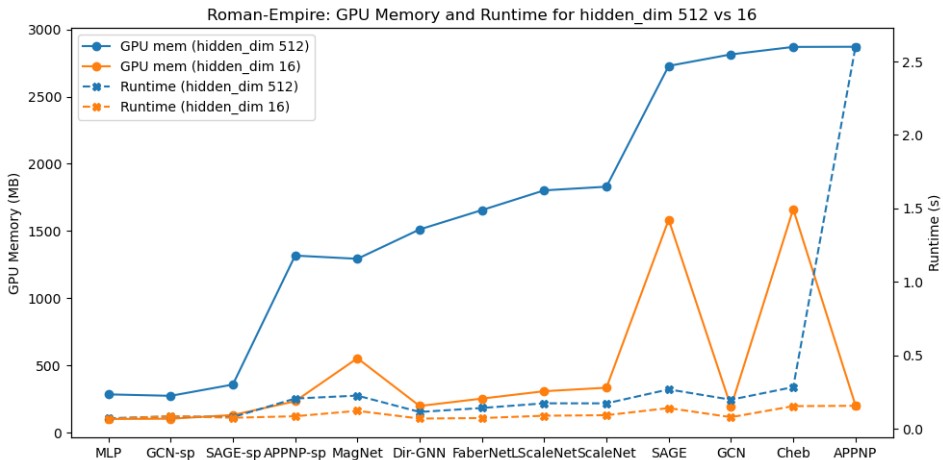

Figure G.7: Empirical evaluation of model complexity on the Roman-Empire dataset. LargeScaleNet is abbreviated as LScaleNet.

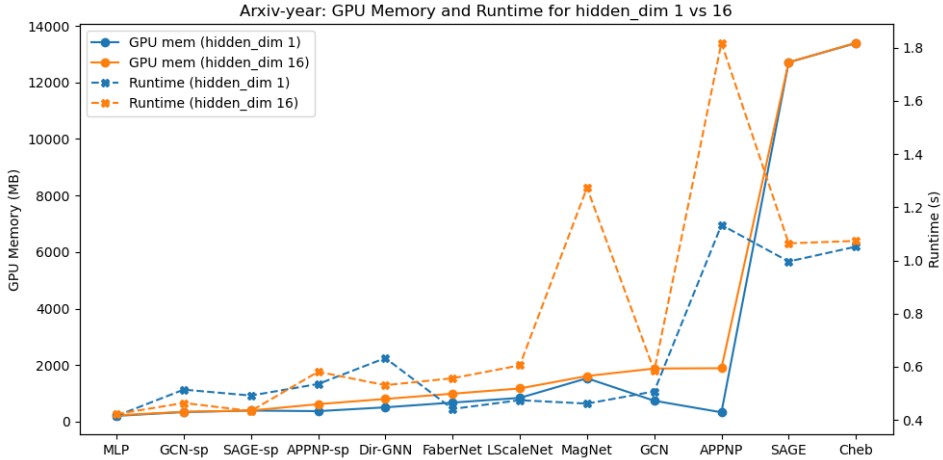

Figure G.8: Empirical evaluation of model complexity on the Arxiv-year dataset. LargeScaleNet is abbreviated as LScaleNet. ScaleNet runs out of Memory.

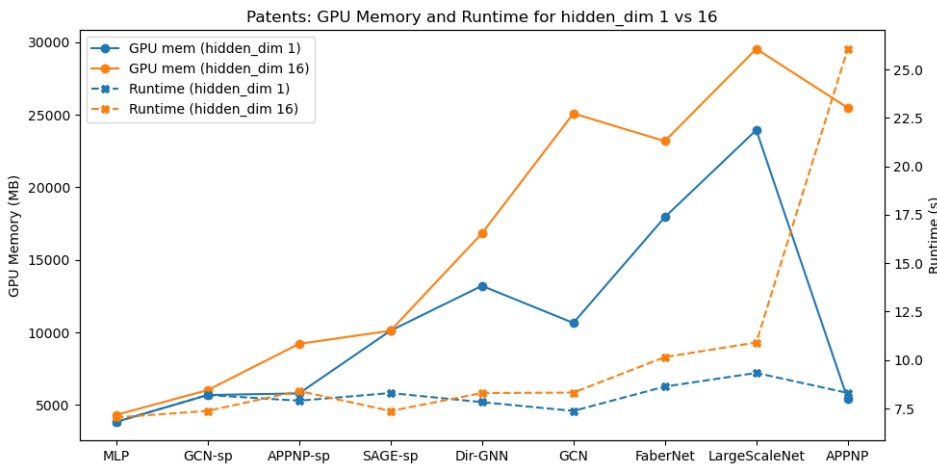

Figure G.9: Empirical evaluation of model complexity on the Snap-Patents dataset. GCN, SAGE, ChebNet, MagNet, and ScaleNet run out of memory. Model-sp denotes the version of each model using SparseTensor edge input.

## H  THE USE OF LARGE LANGUAGE MODELS (LLMS)

Large Language Models (Claude, ChatGPT, and Perplexity) were utilized as general-purpose assistance tools throughout this research. The LLMs did not contribute to research ideation or core scientific insights, but served purely as technical and editorial assistants. Below we detail the specific ways LLMs were employed:

- **Data Management and Analysis:** LLMs were used to generate Python code for extracting accuracy metrics from experimental output files, helping to automate the processing of large datasets and reduce manual coding errors.
- **Visualization:** Python code for generating figures and plots was written with LLM assistance, particularly for creating publication-quality visualizations of experimental results and performance comparisons.
- **Text Polishing:** LLMs assisted in improving the clarity and readability of manuscript text through simplification of complex technical descriptions, grammar and style corrections, and sentence restructuring for better flow.

All LLM-generated content was thoroughly reviewed, verified, and edited by the authors. The LLMs were not involved in experimental design, hypothesis formation, interpretation of results, or drawing scientific conclusions. The authors take full responsibility for all content in this paper, including any LLM-assisted portions, and have verified the accuracy of all facts, claims, and technical details presented.

