# OpenReview forum: "Scale-aware Message Passing for Graph Node Classification"
_ICLR.cc/2026/Conference — Submitted to ICLR 2026_

### Official Review · Reviewer_1jje · 2025-10-21

**Soundness:** 2
**Presentation:** 2
**Contribution:** 2
**Rating:** 2
**Confidence:** 3

**Summary:**

The paper introduces a directed GNN that uses 2-hop information in each step instead of relying on 1-hop information. In particular, the GNN aggregates in-in, in-out, out-in, and out-out edges independently and then aggregates over all of those in a second step.
The model is compared to a number of directed GNNs which are outperformed by the given architecture on several datasets.

**Strengths:**

The suggested network clearly outperforms other directed GNNs on homophilic and heterophilic datasets. By skipping normalization, the network can be run on large graphs too with only a mild impact on performance.
The quality of writing is good and the pictures helpful.

**Weaknesses:**

The main weakness of the paper is that multi-hop aggregation is not new at all. It has been around for multiple years in the undirected setting where it e.g. improved the expressiveness of the networks. None of that is mentioned in the paper. The step of checking the types of edges (io/oi/ii/oo) is not big, especially given that directed GNNs typically treat incoming and outgoing edges differently. I am missing novelty here. (or proper contextualization)

The core terms "scale-aware", "multi-scale", and "scale invariant" are not properly defined in the introduction such that it remains largely unclear what the paper aims to do. I did expect the paper to make connections to undirected graphs and corresponding models - since there are multiple models in which each layer uses information from more than the 1-hop neighborhood. Also the title made me think about graph coarsening methods that are not mentioned at all in the paper, as this is definitely a way of doing multi-scale learning (e.g. DiffPool or anything more recent). This is particularly important as "scaling" of graphs in the terms of this paper are strongly connected to graph coarsening where also multiple levels of the graph exist.

The paper seems to work with directed graphs. This is not mentioned in the title and also not mentioned in the abstract. I consider this a serious oversight. Also in the remainder of the paper it is typically unclear whether statements should hold for directed or undirected graphs.

The related work does not talk about graph transformers at all even though those architectures are SOTA on many graph-learning tasks and can with minor modifications to the positional encoding be applied to directed graphs as well. In general, the related work seemed to focus only on directed GNNs without looking into comparable architectures from the undirected realm.

The experiments do not include any graph transformers, even though the paper claims SOTA performance. Also the set of baseline methods do not include any relevant/recent/large (undirected) GNN such as a simple GatedGCN with virtual nodes, PNA, or GPS.

The proofs seem to drop non-linearities. But when doing so, one looses all the theoretical power and is restrcited to simple linear operations. I do not believe that such proofs are really helpful and "debunking" those two GNN architectures mentioned in the core contributions.

Since there are pretty bold claims about architectures being essentially the same it would have been nice to write a 1-2 sentence summary about the key steps of the corresponding proofs.


**Small remarks:**

62: I did not feel any kind of disruption in the paper.
108ff: Digraph Inception models are not explained, only mentioned as being "overly complicated". I expected the related work to at least minimally cover the connection between directed and undirected GNNs.
130: do graphs have self-loops?
142: essentially this is just k-hop edges, there is no "scaling" happening (as in multiplying with a constant).
242: I strongly disagree with this analogy. In image processing the task is either for the whole picture (typically classification) or per-Pixel (e.g. instance segmentation). What makes the analogy even more confusing is that of course the resolution of images will be changed in a typical ResNet during the processing, but the input is always of a fixed size (at least in classification this is typically the case, pooling is not applied in contrast to graph learning that needs to work on inputs of varying size). Please remove this analogy.
262: the information collected is definitely not global, but still local, even though the receptive field is larger. Transformers and virtual nodes are global, message-passing is local.
264: the definition of scale invariance is highly problematic as it is not clear what exactly it should mean that the GT label is not affected by the graph. In almost all settings that I am aware of, the labels depend on the structure, and if scale-invariance was true, then a simple MLP should be able to solve the task equally well. But that is not what is claimed afterwards.
Appendix: it would have been nice to use booktabs everywhere and consistently

**Neutral:**
- LargeScaleNet effectively simulates a 2-hop aggregation by performing 1-hop aggregation twice. The notable difference is only the missing normalization, which is hidden in the table caption (the bigtilde symbol above AA is not explained in the text). It is not clear how compatible the FaberNet and LargeScaleNet were in terms of concrete model tricks that are used.
- It would have been nice to discuss the connection between self-loops and skip-connections.
- For the hyperparameters: for undirected networks good dropout values are typically 0.1 and 0.2 which are not tested. Positional encodings are vitally important for good performance, sometimes SwiGLU and other more modern activations help. Also using a 2-3 layer head for classification is often a good idea (I could not figure out whether that was part of the architecture).

**Questions:**

- how does the model relate to existing work, especially in the unidrected setting
- how is the performance compared to graph transformers
- how well does it work on standard undirected datasets such as molpcba or the LRGB datasets? (especially since the performance on Cora/CiteSeer is rather low)

---

> ### Author Response · Authors · 2025-11-18
> **A. Definition of Scale Invariance (1)**
>
> We thank the reviewer for acknowledging our strengths. In particular, the reviewer noted that the proposed network clearly outperforms other directed GNNs on both homophilic and heterophilic datasets, that our LargeScaleNet can scale to large graphs with only a mild impact on performance, and that the quality of writing and the figures are helpful.
>
> Below, we clarify a number of important points raised by the reviewer.
>
> **A. Definition of Scale Invariance**
>
>  - >**(A1)** Re: The core terms "scale-aware", "multi-scale", and "scale invariant" are not properly defined in the introduction such that it remains largely unclear what the paper aims to do.
>
> We appreciate the reviewer’s comment. The definition of “scale invariant” is provided in Definition 3.5 (Line 264). Regarding “multi-scale”, since node classification can be performed on ego-graphs of different scales, combining these scaled ego-graphs can improve performance, which we refer to as multi-scale. Finally, “scale-aware” refers to selecting the optimal combination of scaled ego-graphs to achieve the best performance. We will revise the introduction to make these definitions and distinctions clearer.
>
>
> - >**(A2)** Re: 142: essentially this is just k-hop edges, there is no "scaling" happening (as in multiplying with a constant).
>
> We respectfully disagree with the reviewer's assessment. There appears to be a misunderstanding about the graph-theoretic meaning of matrix powers in our context.
>
> The original adjacency matrix $A$ encodes 1-hop (direct) connections between nodes. Specifically, $A_{ij}=1$ indicates a directed edge from node $v_i$ to $v_i$.
> The matrix product $AA$(or $A^2$) captures 2-hop paths(direct). Let us illustrate with a concrete example:
> *Toy graph:* Consider a directed graph with 3 nodes and 2 edges: $v_1 \rightarrow v_2$ and $v_2 \rightarrow v_3$.
> - In $A$: only $A_{12}=1$ and $A_{23}=1$; all other entries are 0.
> - In $AA$: only $AA_{13}=1$; all other entries are 0.
> This is because there exists exactly one 2-hop path from $v_1$ to $v_3$(via $v_2$). The reviewer can verify this through direct matrix multiplication.
>
> Table 1 and Definition 3.3 systematically enumerate different types of k-hop edge patterns and their corresponding representations through products of $A$ and $A^\top$. These are not arbitrary scalings but rather represent meaningful multi-hop structural patterns in the graph. We provide an additional pedagogical example in Appendix F.2.2 to further aid understanding.
>
> We would be happy to provide additional examples or clarifications if needed.
>
> - >**(A3)**  Re: Pooling is not applied.
>
> As explained in Response **A2**, aggregating via $A^2$ instead of A is analogous to pooling:
>
> -- Skips 1-hop neighbors: directly aggregates from 2-hop neighbors
> -- Reduces local redundancy while expanding receptive field
> -- Similar to strided convolution or pooling in CNNs, which downsample local detail while capturing broader spatial context.
>
> To generalize, consider the set defined in Line 212:
>
> ${A}_k = \bigl\{ \prod_{i=1}^k M_i \,\big|\, M_i \in \{ A, A^\top \} \bigr\}$
>
> Aggregating messages via ${A}_k$ for any k>1 effectively performs a pooling-like operation by enlarging the receptive field and suppressing fine-grained locality.

---

> > ### Author Response · Authors · 2025-11-18
> > **A. Definition of Scale Invariance (2)**
> >
> > - >**(A4)**  Re: I strongly disagree with this analogy(image classification VS. node classification). In image processing the task is either for the whole picture (typically classification) or per-Pixel (e.g. instance segmentation)...The input of ResNet is of Fixed size.
> >
> > We appreciate the reviewer’s observation. However, our multi-scale learning does not violate the Fixed Input Size Principle.
> >
> > -- In image classification:
> > A high-resolution image (e.g., 1000×1000) is first resized to a fixed resolution (e.g., 20×20) before being fed into ResNet. Multi-scale features are then obtained by taking patches at different scales and resizing all of them to the same fixed dimensions before convolution.
> >
> > Example(1000×1000 image):
> > | Scale | Window Size | Coverage | Interpretation |
> > |:---:|:---:|:---:|:---:|
> > | Local | 20×20 | Per-pixel detail | Edges, textures |
> > | Mid-range | 200×200→20×20 | Larger receptive field | Object parts |
> > | Global | 1000×1000→20×20 | Whole image | Full object|
> >
> > -- In our graph method:
> > In graphs, the fixed size corresponds to the number of GNN layers, which plays the same role as the fixed resized dimension in CNNs.
> > We use multi-scaled ego-graphs, but all scales are fed into the same fixed-depth GNN.
> >
> > For example, in a 2-layer GNN, the layer depth (2) is analogous to the 20×20 fixed input size in image classification: A 2-layer GNN aggregates 1-hop and 2-hop information.
> >
> > At scale k, the GNN aggregates k-hop and 2k-hop information—different receptive fields, but processed by the same fixed-layer model.
> >
> > Suppose the longest path in the ego-graph is 100 hops. Then multi-scale aggregation is:
> > | Scale | Window Size | Coverage | Interpretation |
> > |:---:|:---:|:---:|:---:|
> > | Local | 1-hop, 2-hop | Per-edge connection | Instant neighbors |
> > | Mid-range | 20-hop, 40-hop | Larger receptive field | Extended neighborhood |
> > | Global | 50-hop, 100-hop | Wholegraph | Full ego-graph |
> >
> > Each scale captures different levels of information, but the graph input and model depth remain fixed. This is directly analogous to CNNs, where local, mid-range, and global image patches are all resized to the same fixed dimensions before convolution.
> >
> > -  >**(A5)** Re: 262: the information collected is definitely not global, but still local, even though the receptive field is larger.
> >
> > We thank the reviewers for this constructive criticism and agree that our terminology needs clarification.
> >
> > For a node $v_i$ with ego-graph diameter $L_{v_i}$, an $l$-layer model with scale factor $k$ covers the full ego-graph only if: $kl\geq L_{v_i}$. Since $L_{v_i}$ varies accross nodes, global coverage is not generally guaranteed—reinforcing that “long-range” is the more accurate term.
> >
> > Images have fixed and bounded spatial extent, so “global” always refers to the entire image. Graphs, in contrast, have irregular topology and node-dependent diameters, and potentially unbounded for loops, so the notion of “global” depends on the ego-graph's diameter, which varies across nodes and datasets.
> >
> > Our method aggregates information from progressively larger multi-hop neighborhoods (e.g., 3-hop, 6-hop), which is long-range but not necessarily graph-wide. We therefore replace “global’’ with “long-range’’ throughout the paper.
> >
> > We appreciate the reviewer’s insight and have revised the manuscript accordingly.
> >
> > - >**(A6)**  Re: In almost all settings that I am aware of, the labels depend on the structure, and if scale-invariance was true, then a simple MLP should be able to solve the task equally well. But that is not what is claimed afterwards.
> >
> > We appreciate this important observation. We clarify structure-agnostic models like MLPs outperform GNNs on certain datasets[2], where node self-features alone are sufficient for classification. Thus node labels can depend on features alone, structure alone, or their combination[3].
> >
> > Our scale invariance property addresses cases where multiple structural scales (not necessarily all scales) contain sufficient information for classification.
> > MLP can be seen as a special case : 0-scaled feature.
> > In application, following FaberNet, our model includes a zero-order hyperparameter that, when enabled, adds the node's original features to its aggregated features, allowing the model to leverage k=0 information when beneficial.
> >
> > We incorporate this as the special case  in our revised Definition of Scale Invariance.
> >
> > [1] Emanuele Rossi et al. Edge directionality improves learning on heterophilic graphs. Learning on Graphs Conference, pp. 25–1. PMLR, 2024.
> >
> > [2] Zheng, W., Huang, E. W., Rao, N., Katariya, S., Wang, Z., and Subbian, K. Cold Brew: Distilling Graph Node Representations with Incomplete or Missing Neighborhoods, March 2022. URL http://arxiv.org/ abs/2111.04840.
> >
> > [3] Qin Jiang, Chengjia Wang, Michael Lones, and Wei Pang. Demystifying mpnns: Message passing as merely efficient matrix multiplication, 2025.

---

> ### Author Response · Authors · 2025-11-18
> **B. "Debunking" 3 models**
>
> **B. "Debunking" 3 models**
> - >**(B1)**  Re: Digraph Inception models are not explained, only mentioned as being "overly complicated"
>
> Digraph Inception models assign different edge weights to in-neighbors and out-neighbors, where the weights are computed using an eigenvalue decomposition of the adjacency matrix—an operation that becomes expensive on large graphs. In Line 108, we provide a brief summary of these models, and a full description is included in Appendix E.1. In the revised version, we clarify the wording in the main text to make this explanation more accessible to first-time readers.
>
>
>
>
> - >**(B2)** Re: The proofs seem to drop non-linearities. But when doing so, one looses all the theoretical power and is restrcited to simple linear operations. I do not believe that such proofs are really helpful and "debunking" those two GNN architectures mentioned in the core contributions.
> >Re: I did not feel any kind of disruption in the paper
>
> Non-linearities are indeed essential components of our models. We omitted them in several parts of the theoretical exposition solely for notational simplicity. In Line 1331, 1511, 2093, we repeatedly state that:
> - > For notational simplicity, we omit the non-linearity, while the model includes non-linearities,
>
> We apologise that this may not have been sufficiently clear on first reading. We will make this clarification more prominent and repeat it where necessary to avoid any further confusion.
>
>
> ---
>
> Regarding our “debunking’’ of the three models (DiGib, MagNet, and FaberNet):
>
> (1). **DiGib:**
>    Our concerns are supported by experimental results. We also reached out to multiple authors of DiGib[5] for clarification, but unfortunately did not receive replies.
>
> (2). **MagNet[6]:**
>    Our critique is based on a mathematical proof. We hope the reviewer can evaluate this more effectively now that we have clarified that our theoretical analysis does *not* drop non-linearities from the model.
>
> (3). **HoloNet:**
>    Our finding is based on a code flaw that leads to the reported strong performance of HoloNet(the same paper of FaberNet)[7]. We contacted the first author, who kindly acknowledged the issue. He raised an issue on GitHub(https://github.com/ChristianKoke/HoloNets/issues/1) and indicated that revisions would be made in the arXiv version and on OpenReview.
>    To the best of our knowledge, no such updates have been posted yet. We believe this is because the underlying issue is fundamental and not easily resolved.
>
> We hope this clarifies our contributions.
>
> [5] Zekun Tong, Yuxuan Liang, Changsheng Sun, Xinke Li, David Rosenblum, and Andrew Lim. Digraph inception convolutional networks. Advances in neural information processing systems, 33: 17907–17918, 2020a.
>
> [6] Xitong Zhang, Yixuan He, Nathan Brugnone, Michael Perlmutter, and Matthew Hirn. Magnet: A neural network for directed graphs. Advances in neural information processing systems, 34: 27003–27015, 2021
>
> [7] Christian Koke and Daniel Cremers. Holonets: Spectral convolutions do extend to directed graphs. In The Twelfth International Conference on Learning Representations, 2024.

---

> ### Author Response · Authors · 2025-11-18
> **C. Directed VS. Undirected (1)**
>
> **C. Directed VS. Undirected**
> - >**(C1)** Re: The paper seems to work with directed graphs... Also in the remainder of the paper it is typically unclear whether statements should hold for directed or undirected graphs.
>
> This paper deals with general graph. In Line 131, we state:
> > We focus on node classification on directed graphs; an undirected graph can be seen as a special case where each edge is accompanied by its reverse.
>
> A similar statement appears in Line 1214. Following the approach of Dir-GNN [1], our models aggregate in-neighbors and out-neighbors separately. When applied to undirected graphs, this separate aggregation produces the same result as aggregating only in-neighbors or only out-neighbors.
>
> We are happy to provide a more detailed explanation if this distinction remains unclear.
>
>
>
> - >**(C2)** Re: I expected the related work to at least minimally cover the connection between directed and undirected GNNs.
>
> Their commonality is that GNNs use neighbors’ features as input to an MLP. The main difference lies in which neighbors are considered. Undirected GNNs do not distinguish between in-neighbors and out-neighbors, treating all neighbor equally; their adjacency matrix is symmetric, with all edges encoded as 1. Directed GNNs, on the other hand, handle in- and out-neighbors differently. For exmaple:
> - **DiGib** assigns different edge weights to in-neighbors and out-neighbors.
> - **MagNet** uses a Hermitian matrix to encode in-neighbor and out-neighbor.
> - **Dir-GNN** aggregates in-neighbors and out-neighbors separately.
>
> We hope this clarifies the connection. If you’d like, we’re happy to provide further explanation.
>
>
>
> - >**(C3)** Re: how well does it work on standard undirected datasets such as molpcba or the LRGB datasets? (especially since the performance on Cora/CiteSeer is rather low)
>
> Thank you for pointing out the low performance on Cora/CiteSeer. After further investigation, we found that the versions of Cora/CiteSeer commonly used in directed-graph research differ substantially from the versions used in undirected GNN or Graph Transformer benchmarks. These datasets are not equivalent, and the differences go beyond simply adding reverse edges.
>
> Below is a comparison illustrating the mismatch:
> |Dataset Variant  | #Nodes | #Edges | #Features |
> |---|---|---|---|
> |CoraML (Directed) | 2,995 | 8,416 | 2,879 |
> | Cora (Undirected)| 2,708 | 5,278 | 1,433 |
> |  |  |  |  |
> | CiteSeer (Directed) | 3,312 | 4,715 | 3,703 |
> | CiteSeer (Undirected) | 3,327 | 4,522 | 3,703 |
>
> As shown, the datasets differ even in the number of nodes and feature dimensions. These variants were constructed by different groups using different preprocessing pipelines. Therefore, performance numbers reported in the undirected-graph literature are not directly comparable to results obtained on the directed versions used in our experiments.
>
> As the review 1jje have confirmed, our model clearly outperforms existing directed GNN baselines when evaluated on the same directed datasets, highlighting its strength in this setting. We will evaluate our models on MolPCBA and the LRGB datasets as soon as we complete our Graph Transformer experiments.
>
> [1] Emanuele Rossi et al. Edge directionality improves learning on heterophilic graphs. Learning on Graphs Conference, pp. 25–1. PMLR, 2024.

---

> ### Author Response · Authors · 2025-11-18
> **C. Directed VS. Undirected  (2)**
>
> - >**(C4)**  Re: I did expect the paper to make connections to undirected graphs and corresponding models - since there are multiple models in which each layer uses information from more than the 1-hop neighborhood.
>
> multi-scale undirected VS. multi-scale directed
>
> - For multi-scale undirected models:
>   - (1) if the model doesn't add reverse edges:
>
> | Hop | Adjacency Matrix | Node Embedding |
> |--------------|---------------------|-------------------|
> | 1-hop | $A$ | $AXW$ |
> | 2-hop | $A^2$ | $A^2XW$ |
> | k-hop | $A^k$ | $A^kXW$ |
>
>   - (2) if the model adds reverse edges for each edge:
> For performance, undirected models would add reverse edges before aggregation, making its factual adjacency matrix to be $A+A^\top$.
>
> | Hop | Adjacency Matrix | Node Embedding |
> |------------|---------------------|-------------------|
> | 1-hop | $A+A^\top$ | $(A+A^\top)XW$ |
> | 2-hop | $(A+A^\top)(A+A^\top)$ | $(A+A^\top)^2XW$ |
> | k-hop | $(A+A^\top)^k$ | $(A+A^\top)^kXW$ |
>
>
>
> - For multi-scale directed models:
> the adjacency matrix to encode 1-hop neighborhood can be $A$, $A^\top$ or $A+A^\top$ based on directional choice $\alpha$, and the node embedding is $\alpha AXW_1 + (1-\alpha)A^\top XW_2$.
> thus its  2-hop neighborhood can be $AA$, $AA^\top$, $A^\top A$, or $A^\top A^\top$.
>
> | Hop | Adjacency Matrix | Node Embedding |
> |------|-----------------------|-------------------------|
> | 1-hop | $A$, $A^\top$ | $\alpha AXW_1 + (1-\alpha)A^\top XW_2$ |
> | 2-hop | $AA$,  $A^\top A^\top$ | $\beta AA^\top XW_1 + (1-\beta)A^\top AXW_2 $|
> |2-hop | $AA^\top$, $A^\top A$ | $ \gamma AAXW_1 + (1-\gamma)A^\top A^\top XW_2  $|
>
>
> In sum, undirected models can only aggregate through one direction. For performance, they usually augment edges into undirected graphs if original graph is directed graph. While directed aggregation models can aggregate in-neighbors and out-neighbors respectively and then add them, providing more flexibility. For some graph datasets(Chameleon), only aggregate in-neighbor is good, while aggregating both in- and out-neighbors get worse performance.

---

> ### Author Response · Authors · 2025-11-18
> **D. Normalization**
>
> **D. Normalization**
> - >**(D1)** Re: The notable difference(of LargeScaleNet) is only the missing normalization, which is hidden in the table caption (the bigtilde symbol above AA is not explained in the text).
> >Re: By skipping normalization, the network can be run on large graphs too...
>
> We would like to clarify that we did not skip normalization. Our network achieves scalability to large graphs by changing the multiplication sequence(from $A$ to $AA$ is a multiplication step, and normalization can also be seen as a multiplication). Table E.5(in the Appendix) shows comparison of the sequences of both models, both of them have normalization, ScaleNet's normalization(turning $AA$ into $\widetilde{AA}$) is in Step 2, while LargeScaleNet's normalization(turning $A$ into $\tilde{A}$) is in Step 1.
>
> We did explain wildetilde of AA implicitly in text(Line 375-376).
> > ScaleNet computes the scaled adjacency matrix $AA$  before normalization, then applies it as $\widetilde{AA}X$, where $X$ is the node feature matrix.
>
> We have clarified the wording in the revised version to make this explicit.
>
> Additionally, reviewer JKmj correctly understood our scalability approach:
> > LargeScaleNet improves scalability by replacing explicit adjacency multiplications with sequential sparse operations.
>
> We are happy to provide further clarification on this approach if needed.
>
> - >**(D2)** Re: It is not clear how compatible the FaberNet and LargeScaleNet were in terms of concrete model tricks that are used.
>
> The comparison between FaberNet and LargeScaleNet is detailed in Table E.6 (in the Appendix). Both models follow the same sequence of matrix multiplications, while LargeScaleNet is more expressive than FaberNet in terms of direction options.

---

> ### Author Response · Authors · 2025-11-19
> **E: Graph Transformer**
>
> **E. Graph Transformer**
> - >**(E1)** Re: graph transformers not mentioned
>
> We did mention Graph Transformers in Appendix E (Lines 2037–2054), where we explained why they are not the focus of this work. Graph Transformers are indeed valuable for exploring all-pair interactions. However, our experimental findings suggest that learning specific, real-valued edge weights is less beneficial than simply modeling the existence of edges. If we were to investigate Graph Transformers, we would therefore focus on using self-attention to infer the presence of edges, rather than assuming a fully connected graph and learning dense edge weights. Incorporating structural information—for example, through positional encodings concatenated with the original node features—could also enhance such an approach.
>
> Since the title and core contribution of this paper center on message passing, our primary focus remains on MPNNs. These directions for Graph Transformers are promising but fall outside the current scope and are left for future work.
>
> In the revised version, we have added a brief literature review on Graph Transformers in Appendix D.3.
>
> - >**(E2)**  Re: claims SOTA performance but no experiments on graph transformers.  Also the set of baseline methods do not include any relevant/recent/large (undirected) GNN such as a simple GatedGCN with virtual nodes, PNA, or GPS.
>
> As requested by the reviewer, we are evaluating three strong Graph Transformer models—GraphGPS, SGFormer, and Polynormer—and will incorporate the updated results into Tables 2 and 4 of the revised manuscript.
>
> Running experiments on our large datasets requires several days per model. After completing the Graph Transformer experiments, we will proceed to evaluate the additional models suggested by the reviewer (e.g., GatedGCN with virtual nodes, PNA). If the reviewer considers the current set of baselines insufficient to support a “SOTA” claim, we are happy to remove that wording. As the reviewer noted, our method clearly outperforms existing directed GNNs, which are the most relevant baselines for our problem setting.

---

> ### Author Response · Authors · 2025-11-19
> **F. Other Remarks**
>
> **F. Other Remarks**
>
>
> - >**(F1)**  Re: graph coarsening methods are not mentioned at all in the paper
>
> Thank you for this valuable suggestion. We agree that connecting multi-scale learning with graph coarsening methods is important. We have actually conducted a literature review on graph pooling methods, in Appendix D.3.2 (Line 1644) of our original submission. In the revised version, we have added citations to DiffPool.
>
> - >**(F2)**  Re: 130: do graphs have self-loops?
>
> Original graphs may contain some self-loops. For example, a web page might cite itself. Whether a node has a self-loop depends on the individual node and the dataset. For reference:
> Chameleon: Out of 2,277 nodes, 50 nodes have self-loops.
> Squirrel: Out of 5,201 nodes, 140 nodes have self-loops.
>
>
> - >**(F3)** Re: Since there are pretty bold claims about architectures being essentially the same it would have been nice to write a 1-2 sentence summary about the key steps of the corresponding proofs.
>
>
> We theoretically deduce the output of MagNet as Equation A.13, then we claim in Line 1959:
> > As layer-wise propagation of GraphSAGE is $XW_1+ D^{-1}AXW_2$, the first two terms of Equation A.13 are equivalent to GraphSAGE but with GCN's normalization in place of the original row normalization scheme.
>
> We are happy to provide further clarification if this is still unclear, or to revise the text to make it more explicit.
>
>
>
> - >**(F4)**  Re: Appendix: it would have been nice to use booktabs everywhere and consistently
>
> Thank you for your suggestion on improving the presentation of our paper. We have replaced all \hline commands with booktabs commands for consistency.
>
>
> - >**(F5)** Re: Neutral: It would have been nice to discuss the connection between self-loops and skip-connections.
>
> We are the first to theoretically discuss the influence of self-loops (see Appendix F). Could you clarify what you mean by “skip-connection” so we can address the connection appropriately?
>
>
>
>
> - >**(F6)** Re: For undirected networks good dropout values are typically 0.1 and 0.2 which are not tested.
>
> Thank you for the reviewer’s recommendation. We will test dropout values of 0.1 and 0.2 as suggested, and any updated results will be reported in Tables 2 and 4 in the revised PDF.
>
>
> Finally, we have addressed all reviewer questions according to their respective categories. If any question has been overlooked, we would appreciate the reviewer pointing it out, and we would be happy to provide a response.

---

> ### Comment · Reviewer_1jje · 2025-11-20
> **Overall answer to author rebuttal**
>
> Thanks for reaching out and giving a highly-detailed answer spanning eight(!) answers at once. Let me first go to the most important points to not get lost in detailed answers.
>
> One of the essential points that led to my low overall score for the paper is that the idea of using k-hop aggregation has been around for about six years [1-6], where Mixhop [2] is one of the early papers doing it and one that is already cited in the appendix. Six years is approximately as long as people have been looking at GNNs in earnest. The method itself is lacking proper embedding into the related literature which often does things that are highly related. There is virtually no comparison to earlier work (which is mentioned at page 31 of the appendix but missing a multitude of other approaches that perform k-hop aggregation).
>
> Then for the proofs: dropping nonlinearities is *not* just improving notational simplicity, but significantly changing what a model can do. While it is known that purely linear models are powerful (see e.g. FastGCN), they are trivially less expressive. I would also like to note that the universal approximation theorem necessarily depends on nonlinearities. Lets just say, the presentation without nonlinearities is oversimplifying things and thus not a good idea - especially since UAT is used without this disclaimer.
>
> A little more detailed about the proofs: What exactly is the shape of matrix $\mathbf {X_i}$? While for a single node in $\mathbf X$ the substitution of the multiplication with $\mathbf A$ clearly makes sense, I have trouble seing how this step is done when $\mathbf X$ contains all nodes from a graph (or e.g. the 2-hop neighborhood of a node $u$). What are the dimensions of all of those matrices?
>
> And one comment on GTs: The related work section does *not* contain why transformers are ignored, this happens at the end of Appendix E3 and with a non-convincing argument since GTs nowadays dominate nearly all (undirected) graph benchmarks. It is thus hard to state that GTs "offer limited practical benefit in most real-world tasks". Also I did not understand why transformers "focus on learning exact edge weights" - while they are performing their global aggregation based on the embeddings of source and target (i.e. edge weights are only implicitly defined). While I do agree that GNNs have value and not obsolete, a discussion of GTs is important, especially since GTs do perform aggregation at "all scales" at once.
>
> So overall, I have no problem with the method itself, its just that it has been proposed in a way that does not acknowledge previous work doing highly similar things. Then the formal part is hard to read (and I am unsure about the proofs, but I only took a cursory glance at them). And finally the experiments are potentially problematic: the values in the paper for plain GCN are a lot lower than the ones in [7]. In particular, GCN outperforms ScaleNet on all homophilic datasets (except for telegram which is not in [7]). It is a little suprising though that the performance of GCN on Chameleon and Squirrel reported in the paper is a lot higher than the one reported in [7] - this could indicate that something about the experiment is off. In general, I would work with the baseline values provided by [7] and double check the heterophilic datasets.
>
>
> [1] How Powerful are K-hop Message Passing Graph Neural Networks (Neurips 2022)
> [2]  Mixhop: Higher-order graph convolutional architectures via sparsified neighborhood mixing (ICML 2019)
> [3] k-hop graph neural networks (Neural Networks 2020)
> [4]  Multi-hop attention graph neural networks (IJCAI 2021)
> [5] Adaptive universal generalized pagerank graph neural network (ICLR 2021)
> [6] Graph convolutions that can finally model local structure. (arxiv 2020)
> [7] Classic GNNs are Strong Baselines: Reassessing GNNs for Node Classification (NeurIPS 2024)

---

> > ### Author Response · Authors · 2025-11-20
> > **Novelty and Nonlineary**
> >
> > Thank you for your prompt response. We apologize for not uploading the revision before your reply; it is now available. At this stage, I can address part of your questions while I continue preparing additional experiments and responses to other reviewers. We will provide full answers to all your questions once these are completed.
> >
> > **1. Novelty of k-hop aggregation**
> > We have addressed this question in our response to Reviewer mGZC, Q1, regarding the difference from other multi-scale models.
> >
> > **2. Dropping nonlinearities**
> > Our models are not linear—they do contain nonlinearities. We omit them in expressions solely for simplicity in writing. We can add them if that is still confusing.
> >
> > We will address the remaining issues in a subsequent response. Thank you very much for your support and feedback.
> >
> > Best regards,
> >
> > The authors

---

> > ### Author Response · Authors · 2025-11-20
> > **I: Graph Transformer(edge weights)**
> >
> > > Re: Also I did not understand why transformers "focus on learning exact edge weights" - while they are performing their global aggregation based on the embeddings of source and target (i.e. edge weights are only implicitly defined).
> >
> > Graph Transformers assume that all pairs of nodes are connected, which is equivalent to operating on a fully connected graph.
> > They learn attention scores for every node pair, which is equivalent to learning the edge weights of all possible connections.
> >
> > > Re: It is thus hard to state that GTs "offer limited practical benefit in most real-world tasks".
> >
> > As the reviewer themselves cited paper [7], for node classification, Graph Transformers perform worse than classic baselines such as GCN, GAT, and GraphSAGE.
> > That is why we make this statement and cite [7] to support it.
> >
> > We guess the reviewer might be deeply involved in research on Graph Transformers, and it might hurt their feelings when GT is disputed. We understand that, but we speak in a “call a spade a spade” way and hope the reviewer can be fact-oriented. We are also happy to learn more about the potential of GTs, but this paper focuses on MPNNs. After adding a review of Graph Transformers and experiments on three GT models, we believe we have covered GTs sufficiently. We would now like to focus on the contributions and experiments of this paper on MPNNs.
> >
> >
> > [7] Classic GNNs are Strong Baselines: Reassessing GNNs for Node Classification (NeurIPS 2024)

---

> ### Author Response · Authors · 2025-11-20
> **G: Reviewer 1jje is suspicious our experimental results**
>
> Thank you to Reviewer 1jje for directly raising concerns regarding the validity of our experimental results.
> We understand the suspicion, and we address it clearly below.
>
>
> - (G1) The performance is "too good to be true"
>
> Our reported results on Chameleon (80%) and Squirrel (76%) indeed appear significantly higher than those reported in [7], where the best results are around 46 and 45, respectively.
> However, as shown in the table below, our improvements are entirely consistent with existing directed-graph baselines.
> |   Model  |   Chameleon  |   Squirrel   |
> |:--------:|:------------:|:------------:|
> | Dir-GNN  | 78.77 ± 1.72 | 74.43 ± 0.74 |
> | ScaleNet | 80.1 ± 1.5   | 76.0 ± 2.0   |
>
> Our results are only a modest improvement over Dir-GNN—not an unexplained leap.
>
> - (G2) Dataset mismatch: “Undirected vs. Directed”
>
> > Re: GCN outperforms ScaleNet on all homophilic datasets
> > Re: It is a little suprising though that the performance of GCN on Chameleon and Squirrel reported in the paper is a lot higher than the one reported in [7] - this could indicate that something about the experiment is off.
> > Re: I would work with the baseline values provided by [7]
>
> We guess Reviewer 1jje did not read our rebuttal (“C. Directed vs. Undirected”) before responding.
> In our rebuttal C, Undirected vs. Directed, we detailed that some homophilic datasets used in the directed-graph research community differ essentially from those used in undirected GNN works—even when they share the same names. These versions cannot be converted into each other by simply adding reverse edges.
>
> For the heterophilic datasets, we now report that the datasets used in [7] are also essentially different from ours, as shown below:
>
> |        Dataset       | #Nodes |  #Edges |
> |:--------------------:|:------:|:-------:|
> | Chameleon (in [7])   | 890    | 8,854   |
> | Chameleon (directed) | 2,277  | 36,101  |
> | Squirrel (in [7])    | 2,223  | 46,998  |
> | Squirrel (directed)  | 5,201  | 217,073 |
>
> The reported number of nodes and edges in [7] are very different from our datasets.
>
> After reading the code of [7] (https://github.com/LUOyk1999/tunedGNN
> ), Chameleon and Squirrel are read from a local .npz file, which does not exist in their repository.
> In contrast, our Chameleon and Squirrel datasets come directly from torch_geometric.datasets, ensuring full reproducibility.
>
> This dataset mismatch fully explains the performance discrepancy.
>
> Based only on this discrepancy, Reviewer 1jje judged that our experiment is “off” and preferred [7].
> We think this is unfair.
>
>
> We sincerely encourage Reviewer 1jje to run our publicly released code.
> When we first read the Dir-GNN paper [1], we were suspicious too, so we experimented and confirmed it.
> Then we desgined models that achieved better performance than Dir-GNN [1].
> When we read the HoloNet paper [2], we did not believe spectral GNN works, and we discovered that it was based on a code error, and the authors very nicely admitted their mistake.
> Our discussion with Reviewer 1jje would be much more productive if Reviewer 1jje could make judgments based on experimental evidence or after actually read the corresponding parts in our paper and rebuttal.
>
> - **Easy to run our code**: It is quite simple to reproduce our models’ performance.
> You don't have to tune the model at all, as all hyperparameters are stored in a file.
> You just set use_best_hyperparams = 1 in args.py, then run any dataset you wish to test.
>
>
> We are happy to answer any more questions the Reviewer might still have.
>
>
>
>
> [1] Rossi, Emanuele, et al. "Edge directionality improves learning on heterophilic graphs." Learning on graphs conference. PMLR, 2024.
>
> [2] Koke, Christian, and Daniel Cremers. "Holonets: Spectral convolutions do extend to directed graphs." arXiv preprint arXiv:2310.02232 (2023).
>
> [7] Classic GNNs are Strong Baselines: Reassessing GNNs for Node Classification (NeurIPS 2024)

---

> ### Author Response · Authors · 2025-11-20
> **H: main reason for so low score**
>
> > Re: One of the essential points that led to my low overall score for the paper is that the idea of using k-hop aggregation has been around for about six years [1-6], where Mixhop [2] is one of the early papers doing it and one that is already cited in the appendix
> > Re: So overall, I have no problem with the method itself, its just that it has been proposed in a way that does not acknowledge previous work doing highly similar things.
>
>
> As the reviewer confirmed, we have cited several k-hop aggregation methods in the appendix. We actually included a review of high-order aggregation. However, the reviewer claimed that we “do not acknowledge previous work doing highly similar things.” This is paradoxical.
>
> In Rebuttal “Q1: Difference from other multi-scale models” to Reviewer mGZC and Question (3) to Reviewer c5V6, we stated that previous multi-scale models are all designed for undirected graphs.
> Ours is the first to handle directed graphs, with many more detailed advantages explained in those rebuttals.
> Therefore, Reviewer 1jje’s claim that previous works have done “highly similar things” is unfounded.
>
> Reviewer 1jje also ignore our strong performance and instead claims the results are “off” simply because they are too good, which unfortunately prevents the reviewer from recognizing our actual contribution.

---

> ### Author Response · Authors · 2025-11-21
> **J: Dimension of $X_i$**
>
> > Re: A little more detailed about the proofs: What exactly is the shape of matrix $X_{i}$? While for a single node in $X$ the substitution of the multiplication with $A$ clearly makes sense, I have trouble seing how this step is done when $X$ contains all nodes from a graph (or e.g. the 2-hop neighborhood of a node $u$). What are the dimensions of all of those matrices?
>
> This is an excellent question, and we appreciate the opportunity to clarify the notation.
>
> In Line 1045 of our original manuscript, we say:
> > $X_i$ denote 1-hop in-neighbors of X.
>
> Correspondingly, $X_o$ denotes 1-hop out-neighbors of $X$, $X_{ii}$ denotes 2-hop in-neighbors of $X$, and so on.
>
>
> ### (J1) Clarifying the Shape of $X$ and $X_i$
>
> $X$ is an $n\times f$ matrix, where each of the $n$ rows is the $f$-dimensional feature vector of a node.
>
> In contrast, $X_i$ is **not** a matrix. It has $n$ rows, but each row is a **list of feature vectors**, one for each 1-hop in-neighbor of that node.
> For example, row $k$ of $X_i$ contains the feature vectors of all 1-hop in-neighbors of node $k$.
> If node $k$ has 10 in-neighbors, then row $k$ of $X_i$ is a list of 10 vectors, each of length $f$.
>
> In sum, the structure of $X_i$ is:
>
> - $X_i$ has $n$ rows, one per node.
> - Each row is a **list of feature vectors** of that node’s neighbors.
> - Each feature vector has dimension $f$.
> - The number of vectors in each row equals the number of neighbors of that node, which varies across nodes.
>
>
> Formally:
>
> $$
> X_i =
> \begin{bmatrix}
> X_i[1] \\
> X_i[2] \\
> \vdots \\
> X_i[n]
> \end{bmatrix},
> $$
>
>
>
> where each $X_i[k]$ is a **list of vectors of length $f$**.
> Since different nodes have different numbers of neighbors, the “width’’ of rows varies, so $X_i$ does not have a standard matrix dimension.
>
>
>
> Our notation differs slightly from conventional usage. We recognize that this may be confusing when compared to standard formulations, but the definitions in our manuscript are internally consistent and correct. We genuinely appreciate the reviewer’s effort to interpret our notation.
>
> To answer your question about all dimension of the matrices mentioned, the adjacency matrix $A$ has shape $n \times n$.
>
>
>
>
> ---
> ### (J2)Clarifying your comment about 2-hop neighborhoods
>
> You mentioned "$X$ contains all nodes from a graph (or e.g. the 2-hop neighborhood of a node $u$)". We fully agree that $X$ contains all nodes from the graph, as defined.
> However, we are not entirely certain how you intend to connect $X$ to the 2-hop neighborhood of a specifica node $u$ in this context.
> Could you please clarify what you mean by this? Once we understand your interpretation, we will be happy to give a precise and complete answer.
>
> ### (J3) Correction to Line 1067 (Appendix)
>
> There is a typo in Line 1067 of the appendix (of our original manuscript). It should read:
> >$AX = \sum X_i$, where $i$ represents the 1-hop in-edges.
>
> In the original manuscript, we mistakenly used the capital letter $I$ after “where,” which may cause confusion.
>
> ---
> Does the above explanation make sense?
> We thank the reviewer for their careful reading and insightful question. We are very happy to clarify this point.

---

> > ### Comment · Reviewer_1jje · 2025-11-21
> >
> > Dear authors,
> >
> > **I do not have any problem with the method itself.** It clearly makes sense and does work well as shown on the heterophilic datasets. My problem is with the **presentation** and **context**.
> >
> > I did indeed not check the dataset statistics when looking for baseline results, having a lot more data of course explains better results.
> > I have to admit that it is extremely annoying that the directed and undirected datasets are different which makes it incredibly hard to compare with literature values, but that is not your problem, but rather a problem of the community as a whole. I do believe though, that it is worth pointing that fact out close to the experiments.
> >
> > Regarding the presentation:
> > - It is still stated only indirectly, that the proposed method works primarily for directed graphs (of course, every undirected graph can trivially be turned into a direct one - but at the moment no experiment and nothing in the related work section talks about undirected graphs, so one can make it more clear that the focus lies on directed graphs)
> > - the multi-scale analogy to CNNs is not proper. Of course there is a connection to the stride of CNNs but for k-hop with k>2 "backwards" edges are also included in contrast to CNNs. Also the graph size is not modified while a CNN does that as its core feature, only this way a U-Net can go down to a single long vector which is then expanded back. This is clearly not possible with ScaleNet (neither the encoding, nor the decoding).
> > - a ton of papers use either skip-connections or aggregate all layers in the end. The statement that "exactly k-hop neighbors" are looked at for a k-layer GNN is simply not true.
> >     - for undirected graphs and most standard GNNs, a skip connection adds the aggregation to the current state, while a self-loop adds the old state inside the nonlinearity. Both ways ensure that all information up to k hops away is available for classification. (Honestly, the details are not super important in most cases, but mentioning whether self-loops are added and whether skip connections/residual connections are used is often helpful)
> > - for the given paper, the related work section should contain both information on directed graphs (but much more limited), as well as information about the whole k-hop business. GTs must be mentioned to state why they are not relevant for the paper (I have absolutely no problem with that when its properly motivated).
> > - I am pretty sure that I have read about the strategy with the multiplication order before, even if I cannot state where exactly. I would be careful with the statement about UAT in 388 since your experiments clearly show that while theoretically possible, this is not learned.
> > - I don't see the mentioned paradoxiacal situation: noone is going to read the detailed related work in the appendix and the main paper does currently not mention that k-hop aggregation existed before. I find this a very serious oversight and one that is very easily resolved (as outlined above).
> >
> > I admit that the notation in the proofs is highly non-standard (the explanation with $X_i$ containing list of vectors was helpful and should be added to C1). It should thus be properly explained in C.1 which is currently a little lacking.
> > The proof of Theorem C.1 is currently incorrect as UAT needs a non-linearity and a hidden layer to be applied (at least the most commonly used version). This is possible for a GIN but not a standard GCN.
> > Also "learning" the normalization cannot work for the graph classification setting where different graphs are possible inputs. Integrating the normalization directly into W also does not work since the multiplication with W has the same effect on every node. Is there a simple fix for that?
> >
> > C.2.1 is highly dependent on an odd decision by the original GCN authors to not distinguish between the node itself and its neighbors, implying a lowered expressivity. It is worth mentioning, that the result does not extend to SAGE or cases with skip connections (in both cases, the network can learn to perform only a single round of aggregation and thus learn e.g. whether there is a blue node in the neighborhood).
> > So the core insight in this section is that GCN throws away its own state after each round of aggregation such that for this exact model, the statement holds. At least to me this sounds like it could be worth mentioning that the equivalence is brittle.
> >
> > C.2.2 and C.3.1: what effect does a dropout model have? Is it less expressive? Or more? Or incomparable? I am missing the conclusion of the theory section C. Without further explanation, I do not see how it would help explaining the good performance of ScaleNet on heterophilic datasets.
> >
> >
> > Thanks for fixing:
> > - the scale invariance definition (actually, it could be further simplified since one can always choose $k_1=1$ such that only a single $k>1$ has to be found)
> > - the "global" vs "long-distance" wording

---

> > > ### Comment · Reviewer_1jje · 2025-11-21
> > >
> > > **Not about content**:
> > > I would like to point out that more text does not always help, especially on openreview where it quickly gets too much. I tried to criticize the extensiveness/verboseness answers nicely, but clearly to no effect. Recall that both the AC and the other reviewers are also supposed to read the whole interaction (I was also pointed to another answer). It also does not help to offend reviewers by first overloading them with trivial statements (see e.g. matrix multiplication section) and then stating that they did not properly read through the whole interaction.
> > >
> > > I would also like to state that the "spade" comment was unprofessional. Lets try to get back to a professional tone.

---

> > > > ### Author Response · Authors · 2025-11-23
> > > > **Explanation of Rebuttal Organization to Reviewer 1jje**
> > > >
> > > > Regarding Reviewer 1jje’s comment that we *“offend reviewers by first overloading them with trivial statements,”* we would like to kindly clarify the structure and intention of our rebuttal.
> > > >
> > > > Because Reviewer 1jje raised a large number of points in the main review, we aimed to present our responses in a clear and organized manner to support the reviewer’s reading. To achieve this, we grouped Reviewer 1jje’s comments into thematic topics, each labelled with a capital letter (e.g., A. Definition of Scale Invariance, B. Critique of Three Models). Within each topic, we further divided the points into sub-items such as A1, A2, B1, B2, etc., with each sub-item quoting the reviewer’s comment verbatim and providing a direct response beneath it. We have no intention, nor opportunity, to overload the reviewer with *"trivial statements"*.
> > > >
> > > > In the subsequent round, Reviewer 1jje raised similar or identical questions that had already been addressed in our structured responses. When we pointed this out, the reviewer claimed us *“offend reviewers by first overloading them with trivial statements,”* and explained the situation as follows:
> > > > > *“I tried to criticize the extensiveness/verboseness answers nicely, but clearly to no effect.”*
> > > >
> > > > We sincerely apologize if our earlier replies appeared overly detailed; our only intention was to answer each of the reviewer’s many comments thoroughly and unambiguously. For example, the “trivial statement” in the matrix multiplication section that Reviewer 1jje mentioned was included intentionally, as we explained in our previous response, to clarify how the scaling occurs in our models—addressing the reviewer’s original comment that “no scaling happened.”
> > > >
> > > > We have always aimed to respond with clarity, completeness, and respect, particularly when certain comments seemed to require careful explanation to avoid misunderstanding. While the reviewer interpreted this as “extensiveness/verboseness,” our intention was never to be verbose but rather to provide precise and helpful clarifications.
> > > >
> > > > We hope this explanation of our rebuttal organization helps streamline the review process and makes it easier for the reviewer to navigate and understand, and we remain grateful for the reviewer’s time and effort.

---

> > > ### Author Response · Authors · 2025-12-02
> > > **V: Encoding and Decoding**
> > >
> > > > Reviewer 1jje: the multi-scale analogy to CNNs is not proper. Of course there is a connection to the stride of CNNs but for k-hop with k>2 "backwards" edges are also included in contrast to CNNs. Also the graph size is not modified while a CNN does that as its core feature, only this way a U-Net can go down to a single long vector which is then expanded back. This is clearly not possible with ScaleNet (neither the encoding, nor the decoding).
> > >
> > > Regarding Reviewer 1jje’s remarks, we would like to clarify two points:
> > > (1) why the edges are not “backwards”, and
> > > (2) how encoding and decoding naturally follow from scale invariance.
> > >
> > > **(1) No *"backwards"* edges**
> > >
> > > The reviewer’s concern seems to arise from confusion between the ego-graph and the scaled ego-graph. All our discussions are based on ego-graphs, and the scaled ego-graphs are extracted from these ego-graphs. For each type of scaled edge, the neighborhood is clearly defined within the scaled ego-graph. As the scale increases, the corresponding scaled neighbors are aggregated, and therefore no *backwards* edges are introduced.
> > >
> > > For $k = 3$, a 3-scaled *out–in–out* path may reach the same node that is reachable by a 1-scaled *out* edge. This is **not** a reversal of direction. Rather, it reflects the fact that the 3-scaled (*out–in–out*) neighborhood strictly contains the 1-scaled (*out*) neighborhood.
> > >
> > > This behavior is expected and consistent with scale-based neighborhood expansion.
> > >
> > > We explicitly discussed related cases in Appendix F.2 (Generated Self-loops). For a 2-scaled *out–in* path, certain nodes map back to themselves, effectively creating a self-loop in the 2-scaled neighborhood. For example, in the structure 1→2←3, the *out–in* neighborhood of node 1 includes node 3 **and** node 1 itself, because the path 1→2←1 is valid.
> > >
> > > Had this section been read, the concern about “backwards” edges would likely not have arisen. These are not reversed edges but natural consequences of path composition in directed graphs.
> > >
> > >
> > >
> > >
> > > **(2) Encode and Decode**
> > >
> > > We sincerely appreciate the reviewer’s interest in extending ScaleNet toward U-Net–style architectures. Below, we clarify how encoding and decoding can be performed in this context.
> > >
> > >
> > > **Encoding** is already present in ScaleNet: we move from 1-scaled to $k$-scaled representations. For a 2-layer GNN, the 1-scaled version includes only 1-hop and 2-hop neighborhoods, whereas the $k$-scaled version contains $k$-hop and $2k$-hop neighborhoods. This can be viewed as sparsifying the graph while expanding the receptive field.
> > >
> > > **Decoding** reconstructs a 1-scaled ego-graph from the $k$-scaled ego-graph using learnable upsampling operations.
> > >
> > > Following the U-Net design, the original 1-scaled ego-graph is copied and concatenated with the decoded 1-scaled ego-graph to form the final combined representation.

---

> ### Author Response · Authors · 2025-11-21
> **Dear Reviewer 1jje,**
>
> Dear Reviewer 1jje,
>
> We apologize for any discomfort caused by the expression “call a spade a spade”. We had no intention of offending you.
>
> Regarding your concern that we “offend reviewers by first overloading them with trivial statements,” such as those in the matrix multiplication section, our intention was solely to respond to your earlier comment that *“no scaling is happening”*. We included that section to explicitly clarify the mathematical steps underlying the scaling effect, not to overwhelm the reader.
>
>
> Best Regards,
>
> The Authors

---

> ### Author Response · Authors · 2025-11-24
> **U: Improving the Presentation and Context(U1-U3)**
>
> We thank Reviewer 1jje for their constructive re-evaluation, which confirms the soundness of our methodology and its strong empirical performance. We also sincerely value the reviewer’s constructive suggestions for improving the quality and clarity of the paper.
>
> Below, we provide detailed responses to each suggested improvement.
>
> >**(U1)** Re: proposed method works primarily for directed graphs...so one can make it more clear that the focus lies on directed graphs
>
> Thank you for this suggestion. We have revised the Introduction to make our focus on directed graphs clearer, as follows:
> >While most existing multi-scale GNN models are designed for undirected graphs (see Appendix D.4.3), recent work (Rossi et al., 2024) has shown that edge directionality can substantially improve performance on heterophilic graphs. Although some GNN architectures for directed graphs do incorporate information from multiple hop distances, they do not explicitly recognize multi-scale learning as the unifying principle.
>
> In addition, we note that the baseline models and Graph Transformer variants in our experiments are evaluated on undirected graphs because these models typically perform better when directed graphs are symmetrized. For this reason, we keep the experimental setting general, while our methodology explicitly accounts for graph directionality.
>
> Overall, our work focuses on general graphs, with particular attention to how directional information can be leveraged to enable effective multi-scale learning.
>
> >**(U2)** Re: k-hop with k>2 "backwards" edges are also included in contrast to CNNs. ...,only this way a U-Net can go down to a single long vector which is then expanded back. This is clearly not possible with ScaleNet (neither the encoding, nor the decoding).
>
> We thank the reviewer for the comment regarding “backwards” edges in k-hop neighborhoods. While this occurs in raw k-hop aggregation, our analogy to CNNs is made on the **ego-graph of each node**, not the original graph (which likely led to the reviewer’s concern), as explained in the first paragraph of **Section 3.3** and illustrated in **Figure 1(b)**. The scaled ego-graphs used by ScaleNet are structured representations without such backward paths, forming a definite multi-scale hierarchy analogous to spatial scaling in images.
> The reviewer's confusion: for k=3, a 3-scaled path of *out–in–out* may lead to the same node as a 1-scaled *out* edge. This is not *"backwards"*, it indicates that the 3-scaled (*out-in-out*) neighborhood include 1-scaled (out) neighborhood.  More details are provided in **V: Encoding and Decoding**.
>
> >**(U3)** Re: The statement that "exactly k-hop neighbors" are looked at for a k-layer GNN is simply not true.
> >Re: a skip connection adds the aggregation to the current state, while a self-loop adds the old state inside the nonlinearity.
> >Re: mentioning whether self-loops are added and whether skip connections/residual connections are used is often helpful
> >Previous question: It would have been nice to discuss the connection between self-loops and skip-connections.
>
> Thank you for providing more details about skip-connections. Now I understand your previous question better, and I take what you mean about skip-connections as in this paper [1], which we referred to in our paper as "Jumping Knowledge."
>
> For a k-layer GNN adding self-loops, the statement that it aggregates "exactly k-hop neighbors" still holds true. There can be all kinds of loops in the k-hop paths [2], such as self-loops, undirected edges (two-node loops), etc. As you correctly noted, *"self-loop adds the old state"*, because with self-loops, a 1-hop neighbor is also considered a k-hop neighbor (for any $k\neq1$), by adding (k-1) self-loop hops. (We believe you prefer succinct answers, so we omit an example and a matrix representation here. However, we are happy to provide them if needed.)
>
> In Table 2, the last line but one, we explicitly mention adding self-loops in our best model hyperparameters. In Section 4.1.2, we mentioned Jumping Knowledge as an option:
> > combines outputs across layers, using techniques such as Jumping Knowledge (JK) (Xu et al., 2018) or simply the final layer’s output.
>
> In the Appendix, we also list Jumping Knowledge aggregation and adding self-loops as hyperparameters.
>
> Regarding the connection between self-loops and skip-connections, this is very insightful as you highlighted: self-loops add the old state inside the nonlinearity, while skip-connections add states outside the nonlinearity.
>
> Node embedding formulas:
> - Adding self-loops:  $\sigma(A+I)^kXW$, where $I$ is the identity matrix
> - Ski-connections: $\sigma(AXW_1)+ \sigma(A^2XW_2) + ... +\sigma(A^kXW_k)$
> ---
> [1] Keyulu Xu et al., Representation learning on graphs with jumping knowledge networks. In International conference on machine learning.
>
> [2] Qin Jiang, et al.. Demystifying mpnns: Message passing as merely efficient matrix multiplication, 2025.

---

> ### Author Response · Authors · 2025-11-24
> **U: Improving the Presentation and Context(U4-U8)**
>
> >**(U4)** Re: the related work section should contain both information on directed graphs (but much more limited), as well as information about the whole k-hop business. GTs must be mentioned to state why they are not relevant for the paper (I have absolutely no problem with that when its properly motivated).
>
> Thank you very much for the suggestions to improve the presentation. We have expanded the related work section by adding discussions on multi-scale neighborhood aggregation and Graph Transformers, as recommended. We sincerely appreciate your guidance and support.
>
> ---
>
> >**(U5)**  Re: I would be careful with the statement about UAT in 388 since your experiments clearly show that while theoretically possible, this is not learned.
>
> Thank you for pointing this out. There was confusion because we were not clear enough in the original manuscript, and the learned linear matrix is $W$. Here is the new revision:
> > Instead of computing $\widetilde{AA}X$, it computes $\tilde{A}(\tilde{A}X)$. While these operations are not numerically identical, they preserve the same sparsity patterns and connectivity structures. Thus, the node embedding changes from $\sigma(\widetilde{AA}XW)$ to $\sigma(\tilde{A}(\tilde{A}X)W)$. By the Universal Approximation Theorem (UAT), a sufficiently expressive linear transformation $W$ could compensate for the numerical differences introduced by multiplication reordering, making the two approaches functionally equivalent for feature aggregation.
>
> We also found two papers that formally analyze and prove multi-scale models[3, 4].
> Thank you for your support.
>
> ---
>
> >**(U6)**   Re: I don't see the mentioned paradoxiacal situation: noone is going to read the detailed related work in the appendix and the main paper does currently not mention that k-hop aggregation existed before.
>
> Thank you for the comment. We have now moved the review of multi-scale models from the appendix into the Related Work section as requested. In the previous version, we placed it in the appendix due to space limits.
>
> ---
>
> >**(U7)** Re: the explanation with $X_i$ containing list of vectors was helpful and should be added to C1...
> >Re: The proof of Theorem C.1 is currently incorrect as UAT needs a non-linearity and a hidden layer to be applied...
> >Re: Also "learning" the normalization cannot work for the graph classification setting where different graphs are possible inputs. Integrating the normalization directly into W also does not work since the multiplication with W has the same effect on every node. Is there a simple fix for that?
>
> Thank you for the helpful comments. We have added the dimension of $X_i$ in Appendix C.1.
> Regarding non-linearity: as noted earlier, our models do include non-linear activations, but we had omitted them in the notation for simplicity. Since you raised this point again, we now explicitly include all non-linearities in the proofs for clarity.
>
> On normalization: there may have been a misunderstanding. We are not attempting to learn the normalization matrix. You are correct that, at the local level, $W$ acts identically on all nodes while the normalization $N$ is node-specific. However, this does not imply that learning precise edge weights is necessary.
> Empirically, replacing the original edge weights with random values in [0.0001,10000]yields similar performance on most datasets, suggesting that exact edge weights play only a limited role. A simple intuition example:  in CiteSeer, node features are one-hot word indicators, so the presence of specific words largely determines classification, while edge-weight magnitudes matter little. For further details, see Paper [4].
>
> We would be grateful if you could clarify further if this does not address your concern.
>
> ---
>
> >**(U8)** Re: original GCN authors to not distinguish between the node itself and its neighbors, implying a lowered expressivity. It is worth mentioning, that the result does not extend to SAGE or cases with skip connections
>
> Thank you for your insightful comment regarding the limitation of GCN. In our paper, we consider two versions of GCNs: GCN with self-loops (the original GCN authors’ model), which is discussed in Appendix C.2, and GCN without self-loops, which is discussed in Appendix C.3. All proofs in the appendix are divided according to these two cases.
> With this distinction, many existing models can be viewed as variants of these two basic types, obtained by replacing certain components or adding additional modules:
> - **SAGE** can be viewed as a variant of the mean-normalized GCN without self-loops, augmented by concatenating the self-features; and
> - **skip-connection models** can be understood as combinations of multiple GCN layers.
>
> Here is SAGE:  $\tilde A XW_1+XW_2$, we can add proof of scale invariance of SAGE in the appendix if you find it helpful.
>
>
> [4 ]Qin Jiang, Chengjia Wang, Michael Lones, and Wei Pang. Demystifying mpnns: Message passing as merely efficient matrix multiplication, 2025.

---

> ### Author Response · Authors · 2025-11-24
> **U: Improving the Presentation and Context(U9)**
>
> >**(U9)** Re: what effect does a dropout model have? Is it less expressive? Or more? Or incomparable? I am missing the conclusion of the theory section C. Without further explanation, I do not see how it would help explaining the good performance of ScaleNet on heterophilic datasets.
>
> We performed a preliminary experiment (1 split) on Chameleon to study the effect of dropout on ScaleNet compared with Dir-GNN. The results are summarized below:
>
> | dp  | 0.0   | 0.1   | 0.2   | 0.3   | 0.4   | 0.5   | 0.6   | 0.7   | 0.8   | 0.9   |
> | --- | ----- | ----- | ----- | ----- | ----- | ----- | ----- | ----- | ----- | ----- |
> | ScaleNet | 78.29 | 77.19 | 79.17 | 77.19 | 78.29 | 78.51 | 78.07 | 78.07 | 78.29 | 79.17 |
> | Dir-GNN | 80.48 | 80.04 | 79.39 | 79.17 | 78.07 | 77.85 | 77.19 | 76.54 | 77.85 | 74.78 |
>
> From above, the comparison in mean and std:
> | Model    | Mean Acc (%) | Std (%) |
> | -------- | ------------ | ------- |
> | ScaleNet | 78.33        | 0.77    |
> | Dir-GNN  | 78.34        | 1.83    |
>
> The mean accuracy is almost identical, but ScaleNet is much more robust (lower std), while Dir-GNN is more variable, especially at high dropout. This suggests that ScaleNet’s multi-scale learning enables it to maintain stable performance under high dropout, contributing to its robustness.
>
> ---
>
> We hope the responses above address all concerns. We would like to highlight the productive discussions with Reviewer 1jje:
>
> - During Reviewer 1jje's challenge on our models' performance, we investigated and discovered a mismatch of datasets used by the directed graph and undirected graph research groups.
>
> - During the challenge on scale invariance and MLP, we extended the MLP to 0-scaled graphs.
>
> - Regarding the challenge on "global" and "local," we rephrased our wording to "long-range."
>
> - After Reviewer 1jje challenged the Graph Transformer and encouraged us to add it in main paper, we improved the related work section by including Graph Transformer, which helps this paper present a more complete picture of graph learning.
>
> We especially appreciate the reviewer for re-evaluating our paper and confirming the validity of our methods and the effectiveness of our models.
>
>
>
> [1] Keyulu Xu, Chengtao Li, Yonglong Tian, Tomohiro Sonobe, Ken-ichi Kawarabayashi, and Stefanie Jegelka. Representation learning on graphs with jumping knowledge networks. In International conference on machine learning, pp. 5453–5462. PMLR, 2018.
>
> [2] Qin Jiang, Chengjia Wang, Michael Lones, and Wei Pang. Demystifying mpnns: Message passing as merely efficient matrix multiplication, 2025.
>
> [3] Ke Sun, Zhanxing Zhu, and Zhouchen Lin. AdaGCN: Adaboosting Graph Convolutional Networks into Deep Models.
>
> [4] Felix Wu, Amauri Souza, Tianyi Zhang, Christopher Fifty, Tao Yu, and Kilian Weinberger. Simplifying graph convolutional networks. In International conference on machine learning, pp.6861–6871. Pmlr, 2019

---

### Official Review · Reviewer_c5V6 · 2025-10-28

**Soundness:** 2
**Presentation:** 3
**Contribution:** 2
**Rating:** 6
**Confidence:** 3

**Summary:**

The paper argues that most GNNs implicitly operate at a single “first-order” scale (fixed k-hop depth), while real graphs benefit from multi-scale representations—analogous to multi-resolution features in vision. It formally introduces scale invariance for node classification, gives theory and empirical evidence for it, and proposes ScaleNet, a message-passing architecture that aggregates directed multi-scale features and uses an adaptive self-loop mechanism. ScaleNet attains SOTA on six medium-scale benchmarks (homophilic & heterophilic). To scale to million-node graphs, the authors develop LargeScaleNet, which avoids explicit higher-order adjacency products and achieves new SOTA on three large benchmarks. They also reinterpret several spectral/digraph models through the lens of message passing, attributing their gains largely to multi-scale feature integration.

**Strengths:**

The presentation and demonstration is easy to understand. The notations are well defined

**Weaknesses:**

See below

**Questions:**

1. Are the in neighbours and out neighbours predefined in the data or you select it with some algorithm? Does this process consider the different label distributions of different tasks?
2. I still cannot see why do we have to "leverage the scale invariance in node classification"? What is its advantage, especially in heterophilic setting, as it is found that expanding the scale is harmful in many heterophilic datasets [1].
3. The whole story of this paper can actually be applied to undirected graph as well. If the authors want to highlight the contribution of scale invariance principle, why not try it on undirected graph? Applying it on directed graph makes the proposed model look like a simple extension of Dir-GNN.
4. It seems that your proposed model add lots of hyperparameter to fine-tine.
5. Did other baseline models apply 10-fold cross validation as well?
6. Missing related work and baselines for multi-scale GNNs, e.g. LanczosNet [2] and Truncated Krylov Network [3]
7. The performance on heterophilic graphs are only tested on benign heterophilic graphs, which is not enough to show the effectiveness of your proposed method on heterophily problem. It's better to test on the malignant and ambiguous heterophilic datasets listed in [4]



[1] Less is More: on the Over-Globalizing Problem in Graph Transformers. In Forty-first International Conference on Machine Learning 2024.

[2] LanczosNet: Multi-Scale Deep Graph Convolutional Networks. In ICLR 2019.

[3] Break the ceiling: Stronger multi-scale deep graph convolutional networks. Advances in neural information processing systems. 2019;32.

[4] The heterophilic graph learning handbook: Benchmarks, models, theoretical analysis, applications and challenges. arXiv preprint arXiv:2407.09618. 2024 Jul 12.

---

> ### Author Response · Authors · 2025-11-20
> **Response to Reviewer c5V6(1-3)**
>
> Thank you to the reviewer for the clear summary of our paper and for acknowledging that the presentation is easy to understand and that our notations are well defined. We appreciate the reviewer’s assessment and the constructive feedback provided. We provide point-by-point responses below.
>
> >**(1)** Re: Are the in neighbours and out neighbours predefined in the data or you select it with some algorithm? Does this process consider the different label distributions of different tasks?
>
> Thank you very much for asking this question.
> The in neighbours and out neighbours are predefined in the dataset. As the dataset would provide nodes and edges. each edge is denoted with a tuple with two element, the first element is the source node and the second the target node. For the source node, the target node is one of its out-neighbor, vise verse, for the target node, the source node is its in-neighbor.
> In this paper, for all datasets, each node only has one label, thus there is no different tasks in this paper.
>
> >**(2)** Re: I still cannot see why do we have to "leverage the scale invariance in node classification"? What is its advantage, especially in heterophilic setting, as it is found that expanding the scale is harmful in many heterophilic datasets[1]
>
> Thank you for sharing this insightful paper . We carefully read the paper you suggested, which highlights the Graph Transformer problem in over-globalization, bringing more noise than beneficial information, leading to worse performance. However, our expanded aggregation is structured-expanding along directed scale of our choice-unlike Graph Transformer, which aggregate blindly. Below, we provide a more detailed explanation of why our approach works, and still works for heterophilic graphs.
>
> - Benefits of Multi-Scale Learning:
> The idea is analogous to CNNs in computer vision: leveraging multiple scales allows combining fine-grained local features with coarser global features, improving classification accuracy (see Line 251 in the original manuscript). Similarly, in graphs, multi-scale learning can capture both local and broader structural information.
>
> - Scale Invariance and Heterophily:
> Scale invariance, as defined in Definition 3.5 of our paper, is independent of homophily or heterophily. It means that there exist at least two scales that are individually predictive, and combining them can improve performance because their predictive benefits might be complementary.
>
> Heterophilic graphs are characterized by nodes differing from their immediate neighbors. However, distant nodes (e.g., neighbors of neighbors) can share class labels:
> >"in heterophilous graphs, linked nodes are more likely to be dissimilar while distant nodes could share a certain similarity"[1]
>
> Thus, expanding the scale is often useful or even necessary in heterophilic graphs, because higher-order neighborhoods capture these longer-range dependencies.
> > "Generally, these non-homophilous GNNs enhance the performance of GNNs through the following three main designs: (1) Using High-Order Neighborhoods. ..."[2].
>
> Using high-order neighborhoods is what you mean by expanding the scale.
>
> In our experiments on five heterophilic datasets (Chameleon, Squirrel, Roman-Empire, arxiv-year, and snap-patents), all benefit from multi-scale learning, confirming the practical value of scale invariance even in heterophilic settings.
>
>
> >**(3)** Re: The whole story of this paper can actually be applied to undirected graph as well. If the authors want to highlight the contribution of scale invariance principle, why not try it on undirected graph? Applying it on directed graph makes the proposed model look like a simple extension of Dir-GNN.
>
> We focus on directed graphs because edge direction is crucial for prediction in many tasks. While high-order neighborhood aggregation for undirected graphs has been extensively studied (see Appendix D4.3), our work is, to the best of our knowledge, the first to present multi-scale learning specifically for directed graphs, achieving significantly better performance than undirected counterparts. For more details about our difference from other multi-scale models, please refer to our responds to Reviewer mGZC, Q1. We appreciate your support.
>
> Our work builds upon the Dir-GNN seminal model, which showed strong performance on heterophilic graphs. We extend it in two key ways:
> For Homophilic graphs: We incorporate self-loops and provide a theoretical analysis of their effect, whereas Dir-GNN, which does not consider self-loops, shows poor performance. For Heterophilic graphs: We introduce stacked multi-scale learning to leverage complementary information across scales, achieving even better performance than the Dir-GNN seminal model.
>
> In short, our model leverages the strengths of Dir-GNN while improving on both homophilic and heterophilic graphs, rather than being a simple extension.

---

> ### Author Response · Authors · 2025-11-20
> **Response to Reviewer  c5V6 (4-8)**
>
> >**(4)** Re: It seems that your proposed model add lots of hyperparameter to fine-tine.
>
> We acknowledge the increase in hyperparameters (e.g., choice of directed scales α, β, γ). We provide extensive details on our fine-tuning procedures and budgets in our response to Reviewer JKmj and Appendix B.4.2/B.4.3.
>
> >**(5)** Re: Did other baseline models apply 10-fold cross validation as well?
>
> Yes, all experiments use the same train/validation/test splits. For medium-sized datasets, we applied 10-fold cross-validation. For large graphs, such as SNAP-Patents and arxiv-year, we followed the original sources, which provide only 5 splits, to ensure comparability with prior work. Detailed dataset information can be found in Appendix B.4.2 and B.4.3.
>
>
> >**(6)** Re: Missing related work and baselines for multi-scale GNNs, e.g. LanczosNet [2] and Truncated Krylov Network [3]
>
> Thank you for these valuable suggestions. We have included LanczosNet and Truncated Krylov Network in the revised manuscript in our review of HIGHER-ORDER NEIGHBORHOOD AGGREGATION in the Appendix D.4.3.
>
>
>
>  >**(7)** Re: The performance on heterophilic graphs are only tested on benign heterophilic graphs, which is not enough to show the effectiveness of your proposed method on heterophily problem. It's better to test on the malignant and ambiguous heterophilic datasets listed in [4]
>
> Among the nine datasets we evaluate on (Tables 2 and 4), five—Chameleon, Squirrel, Roman-Empire, arXiv-year, and snap-patents—are heterophilic graphs [3]. Notably, Roman-Empire is categorized as **malignant** and snap-patents as **ambiguous** heterophilic datasets in [4]. Therefore, our experiments already include the malignant and ambiguous settings the reviewer requested, and the strong SOTA performance on these datasets further demonstrates the effectiveness of our method.
>
>
>
>  >**(8)** Re: soundness scored 2
>
> We noticed that the reviewer scored 2 for the soundness of our paper. We believe this concern was mainly due to the original definition of scale invariance, which was indeed too strong.
>
> Specifically, the original definition might have been interpreted as requiring any possible scale to maintain the classification outcome. Our revised definition, which is grounded in practical observations of GNN behavior, shifts this requirement to the existence of at least two distinct scales that provide a consistent classification outcome.
> We have updated the manuscript in Definition 3.5.
> This correction aligns with practical observations and strengthens the conceptual soundness of our framework.
>
>
>
> References:
> [1] Xiang Li, et al. Finding Global Homophily in Graph Neural Networks When Meeting Heterophily, ICML 2022.
>
> [2] Ruihao Zhang, et al. Discovering Invariant Neighborhood Patterns for Non-Homophilous Graphs, NeurIPS 2023.
>
> [3] Christian Koke and Daniel Cremers. Holonets: Spectral convolutions do extend to directed graphs. In The Twelfth International Conference on Learning Representations, 2024.
>
> [4] The heterophilic graph learning handbook: Benchmarks, models, theoretical analysis, applications and challenges. arXiv preprint arXiv:2407.09618. 2024 Jul 12.
>
>
> We thank the reviewer again for their rigorous review and hope that these revisions satisfactorily address their comments.
>
>
> Warm Regards
> The authors

---

### Official Review · Reviewer_JKmj · 2025-11-01

**Soundness:** 3
**Presentation:** 3
**Contribution:** 3
**Rating:** 6
**Confidence:** 4

**Summary:**

The paper proposes ScaleNet and LargeScaleNet, two message-passing GNNs that achieve scale awareness by aggregating information across multiple hop distances and edge directions. The authors formalize scale invariance (the idea that node predictions remain stable across scaled egographs) and design ScaleNet to fuse features from first- and second-order adjacency products. LargeScaleNet improves scalability by replacing explicit adjacency multiplications with sequential sparse operations. Across six medium and three large benchmarks, the proposed models achieve or match SOTA performance and are supported by ablations and Wilcoxon tests.

**Strengths:**

1. Elevates multi-scale aggregation to a unifying principle for directed GNNs. The architecture is straightforward to implement and reason about.
2. The empirical experiments demonstrate consistently higher/tie accuracy with diverse baseline.
3. Achieves practical scalability through LargeScaleNet and enable efficient training on million-node graphs.
4. Reinterprets spectral methods from a message-passing perspective.

**Weaknesses:**

1. The discussion of scale invariance theory in Appenidx C removes nonlinearities and uses the UAT to justify reordering operations. This shows that the model can represent similar functions, but not necessarily that trained solutions remain invariant. It would be helpful to
include a clearer assumptions and proofs, especially when including activations/normalization.
2. Hyperparameter selection over (α, β, γ), self-loops, and JK variants in Appendix B involves grid search per-dataset. Would be helpful to clarify search budgets and ensure baselines receive comparable tuning.
3. The paper’s reporting focuses on accuracy, while more runtime/memory numbers would clarify the practical impact of LargeScaleNet vs. alternatives.

**Questions:**

1. Could you state the scale invariance results with explicit conditions on activations/normalizations and clearly separate exact equalities from UAT-based approximations?
2. How were search budgets for α, β, γ, JK, and self-loops allocated, and did baselines receive comparable per-dataset tuning?

---

> ### Author Response · Authors · 2025-11-19
> **Response to Reviewer JKmj**
>
> Thank you to the reviewer for confirming our strengths in elevating multi-scale aggregation to a unifying principle for directed GNNs, achieving consistently strong accuracy, enabling practical scalability with LargeScaleNet, and offering a message-passing view of spectral methods. We appreciate the positive assessment of our conceptual and empirical contributions.
>
> Here we are clarifying some issues raised by the reviewer.
>
> - >Re:  removes nonlinearities
> - >Re: Could you state the scale invariance results with explicit conditions on activations/normalizations and clearly separate exact equalities from UAT-based approximations?
>
> We apologize for the confusion. Our models do **not** remove nonlinearities—they were omitted in Appendix C solely for notational simplicity. Reviewer 1jje raised a similar concern, which we addressed in Comment B2. Although this simplification was mentioned multiple times in the original manuscript, we have now revised the paper to explicitly include nonlinearities in all formulas to avoid any further misunderstanding.
>
> With this clarification, we believe the second question is addressed. We are happy to elaborate further if needed.
>
> - >Re: Hyperparameter selection over (α, β, γ), self-loops, and JK variants in Appendix B involves grid search per-dataset. Would be helpful to clarify search budgets and ensure baselines receive comparable tuning.
> - >Re: How were search budgets for α, β, γ, JK, and self-loops allocated, and did baselines receive comparable per-dataset tuning?
>
> Thank you for raising this question. The full grid search ranges for all hyperparameters are listed in Appendix B.4.4. The search options for α, β, γ are {0, 0.5, 1, 2, 3, -1}. Taking α as an example:
> (1) 0 corresponds to aggregation via $A$;
> (2) 1 corresponds to aggregation via $A^\top$;
> (3) 0.5 corresponds to aggregating via both $A$ and $A^\top$;
> (4) 2 corresponds to aggregating via the union of  $A$ and $A^\top$;
> (5) 3 corresponds to aggregating via the intersection of $A$ and $A^\top$;
> (6) -1 corresponds to no aggregation in this scale.
>
> To reduce the computational budget during grid search, we used practical shortcuts, such as running only a single split and applying early stopping after 50 or 100 epochs without improvement. The top three hyperparameter combinations were then used for full evaluation to determine the best configuration.
>
> Furthermore, we further reduce the search space for β and γ by conditioning their ranges on the best value of α. For example, if α performs best at 0.5—meaning the dataset benefits from using both directions—we restrict β and γ to {-1, 0.5}.  If α performs best at 0 or 1, we restrict β and γ to options consistent with those directions (e.g., {0,-1} or {1,-1}, respectively)· This conditional narrowing significantly reduces search cost while maintaining fairness and consistency.
>
>
> All baselines were tuned using the same grid search strategy, including the same hyperparameter ranges and practical shortcuts. For baselines designed for undirected graphs, we also tested converting directed graphs into undirected ones, which generally improved performance compared to the original directed graphs.
>
> - >Re: The paper’s reporting focuses on accuracy, while more runtime/memory numbers would clarify the practical impact of LargeScaleNet vs. alternatives.
>
> Thank you for raising this question. We address all runtime- and memory-related concerns in Q4 of our response to Reviewer mGZC. We have added both theoretical analysis and empirical comparisons of model complexity in Appendix G.
>
> We would be happy to answer any additional questions the reviewer may have.
>
>
> Best Regards,
>
> The authors

---

### Official Review · Reviewer_mGZC · 2025-11-01

**Soundness:** 3
**Presentation:** 3
**Contribution:** 3
**Rating:** 4
**Confidence:** 3

**Summary:**

This paper challenges the standard Graph Neural Network (GNN) paradigm of operating at a single, fixed $k$-hop scale, which is typically just a hyperparameter to be tuned. Inspired by the success of multi-scale representations in computer vision, the authors propose that GNNs can also benefit from explicitly integrating information across different scales.

The authors formalize scale invariance for graphs, and claim that a node should be classifiable from differently zoomed out graphs, and propose ScaleNet, which at each layer aggregates from six directed adjacency variants.

**Strengths:**

The paper's core premise is strong, clear, and intuitive. Moving beyond a fixed $k$-hop neighborhood and formalizing multi-scale learning as a core principle is a valuable conceptual contribution to the GNN field.

The paper does not just propose a complex model (ScaleNet) but also proactively addresses its scalability limitations.

**Weaknesses:**

While the paper idea is original, the major limitations in that the paper miss the discussion of other works that, with different ideas, share the same purpose as this paper. This include,

1/ Adaptive GNN that aggregate information for a node from different scales: Different paper have proposed to merge information at different levels using adaptive GNNs.

[1] Chien, Eli, et al. "Adaptive universal generalized pagerank graph neural network." arXiv preprint arXiv:2006.07988 (2020).

[2] Sun, Ke, Zhanxing Zhu, and Zhouchen Lin. "Adagcn: Adaboosting graph convolutional networks into deep models." arXiv preprint arXiv:1908.05081 (2019).


2/ Centrality Graph Shift Operator: I recently read [3], and found that the authors normalize the adjacency matrix with global centrality metric (instead of degree matrix) via a learnable Graph Shift Operator (GSO), and even create a GSO that merge both local and global centrality metric. Since you start your discussion about some graph filters, would be nice to discuss the difference of points of view.

[3] Abbahaddou, Yassine, et al. "Centrality Graph Shift Operators for Graph Neural Networks." arXiv preprint arXiv:2411.04655 (2024).

3/ Would be also nice to include a discussion about Graph Transformers, which can capture all ranges of interaction via the full attention, and explain cases where keeping the MPNN is better using your technique.

I think comparison, or at least a discussion of these paper will strength the paper.

**Questions:**

I cannot find any analysis of the time complexity, and how you method affect the training and test time.

I am happy to update my score depending your answers.

---

> ### Author Response · Authors · 2025-11-19
> **Q1: difference from other multi-scale models**
>
> We sincerely thank the reviewer for their thoughtful evaluation and for recognizing the key strengths of our work. We are glad that the reviewer found the core premise of our paper—moving beyond a fixed 1−hop neighborhood and formalizing multi-scale learning—as both strong and intuitive. We also appreciate the acknowledgment of our efforts to not only propose ScaleNet but also address its scalability considerations.
>
> Below, we provide our responses to the questions raised by the reviewer.
>
> - >**(Q1)** Re: difference from other multi-scale models
>
> Thank you for providing additional references on multi-scale learning. We have GPR-GNNS[2] in our original review in Appendix, and We have added ADAGCN[1] to our discussion of Higher-Order Neighborhood Aggregation (Appendix D.4.3 in the revised manuscript).
>
> (1) ADAGCN[1] is designed for undirected graphs, and therefore their experiments are conducted on homophilic datasets. For heterophilic graphs, where the choice of edge direction is crucial, their approach performs poorly.
> For instance, on Chameleon and Squirrel, using $A^lX$ performs very badly because many nodes have only out-edges and no in-edges. In these cases, using ${(A^{\top})} X$ gets best performance. If the directed graphs are converted into undirected graph, ADAGCN[1] is using ${(A+A^{\top})} ^lX$, , which is still worse than ${(A^{\top})} ^lX$.
>
> Our experiments with GCN using different adjacency matrices illustrate this effect:
>
> | Dataset    | A       | Aᵀ      | A + Aᵀ   |
> |------------|---------|---------|-----------|
> | Chameleon  | 28.9±1.5 | 78.1±1.6 | 70.9±1.8 |
> | Squirrel   | 28.4±1.5 | 73.5±2.0 | 59.9±2.2 |
>
> For higher-order propagation, the choice of direction remains essential.
>
> (2) GPR-GNNS[2] also operate on undirected graphs, and their reported performance on Chameleon and Squirrel is substantially worse than ours:
>
> | Model      | Chameleon      | Squirrel       |
> |------------|----------------|----------------|
> | GPR-GNNs   | 67.48 ± 0.40   | 49.93 ± 0.53   |
> | ScaleNet   | 80.1 ± 1.5     | 76.0 ± 2.0     |
>
>
> In addition, similar to the PageRank-based DiGib model[3], GPR-GNNs include an unnecessary trainable parameter—the GPR weights, which essentially act as learnable edge weights. As shown empirically in Appendix E.1, simply setting all edge weights to 1 is sufficient for strong performance.
>
> In summary, our model achieves superior performance compared to prior work, and does so with a straightforward architecture that avoids unnecessary parameterization.
>
> [1]  Sun, Ke, Zhanxing Zhu, and Zhouchen Lin. "Adagcn: Adaboosting graph convolutional networks into deep models." arXiv preprint arXiv:1908.05081 (2019).
>
> [2] Corso, G., Cavalleri, L., Beaini, D., Liò, P. and Veličković, P., 2020. Principal neighbourhood aggregation for graph nets. Advances in neural information processing systems, 33, pp.13260-13271.
>
> [3]  Zekun Tong, Yuxuan Liang, Changsheng Sun, Xinke Li, David Rosenblum, and Andrew Lim. Digraph inception convolutional networks. Advances in neural information processing systems, 33: 17907–17918, 2020a.

---

> ### Author Response · Authors · 2025-11-19
> **Q2: Graph Filters**
>
> - >**(Q2)** Re: I recently read [3], and found that the authors normalize the adjacency matrix with global centrality metric (instead of degree matrix) via a learnable Graph Shift Operator (GSO), and even create a GSO that merge both local and global centrality metric. Since you start your discussion about some graph filters, would be nice to discuss the difference of points of view.
>
> Thank you for the question.
>
> There is related work discussing graph filters: PNA[2]. PNA explores different normalization strategies(mean, min, max, standard deviation, and degree-based scalers) and theoretically explains when certain normalizations fail. Thus they present PNA, which is multiple aggregators, which is be more dataset-robust.
>
> Another recent paper[4] discussed four graph filters: GCN-norm, row-norm, dir-norm, mean-norm, and no-norm. They gave real world examples when some filter fails: when node features are the same, mean aggregator will make the nodes indistinguishable.
>
> In our view, let the GNN representation be written as $\sigma(N\cdot AXW)$, where $N$ is normalisation matrix, Normalisation is equivalent to dot multiplied by $N$.
> We see two key purposes for graph filters:
> (1) Prevent learning failure — ensure the filter does not make nodes indistinguishable.
> (2) Prevent numerical instability — avoid exploding or vanishing signals.
> Beyond these requirements, the exact values of the graph filter are less critical, as the linear layer $W$ can adjust for rescaling. Once these two requirements are satisfied, many normalization choices are viable.
>
> The paper [3] does not demonstrate any fundamental failure of existing normalizations; it proposes a new normalization. Its effectiveness may be dataset-specific rather than universally beneficial. However, it opens up a new possibility: combining local and global centrality metrics in a learnable way. In general, as long as normalization does not eliminate essential information or bringing numerical instability, it should work reasonably well.
>
> We hope this answers the question.
>
>
>
> [2] Corso, G., Cavalleri, L., Beaini, D., Liò, P. and Veličković, P., 2020. Principal neighbourhood aggregation for graph nets. Advances in neural information processing systems, 33, pp.13260-13271.
>
> [3] Abbahaddou, Yassine, et al. "Centrality Graph Shift Operators for Graph Neural Networks." arXiv preprint arXiv:2411.04655 (2024).
>
> [4]  Qin Jiang, Chengjia Wang, Michael Lones, and Wei Pang. Demystifying mpnns: Message passing as merely efficient matrix multiplication, 2025.

---

> ### Author Response · Authors · 2025-11-20
> **Q3: Discussion on Graph Transformers and when keeping MPNNs is better**
>
> - >**(Q3)** Re: Would be also nice to include a discussion about Graph Transformers, which can capture all ranges of interaction via the full attention, and explain cases where keeping the MPNN is better using your technique.
> I think comparison, or at least a discussion of these paper will strength the paper.
>
> Regarding Graph Transformers (GT), in our revised version we have included a discussion in the Appendix and a review of GT, as detailed in our response E1 to Reviewer 1jje.
>
> Thank you for your thoughtful question regarding when keeping MPNNs might be preferable using our technique. We are happy to provide an answer.
>
> Generally, reported performance is often based on GNNs with a specific number of layers. However, the optimal layer number may not be unique, which indicates that the graph exhibits some degree of scale invariance. Combining information from different scales can improve performance.
>
> For example, in Chameleon, using $A^\top$ as aggregation matrix, a 1-layer GCN achieves 78.1±1.6, while a 2-layer GCN (equivalent to $A^\top A^\top$) achieves 77.1 ± 1.3. Aggregating via both $A^\top$ and $A^\top A^\top$ within the same layer can increase performance to 80.1 ± 1.5.
>
> In summary, wherever there is scale invariance, there may be benefits to stacking multi-scale learning, which is precisely what our method leverages.
>
> We hope this addresses your insightful question.

---

> ### Author Response · Authors · 2025-11-20
> **Q4: time complexity**
>
> - >**(Q4)** Re: I cannot find any analysis of the time complexity, and how you method affect the training and test time.
>
>
> Thank you for this valuable question.
> We provide here a comparison of the main computational components of each model.
> For a GCN layer the dominant operations are:
> $AX$: neighborhood aggregation via matrix multiplication
> $XW$: linear transformation
> $N\cdot A$: normalization of the adjacency
> In addition, ScaleNet includes an extra adjacency transformation: $AA$.
>
> The per-layer relative operation counts are summarized below
> (within each row, operations are ordered from left to right approximately from heavier to lighter computation):
>
> | Model          | $AA$       | $AX$        | $XW$       | $N \cdot A$ |
> |----------------|----------|-----------|----------|---------------|
> | GCN            | 0        | 1         | 1        | 1             |
> | Dir-GNN        | 0        | 2         | 2        | 2             |
> | FaberNet        | 0        | 6         | 6        | 4             |
> | ScaleNet       | 4    | 6         | 6        | 6             |
> | LargeScaleNet  | 0        | 10 | 10    |  6           |
>
> This table reflects the relative number of required computations, and therefore accurately captures the increase in theoretical time complexity.
> From this breakdown, our scalable model LargeScaleNet requires roughly 10× more operations than a standard GCN per layer. For GCN time complexity, please refer to [1].
>
> We also benchmarked peak memory (epoch 1) and runtime averaged over epochs 2–11 (to exclude one-time setup overhead). All models use 2 layers.
> We have included the analysis and experiments on model complexity and runtime in the Appendix G.
>
>
> Thank you again for your valuable review.
> We would be happy to provide any additional evidence or clarification that may help address your concerns and assist in updating your score.
>
>
> Warm Regards,
>
> The authors
>
>
> [1] Derrick Blakely, Jack Lanchantin, Yanjun Qi, Time and Space Complexity of Graph Convolutional Networks.

---

### Meta-Review · Area_Chair_S5YW · 2026-01-06

**Summary:**

Multiple reviewers raised concerns about the novelty and conceptual framing, noting substantial overlap with prior multi-hop / multi-scale GNN literature. In addition, reviewers questioned the strength and correctness of the theoretical justification, particularly whether the theory meaningfully supports the proposed framing. These concerns, together with presentation and positioning issues that required extensive clarification during rebuttal, informed the decision.

**Reviewer Concerns:**

Addressed by the rebuttal:

* Clarification of dataset discrepancies and experimental validity.

* Runtime and scalability analysis.

* Expanded related work and clearer discussion of directed vs. undirected settings.

* Improved definitions and explanations of scale invariance and model components.

Still outstanding:

* Whether the method represents a substantive conceptual advance beyond existing multi-hop / multi-scale GNN approaches.

* Whether the theoretical analysis convincingly supports the central claims, including the assumptions under which the results hold.

* Overall maturity of framing and positioning at the time of submission, given the extent of post-hoc clarification required.

**Reviewer Scores:**

Reviewers would likely have maintained their original ratings. In particular, Reviewer 1jje would likely have remained below the acceptance threshold.

---

### Decision · Program_Chairs · 2026-01-26

Reject